# The non-hydrostatic global atmospheric model for CMIP6 HighResMIP simulations (NICAM16-S): experimental design, model description, and impacts of model updates

Chihiro Kodama[1], Tomoki Ohno[1], Tatsuya Seiki[1], Hisashi Yashiro[2], Akira T. Noda[1], Masuo Nakano[1], Yohei Yamada[1], Woosub Roh[3], Masaki Satoh[3,1], Tomoko Nitta[3], Daisuke Goto[2], Hiroaki Miura[4], Tomoe Nasuno[1], Tomoki Miyakawa[3], Ying-Wen Chen[3], and Masato Sugi[5]

[1]Japan Agency for Marine-Earth Science and Technology, Yokohama, 236-0001, Japan
[2]National Institute for Environmental Studies, Tsukuba, Ibaraki, 305-8506, Japan
[3]Atmosphere and Ocean Research Institute, The University of Tokyo, Kashiwa, 277-8564, Japan
[4]Department of Earth and Planetary Science, Graduate School of Science, The University of Tokyo, Tokyo, 113-0033, Japan
[5]Meteorological Research Institute, Tsukuba, 305-0052, Japan

*Correspondence to*: Chihiro Kodama (kodamac@jamstec.go.jp)

Submitted to *Geoscientific Model Development* (29 December 2019)

**Abstract.** The Non-hydrostatic ICosahedral Atmospheric Model (NICAM), a global model with an icosahedral grid system, has been under development for nearly two decades. This paper describes NICAM16-S, the latest stable version of NICAM (NICAM.16) modified for the Coupled Model Intercomparison Project Phase 6, High Resolution Model Intercomparison Project (HighResMIP). Major updates from NICAM.12, a previous version used for climate simulations, include updates of the cloud microphysics scheme and land surface model, introduction of natural and anthropogenic aerosols and a subgrid-scale orographic gravity wave drag scheme, and improvement of the coupling between the cloud microphysics and the radiation schemes. External forcings were updated to follow the protocol of the HighResMIP. A series of short-term sensitivity experiments were performed to determine and understand the impacts of these various model updates on the simulated mean states. The NICAM16-S simulations demonstrated improvements in the ice water content, high cloud amount, surface air temperature over the Arctic region, location and strength of zonal mean subtropical jet, and shortwave radiation over Africa and South Asia. Some long-standing biases, such as the double intertropical convergence zone and smaller low cloud amount, still exist or are even worse in some cases, suggesting further necessity for understanding their mechanisms and upgrading schemes and parameter settings, as well as for enhancing horizontal and vertical resolutions.

## 1. Introduction

Moist processes play a crucial role in the formation of the Earth's climate. Moist convection redistributes mass, energy, and momentum of the atmosphere to form large-scale circulation. Clouds are coupled with large-scale circulation through latent and radiative heating, which can affect climate sensitivity. The accurate treatment of such interactions between clouds and circulation requires high-resolution global cloud-resolving models (Bony et al., 2015; Satoh et al., 2019).

This, as well as an increasing demand from society to project extreme weather, such as tropical cyclones, motivated us to perform climate simulations of the present day and future using the Nonhydrostatic ICosahedral Atmospheric Model (NICAM) (Tomita and Satoh, 2004; Satoh et al., 2008, 2014) with a 14-km mesh. Kodama et al. (2015) and Satoh et al. (2015) provided brief descriptions of the model (hereafter referred to as NICAM.12) and experimental design of the climate simulations. This unique dataset of the high-resolution climate simulation, whose overall performance was reported in Kodama et al. (2015), has been used in many studies, with a focus on tropical cyclones (Satoh et al., 2015, 2018; Yamada et

al., 2017, 2019; Matsuoka et al., 2018; Sugi et al., 2020), extratropical cyclones (Kodama et al., 2019; McCoy et al., 2019; Satoh et al., 2018), intraseasonal oscillations, such as the Madden–Julian Oscillation (MJO) (Kikuchi et al., 2017; Nakano and Kikuchi, 2019), tropical synoptic-scale waves (Fukutomi et al., 2016), cloud radiative feedback (Chen et al., 2016; Satoh et al., 2018; Noda et al., 2019), and regional to global precipitation (Kilpatrick et al., 2017; Satoh et al., 2018; Na et al.,
2020; Takahashi et al., 2000).

Also, some significant climate biases have been identified in the simulation (Kodama et al., 2015), and great effort has been devoted to improving the model for better performance of the simulated climate in a physically consistent manner. Major updates between NICAM.12 and NICAM.16, a stable version of NICAM released in 2017, are updating of the cloud
microphysics scheme based on a comparison with Tropical Rainfall Measuring Mission (TRMM) satellite observations (Roh and Satoh, 2014; Roh et al., 2017), introduction of a wetland scheme in the land surface model (Nitta et al., 2017), implementation of the coupling between cloud microphysics and radiation that considers the non-sphericity of ice particles (Seiki et al., 2014), and introduction of a subgrid-scale orographic gravity wave drag scheme. In addition, some parameters related to the processes of surface albedo and sea ice have been revised. These model updates generally reduce the biases of
the simulated mean states, as discussed below. NICAM.16 has been further modified to support the external forcings of natural and anthropogenic aerosols and the solar cycle defined in the Coupled Model Intercomparison Project Phase 6 (CMIP6), High Resolution Model Intercomparison Project (HighResMIP) protocol (Haarsma et al., 2016). This special version of NICAM.16 for HighResMIP is labelled NICAM16-S, where "-S" represents the use of a single-moment cloud microphysics scheme. A double-moment cloud microphysics scheme is also available in NICAM.16 (Seiki and Nakajima,
2014; Seiki et al., 2014, 2015b; Satoh et al., 2018). However, the double-moment scheme was not used for the HighResMIP simulations and hence is not described in this paper. NICAM.16 is not coupled with an ocean model, and the DECK and CMIP historical simulations (Eyring et al., 2016) are not presented in this study. Note that a coupled ocean-atmosphere model, NICAM-COCO (Miyakawa et al. 2017), is being developed.

This section has provided a summary of NICAM16-S with a focus on the differences from NICAM.12. Section 2 of this paper presents the experimental design of the HighResMIP simulations and a series of sensitivity experiments by NICAM16-S. Section 3 explains the detailed model updates of NICAM16-S from NICAM.12 and their impacts on the simulated mean states. Section 4 briefly describes the resolution dependency of NICAM16-S. Section 5 reports on the computational aspects of the simulation. Section 6 provides a brief summary of the comparison results.

## 2. Experimental design

### 2.1 Spatial and temporal resolutions

Three sets of model configurations were prepared for the HighResMIP simulations, and initial and boundary conditions were made for each. NICAM16-S models with specific horizontal resolutions were formally labelled NICAM16-7S (56-km mesh),
NICAM16-8S (28-km mesh), and NICAM16-9S (14-km mesh) in CMIP6 as source IDs. The horizontal mesh size is evaluated as a square root of the mean area of each grid cell (Satoh et al., 2014). The number $n$ in NICAM16-$n$S is a grid division level (glevel), which denotes the number of subdivisions of the icosahedron to generate a mesh (Tomita et al., 2001). The physics schemes, including parameters, and the initial and boundary conditions are common among different horizontal resolutions except for those explicitly noted in Sections 2 and 3.

The number of vertical levels is 38, with a model top height of around 40 km, equivalent to the previous climate simulations (Kodama et al., 2015). The interval between each vertical layer increases from 160 m to 2 km as the altitude increases from the ground to 25 km (see the "K38" setting in Figure 1 of Ohno et al. 2019). Even at such a low vertical resolution, atmospheric phenomena of interests, including tropical cyclones, the MJO, and the diurnal precipitation cycle, may be

simulated practically and accurately, as we have confirmed in the previous study using a 14-km horizontal mesh (Kodama et al., 2015). As a caveat, a coarse vertical resolution in the upper atmosphere leads to an overestimation of the cirrus cloud amount (Seiki et al., 2015b; Ohno et al., 2019) and may cause a different response of high cloud amount to warmer climate (Ohno et al., 2019). Also, it has been suggested that the vertical resolution should be increased when the horizontal resolution is increased in terms of atmospheric gravity waves (Lindzen and Fox-Rabinovitz, 1989; Polichtchouk et al., 2019).

Such coarse vertical resolution in this study could change vertical propagation of gravity waves and zonal wind in the stratosphere (Watanabe et al., 2015; Skamarock et al., 2019).

The time step of the dynamics ($\Delta t$ in Satoh et al. 2008) was set to 4, 2, and 1 min in NICAM16-7S, NICAM16-8S, and NICAM16-9S, respectively. The time loop in the model is based on the dynamics, and physics schemes with a time step

smaller or greater than that of the dynamics are subcycled or skipped, as appropriate. Specifically, a time step of 30 s was used in the cloud microphysics scheme in NICAM16-7S, NICAM16-8S, and NICAM16-9S. A time step of 1 min was used in the turbulence (mainly for the planetary boundary layer) and land and ocean surface schemes in NICAM16-7S, NICAM16-8S, and NICAM16-9S. The radiation scheme, which requires considerable computational time, was executed every 40, 20, and 10 min in NICAM16-7S, NICAM16-8S, and NICAM16-9S, respectively. The gravity wave drag scheme

was called at the same time step as that of the dynamics.

## 2.2 HighResMIP simulations and sensitivity experiments

Table 1 shows the integration periods for the HighResMIP simulations. For the Tier 1 simulations, which start from 1 January 1950, the initial condition of the atmosphere was taken from the ERA-20C reanalysis (Poli et al., 2016). The

NICAM16-7S and NICAM-8S simulations strictly followed the HighResMIP protocol and continued until 31 December 2014. The HighResMIP Tier 3 simulations using NICAM16-7S and NICAM16-8S started from 1 January 2015 as a continuation of the Tier 1 simulations and ended on 31 December 2050. The high computational cost hindered us from running NICAM16-9S for 100 years, and thus a time-slice approach was adopted instead. Specifically, climate simulations were performed in the following timeframes: 1950–1960, 2000–2010, and 2040–2050. For the simulations starting from 1

January 1950 or 1 January 2000, the initial land conditions prescribed for NICAM16-7S, NICAM16-8S, and NICAM16-9S were taken from the monthly mean climatology of the simulation by NICAM with a mesh size of 220 km (glevel 5) under present-day conditions. It was performed for 10 years, and the last 5 years of data were used to obtain the monthly mean land climatology. This approach is the same as that used in previous climate simulations (Kodama et al., 2015). This could partly reduce the initial shock of the land surface model, even though it may cost more than several years for some land variables,

such as soil moisture, to fully settle down (not shown). The initial land condition for the future time slice run with NICAM16-9S was obtained by interpolating the output of the NICAM16-8S simulation.

In addition to the formal HighResMIP simulations, we performed a series of short-term sensitivity experiments to evaluate the impacts of the model changes on the simulated climatology, as listed in Table 2. Here, the model configuration of the

REFFIX run is equivalent to that used in the formal HighResMIP simulations. As noted in Section 3.7, we often prefer to use a slab ocean model with nudging toward the boundary sea surface temperature (SST) rather than the fixed SST condition requested by the HighResMIP protocol because of better performance in the simulated precipitation pattern (Kodama et al.,

2015), particularly with a horizontal mesh size of 14 km (Section 3.7). Therefore, both the fixed SST and slab ocean configurations (REFFIX and REFSLB runs, respectively) were tested in the sensitivity experiments, and we used the REFFIX run with 56-km mesh and the REFSLB run with 14-km mesh as the reference (REF) runs for the other sensitivity experiments. Impacts of the model updates described in Section 3 on the simulated climate states were individually tested by

switching off each update. Additionally, the DDT2M, DDT1M, RDT20M, and RDT10M runs were performed to check sensitivity of the simulated climate to the time step of the model. These sensitivity experiments were started from 1 June 2004 and integrated for 1 year. An exception was the NOLND and the REF runs, which were performed for 4 years to make the land surface state settle down. The initial date was chosen to ensure consistency with previous NICAM studies (e.g., Kodama et al., 2012; Seiki et al., 2015a; Noda et al., 2016). An integration period of 1 year, or even less, is sufficient to

evaluate the basic state of the atmosphere, such as cloud amount, precipitation, radiation, and temperature (e.g., Phillips et al., 2004; Noda et al., 2010; Kodama et al., 2012; Williams et al., 2013; Miyakawa et al., 2018; Miyakawa and Miura, 2019; Stevens et al., 2019; Hohenegger et al., 2020), and tropical variability, including diurnal cycle, tropical cyclones, and MJO (e.g., Sato et al., 2009; Kinter et al., 2013; Stevens et al., 2019; Matsugishi et al., 2020). The interannual variability of the HighResMIP simulation by NICAM16-7S (Table 1) over 101 years was diagnosed to distinguish the impacts of the model

changes from internal variability in a rough manner. The simulation data were re-gridded to the same grid of observations (Table 3) unless it is explicitly specified.

## 2.3 External forcings and boundary conditions

External forcings and boundary conditions of the simulations followed the HighResMIP protocol (Haarsma et al., 2016).
Historical and SSP5-8.5-scenario settings (O'Neill et al., 2016) were used in the Tier 1 and 3 simulations, respectively.

Daily quarter-degree SST and sea ice mass (ICE) prescribed for the model were obtained from HadISST 2.2.0.0 (Kennedy et al., 2017). The SST dataset was extended from 2016 to 2050 using the trend obtained from a CMIP5 model ensemble mean following the RCP8.5 scenario and historical variability from 1980 to 2015 (Kennedy et al., 2019). Because HadISST
provides historical sea ice concentration (SIC), ICE was diagnosed from SIC for NICAM (see Section 3.7). The future SIC was estimated from the future SST data and the observed relationship between SST and SIC (HighResMIP, 2020; Kennedy et al., 2019). Both the SST and ICE were fixed to the boundary conditions in the HighResMIP Tier 1 and 3 simulations.

Figure 1 and Figure 2 display the decadal mean SST and ICE, respectively prescribed for the model; Figure 3 exhibits their
global mean variability. Greater warming over the maritime continent, the Indian Ocean, and the edge of the polar regions are found in the 2000s compared with the 1950s, whereas cooling is noticed in the North Pacific and the North Atlantic. The SST in the 2040s has larger values almost everywhere compared with that in the 2000s, especially in the midlatitudes, the equatorial eastern Pacific, the tropical Atlantic Ocean, and the edge of the Arctic regions. Similar tendencies are observed for the distribution of the SST trend (not shown). The global mean SST is 17.9 °C in the 1950s, 18.2 °C in the 2000s, and
19.0 °C in the 2040s. ICE continues to decrease from the past to the future. The global mean ICE is nearly halved by the 2040s compared with that in the 1950s.

The global annual mean of greenhouse gas (GHG) concentrations (Meinshausen and Vogel, 2016; Meinshausen et al., 2017; Meinshausen and Nicholls, 2018) was prescribed in the model. Specifically, $CO_2$, $CH_4$, $N_2O$, chlorofluorocarbons (CFC-12,
CFC-11, CFC-113, CFC-114, and CFC-115), hydrochlorofluorocarbons (HCFC-22, HCFC-141b, and HCFC-142b), hydrofluorocarbons (HFC-134a, HFC-32, HFC-125, HFC-143a, and HFC-152a), $CCl_4$, $CF_4$, $SF_6$, and $C_2F_6$ were considered

as GHGs in the model. In addition, historical and future monthly concentrations of the three-dimensional ozone field (Hegglin et al., 2016, 2018) was prescribed in the model.

Mass and number concentrations of natural aerosols prescribed in the model were obtained from a low-resolution NICAM simulation with an online aerosol module based on the Spectral Radiation-Transport Model for Aerosol Species (SPRINTARS) (Takemura et al., 2000, 2002, 2005, 2009; Goto et al., 2008, 2011). The simulated climatology of aerosols in NICAM was previously validated (Goto et al., 2018; Suzuki et al., 2008). For HighResMIP, NICAM with a mesh size of 220 km was performed for 100 years using natural aerosol emissions with the anthropogenic aerosol module MACv2-SP (Fiedler et al., 2018; Stevens et al., 2017; see Section 3.4) under a perpetual 2012 condition, and the data for the last 90 years were averaged to obtain a monthly mean climatology of aerosol mass and number concentrations of cloud condensation nuclei (CCN) from a natural origin. A lower-bound limiter of 50 cm$^{-3}$ was applied to the CCN prescribed for the model to avoid numerical instability in the cloud microphysics scheme. Figure 4a shows the annual mean of the natural aerosol optical thickness simulated by NICAM16-7S with the natural aerosol mass and CCN (Figure 4b) prescribed in the model[1]. The climatology of natural aerosol mass and CCN is invariant year by year, whereas anthropogenic aerosols from MACv2-SP are time-dependent in the historical and future simulations (Fiedler et al., 2019; Stevens et al., 2017). Further, the extinction coefficient, the single scattering albedo, and the asymmetric factor were overwritten with the stratospheric aerosol dataset (Thomason et al., to be submitted) above the tropopause to introduce the effect of volcanic eruptions on the radiation field in a consistent way among different models participating in the HighResMIP.

Similar to the implementation of MIROC6 (Tatebe et al., 2019), historical monthly mean solar forcings (Matthes et al., 2017b, 2017a) were prescribed as total solar irradiance and solar irradiance spectra in the radiation scheme mstrnX (Sekiguchi and Nakajima, 2008). In terms of land surface processes, a monthly climatology (2004–2013) of leaf area index was obtained from the Moderate Resolution Imaging Spectroradiometer (MODIS) product (MCD15A2.005, Shabanov et al., 2005; Yang et al., 2006).

As briefly noted in Satoh et al. (2014), a spatial filter was applied to smooth the model topography to avoid numerical instability. Specifically, a hyper-diffusion was repeatedly applied to the GTOPO30 (doi:10.5066/F7DF6PQS), a global digital elevation model with a horizontal spacing of approximately 1 km, to meet a specific criterion for the maximum elevation gradient. The maximum elevation gradient was set to 0.01, 0.01414, and 0.02 m m$^{-1}$ for NICAM16-7S, NICAM16-8S, and NICAM16-9S, respectively, and the resulting topography is called "A-topography." Note that, in previous NICAM studies using a 14-km horizontal mesh (e.g., Kodama et al., 2015), "B-topography," in which A-topography with a 28-km mesh was interpolated to 14-km mesh grid points, was often used for the sake of stable integration. For HighResMIP, A-topography was used to better represent steeper mountains and their effects on atmospheric phenomena.

---

[1] An error was found in the natural aerosol forcing prescribed in the model, as recently reported in ES-DOC Errata website (https://errata.es-doc.org/static/view.html?uid=ada34e91-4a94-d668-a491-fe16556aaf46). Its influence on the results presented in this paper seems to be negligible, according to an additional 56 km mesh experiment with the corrected natural aerosol forcing (not shown).

## 3. Model description and impact of model updates on the simulated fields

### 3.1 Overview

The dynamical core and numerical filters in NICAM16-S are the same as those in NICAM.12. NICAM adopts a fully compressible non-hydrostatic system as the governing equations of the dynamics (Tomita and Satoh, 2004; Satoh et al., 2008). The horizontal discretization is an icosahedral grid system modified with spring dynamics for homogeneity on the sphere (Tomita et al., 2002). Divergence damping and second-order Laplacian horizontal diffusion are used to stabilize the integration (Satoh et al., 2008). Additionally, first-order Laplacian horizontal diffusion is applied above an altitude of 20 km to avoid spurious wave reflection at the model top.

Table 4 gives a summary of the physics schemes used in NICAM16-S and NICAM.12. A single-moment bulk cloud microphysics scheme, NICAM Single-moment Water 6 (NSW6), that solves mass concentrations for six water categories – vapor, cloud water, cloud ice, rain, snow, and graupel (Tomita, 2008; Roh and Satoh, 2014; Roh et al., 2017) – is used instead of a combination of convection and large-scale condensation schemes. While most climate models use convection and large-scale condensation schemes even for a mesh size around 14 km, we used the cloud microphysics scheme to represent interactions between clouds and circulation in an explicit way. This not only lowers the cost of model development but also reduces the uncertainty in the results arising from highly arbitrary tuning. This approach has also been tested by other researchers besides the NICAM group (Maher et al., 2018; Hohenegger et al., 2020). Global mean precipitation is constrained by radiative cooling in large-scale clear-sky regions, which can be captured by a model with relatively coarse resolution without the convection and large-scale condensation schemes. The simulated climatology of the precipitation pattern, even with the lowest resolution setting (NICAM16-7S), is comparable with the observed patterns, as shown below, although our choice leads to patchy precipitation behaviour and dry/wet bias in the middle/lower troposphere in the simulation (Miyakawa et al., 2018). Similar precipitation behaviour was also reported in a climate model study with a mesh size of around $O(10^2$ km) without a convection scheme (Maher et al., 2018). In terms of clouds, Seiki et al. (2015b) conducted NICAM simulations with 28- and 14-km meshes to study the impact of the vertical resolution on the simulated cirrus clouds. They found a similar vertical resolution dependency between the 14- and 28-km meshes. Ohno et al. (2019) used a 28-km mesh NICAM and found a high cloud response to SST increases that is comparable with results using 7- and 14-km meshes (Iga et al., 2007). In terms of MJO, Takasuka et al. (2018) performed an aqua planet experiment with a 56-km mesh NICAM to investigate MJO-like disturbances. Yoshizaki et al. (2012) and Takasuka et al. (2015) even performed NICAM with a mesh size larger than 100 km without the convection and large-scale condensation schemes and found MJO-like disturbances in the simulation.

A modified version of the Mellor–Yamada level 2 scheme (Nakanishi and Niino, 2006; Noda et al., 2010) is used to simulate planetary boundary layer. The radiation scheme, mstrnX (Sekiguchi and Nakajima, 2008), is a broadband model with 29 radiation bands as used here. The land surface model, Minimal Advanced Treatments of Surface Interaction and RunOff (MATSIRO) (Takata et al., 2003) solves land states such as soil temperature, soil moisture, and land surface fluxes. Ocean surface fluxes are calculated following Louis (1979) with a modified roughness length for strong surface wind conditions (Fairall et al., 2003; Moon et al., 2007). The conventional orographic gravity wave drag scheme (McFarlane, 1987) is used to introduce the effect of vertically propagating subgrid-scale orographic gravity waves on the momentum tendency of the atmosphere. Though we did not fine-tune the model due to heavy computational cost, we crudely turned parameters of sea ice thickness with NICAM16-7S (Section 3.7) and the gravity wave drag scheme with NICAM16-9S (Section 3.8). We did not return the model at each resolution under the principles of the HighResMIP.

As we discuss in Section 2.2 and Table 2, a series of short-term sensitivity experiments were performed to monitor impacts of several model updates on the simulated climatology. Figure 5 (right part of each panel) summarizes impacts of the model changes on the global mean climate. All the significant impacts of the model changes shown here could be qualitatively reproduced even when the analysis period was limited to the last 6 months (not shown). The REFFIX and REFSLB runs with each horizontal mesh and the observations are shown on the left in each panel in Figure 5. We will discuss these impacts along with the details of the model updates later in this section.

## 3.2 Cloud microphysics

Recently, Roh and Satoh (2014) and Roh et al. (2017) significantly revised the NSW6 scheme based on comparisons with TRMM observations. We used the revised version of the NSW6 scheme in NICAM16-S and found improvements in the simulated climatology, as shown below.

The NSW6 scheme originated with Lin et al. (1983) and Rutledge and Hobbs (1984). Tomita (2008) modified their cloud microphysics scheme to ensure consistency with the thermodynamics used in NICAM. Tomita (2008) also simplified it to reduce the calculation cost for high-resolution global simulations. NSW6 was evaluated by comparing the simulated optical properties with satellite observations (Satoh et al., 2010; Kodama et al., 2012; Hashino et al., 2013, 2016; Roh and Satoh, 2014, 2018; Roh et al., 2017) using satellite simulators, specifically, the CFMIP Observational Simulator Package (Haynes et al., 2007; Chepfer et al., 2008; Bodas-Salcedo et al., 2008, 2011) and Joint Simulator (Matsui et al., 2009; Masunaga et al., 2010; Hashino et al., 2013). It was revised in each stage of the version management of NICAM (Kodama et al., 2012; Roh and Satoh, 2014; Roh et al., 2017). The revision of the NSW6 scheme by Roh and Satoh (2014) and Roh et al. (2017) represents a significant change between NICAM.12 and NICAM16-S.

Table 5 summarizes key changes in the NSW6 scheme by Roh and Satoh (2014) and Roh et al. (2017). In short, the revision aimed to enhance organizations of tropical convective cloud systems by assuming lighter precipitation of graupel and snow and by moderating the development of cloud ice. Finally, they successfully reproduced the vertical structures of shallow, congestus, and deep convective clouds over the tropics, compared to TRMM and CloudSat satellite observations. They also used microwave satellite observations to evaluate the simulated results (Roh and Satoh, 2018). Therefore, improvements in tropical cloud systems with the revised scheme are robust in terms of optical signals (see the original papers for more details). Note that the separation of convective and stratiform systems in Roh and Satoh (2014) was omitted in this study because of its small impact (not shown) despite the high computational cost.

The sensitivity experiments with and without the update of the cloud microphysics scheme (the REF and NOCLD runs, respectively) were compared. The most noticeable impact was an increase in ice water content (IWC) by more than twofold (Figure 5, e), and this mostly accounts for the snow category in the cloud microphysics scheme. Figure 6 shows the meridional-height cross section of the observed and simulated zonal mean IWC and its breakdown into the categories of cloud ice, snow, and graupel in the NOCLD and REF runs. The simulated IWC is largely underestimated using the model without the updated cloud microphysics scheme (the NOCLD run), as also shown in Seiki et al. (2015a), and it becomes comparable with CloudSat observations with the update (the REF run). A noticeable increase in the snow category was seen in the tropical upper troposphere and midlatitude storm-track region. Cloud ice is also increased in the upper troposphere. Graupel increased in the tropical middle troposphere but decreased in the storm-track region. As a result, global mean column-integrated cloud ice and snow increased by 24% and 399%, respectively, and that of graupel decreased by 8.4%. The net increase in IWC is consistent with the decelerated development of IWC by the modified mass and diameter relationship

of snow, which reduces the snow density (Table 5, d), by the diminished efficiency of accretion of cloud ice by snow (Table 5, g), and by ignoring the accretion of snow and cloud ice by graupel (Table 5, f). Differences in cloud processes (convection versus synoptic system) may cause different signs of the impact of the model update on graupel .

Despite the drastic increase in mass concentrations of snow and cloud ice, the amount of high clouds, particularly, optically thin clouds, is reduced by the update (Figure 5, g). Consistently, global mean outgoing longwave radiation (OLR) is increased by about 4 W m$^{-2}$ (Figure 5, c), opposite to that found in Roh et al. (2017). In addition, the decrease in top-of-atmosphere (TOA) brightness temperature by the update is very small (Figure 7) compared with that in Roh et al. (2017). These differences between Roh et al. (2017) and this study can be mostly explained by the new coupling procedure between

cloud microphysics and radiative transfer, as described in Section 3.3, and be partially explained by the different treatment of the terminal velocity of cloud ice in the reference runs between Roh et al. (2017) and this study. Eventually, the cloud ice terminal velocity was set to zero in both Roh et al. (2017) and this study. Unlike this study, Roh et al. (2017) performed their reference run with non-zero cloud ice terminal velocity, as diagnosed by Heymsfield and Donner (1990), and their comparison before and after the scheme update includes the effects of reduced cloud ice terminal velocity. The reduction of

the cloud ice descent speed, as in Roh et al. (2017), could increase and elevate high clouds and decrease the OLR (Kodama et al., 2012). The low cloud amount increased (Figure 5i) as a result of the compensation between an increase in medium and thick clouds and a decrease in thin clouds. These results indicate that the clouds grow thicker on average by updating the cloud microphysics scheme in this study.

**3.3 Coupling between cloud microphysics and radiative transfer**

In NICAM16-S, the cloud microphysics scheme is fully coupled with the radiation scheme, mstrnX (Sekiguchi and Nakajima, 2008). The effective radius of hydrometeors is calculated with the same assumption of the particle size distribution function as the cloud microphysics scheme, including indirect effects, and then passed to the mstrnX scheme. In contrast, fixed effective radii of 8 μm for liquid hydrometeors and 40 μm for ice hydrometeors were assumed in NICAM.12.

The use of consistent assumptions of coupling between cloud microphysics and radiative transfer can reduce the model bias in the radiation budget (Seiki et al., 2015a) and has non-negligible impacts on climate projection (Chen et al., 2016). In addition, the coupling provides model developers with a better understanding of the origins of model biases (Hashino et al., 2016).

The mstrnX scheme requires a database of single scattering properties of hydrometeors (RADPARA), including parameters such as the volume extinction coefficient, absorption coefficient, asymmetry factor, and second moment of phase function (Nakajima et al., 2000). In NICAM16-S we used the RADPARA database revised by Seiki et al. (2014). The RADPARA database of liquid hydrometeors was pre-calculated according to the Mie theory. The non-spherical RADPARA database developed by Fu (1996) and Fu et al. (1998) was applied to solid hydrometeors. The RADPARA database was then

compiled as a lookup table of the effective radii from 1 μm to 1 mm to cover the size range of most of the hydrometeors in global simulations (Seiki et al., 2014). The effects of precipitating hydrometeors on the radiation budget are detectable, specifically over the intertropical convergence zone (ITCZ) and storm-track region (e.g., Waliser et al., 2011; Li et al., 2014, 2016; Chen et al., 2018; Michibata et al., 2019). The revised RADPARA database was evaluated in depth by comparing it with balloon-borne sonde observations in a midlatitude cirrus case (Seiki et al., 2014), and its effectiveness for global

simulations was evaluated in several studies (Seiki et al., 2015a, 2015b; Satoh et al., 2018).

Because of non-sphericity, the effective radius of ice particles has a controversial definition, whereas the effective radius is well defined in the case of spherical particles. According to Fu (1996), the effective radius, $r_e$, of solid hydrometeors is defined as follows:

$$r_{e,j} = \frac{3}{4\rho_{ice}} \frac{\rho q_j}{\int_0^\infty A_j(D_j)N_j(D_j)dD_j} \quad (j = i, s, g), \tag{1}$$

where $j = i, s, g$ are cloud ice, snow, and graupel, respectively; $\rho$ is the air density; $\rho_{ice} = 916.7 \text{ kg m}^{-3}$; $q_j$ is specific

content; $A_j$ is the projected area of a particle to flow; and $D_j$ is diameter. The integral in equation (1) is analytically calculated using the assumed particle size distribution functions $N_j(D_j)$ and a sponge-like spherical shape for cloud ice and graupel. In contrast, snow has two-dimensional fractal shapes; hence, the numerator–denominator ratio becomes almost constant. Thus, the effective radius of snow is assumed to be constant ($r_{e,s} = 125$ μm with $A_s = 0.45D_s^{2.0}$), and it is derived by approximating the $A$–$D$ relationship of aggregates compiled by Mitchell (1996).

The impacts of the coupling procedure and the non-spherical scattering were examined by comparing the REF run with the NONSI run using the fixed effective radii and the spherical RADPARA database. The impacts of non-spherical scattering alone could be seen from the comparison between the REF and NONS runs using the spherical RADPARA database. Figure 8 shows the zonal mean values of the OLR and the outgoing shortwave radiation (OSR) at the TOA from the sensitivity

experiments. Given the substantial increase in cloud ice and snow from the revised NSW6 scheme (cf. Section 3.2), ice optical thickness increases proportionately to the increases in the ice water path with fixed effective radii. As a result, both the longwave and shortwave radiation budgets are strongly biased in the NONSI run. A major portion of the biases in OLR is drastically offset in the NONS run by assuming larger effective radii in the coupling procedure. Thus, the coupling procedure automatically prevents artificial biases originating from inconsistent parameter settings between the cloud microphysics and

radiative transfer with the model update. The use of the non-spherical RADPARA database slightly increases OSR over the tropics to the midlatitude because the assumed asymmetry factor for non-spherical particles is smaller than that for spherical particles (cf. Seiki et al., 2014).

Finally, NICAM16-S still shows strong negative biases in OLR over the tropical to subtropical regions and OSR over the

subtropical high-pressure belt at the TOA compared with the Clouds and Earth's Radiant Energy Systems (CERES) product (Figure 8, black versus red curves), and these biases are qualitatively similar to those simulated in NICAM.12 (Kodama et al., 2015). The former OLR bias can be solved by increasing vertical resolution to 400 m near the tropopause with 74 vertical layers (Seiki et al., 2015b). The latter OSR bias mainly stems from the underestimation of low-level clouds, and the current updates in the cloud microphysics scheme do not improve the results for warm clouds. The coupling procedure strengthens

the negative biases in OSR at TOA in the midlatitudes (Figure 8, green versus blue curves) because, in NICAM.12, high clouds associated with extratropical cyclones are artificially brightened and, therefore, conceal the biases due to low-level clouds (cf. Fig. 4 in Kodama et al., 2012). Unlike NICAM.12, a strong negative bias of OSR at the TOA is also prominent over the Arctic region, and this seems to relate to the update (reduction) of the surface albedo introduced in Section 3.6.

**3.4 Aerosols in the cloud microphysics and radiation schemes**

In NICAM.12, the direct radiative effect of aerosols is not considered in the radiation scheme, and the number concentration of CCN is set to a constant value of 50 $\text{cm}^{-3}$, a typical value over the ocean, in the cloud microphysics scheme. In NICAM16-S, both the direct and indirect effects of aerosols are considered by prescribing a distribution of aerosol mass concentration in the radiation scheme and CCN in the cloud microphysics scheme. The dataset of natural aerosols in the

troposphere and stratosphere is described in Section 2.3. For anthropogenic aerosols in the troposphere, a simple plume

model, MACv2-SP (Stevens et al., 2017; Fiedler et al., 2019), is used to diagnose the vertical profile of the aerosol optical depth, single scattering albedo, and asymmetry factor and factor of CCN increase arising from anthropogenic aerosols. This means that the magnitude of the anthropogenic increase in CCN depends on CCN from natural origins.

The sensitivity experiments with and without the update of the aerosol treatment (the REF and NOAER runs) are compared in Figure 9. The aerosol update reduces the net downward shortwave radiation at the surface over most of the continents, particularly over Africa and South Asia (Figure 9, b and d), leading to a reduction of the excess insolation there (not shown). The reduced insolation at the surface is partly cancelled out by a reduction in the upward longwave radiation over Africa (Figure 9, a and c) in association with a decrease in surface air temperature (not shown). Meanwhile, an enhancement of the
surface net downward shortwave radiation is dominant over the ocean, particularly in the NICAM16-9S (14-km mesh) run (Figure 9, b and d), in association with a thinning of cloud optical depth and a decrease in cloud amount (not shown). This links to a decrease in the liquid water path with the aerosol update (Figure 5f), which was also found in an online aerosol experiment with the 14-km mesh NICAM (Sato et al., 2018). As a result of these compensations, the global mean net surface radiation change arising from aerosol forcing is around $-2.0$ W m$^{-2}$ in NICAM16-7S and $+0.4$ W m$^{-2}$ in NICAM16-9S in
this study. Such a reverse from a decrease to an increase between the resolutions might be related to the resolution dependency of the low and middle cloud amount in the REF run (left panels of Figure 5, h and i), and a detailed analysis is needed to properly understand the mechanism.

**3.5 Land surface model**

The land surface model MATSIRO (Takata et al., 2003) is used in NICAM. Recently, a wetland scheme was implemented in MATSIRO of NICAM16-S to represent the storage of snowmelt while considering the subgrid-scale terrain complexity (Nitta et al., 2017). The wetland scheme reduces the summertime warm and dry bias over much of Western Eurasia and North America through delayed snowmelt runoff in MIROC5 (Nitta et al., 2017). In addition, the effect of decreased surface albedo associated with the accumulation of water on land ice was implemented in NICAM16-S.

Figure 10 shows the impact of the land surface model update on soil moisture during boreal summers. The soil moisture is increased over most of the Eurasian and North American continents, as expected from Nitta et al. (2017), particularly in Siberia and around the Great Lakes. Though it is expected from Nitta et al. (2017) that increased soil moisture leads to an increase in precipitation and a decrease in surface air temperature, the actual impacts on precipitation and surface air
temperature are still unclear (not shown). It is difficult to show robust reduction of the biases at this stage, and longer integration is needed to assess these impacts appropriately.

**3.6 Surface albedo**

Surface albedo values were updated based on observations. In NICAM.12, they were tuned to reduce the TOA radiation
imbalance, and this caused a higher bias of surface albedo over the Arctic compared with that seen in satellite observations (Hashino et al., 2016). Table 6 shows the surface albedo values used in NICAM16-S and NICAM.12. The albedo values of the sea ice for the visible and near-infrared wavelengths and of fresh snow over land for the visible wavelengths were set to be smaller in NICAM16-S than those in NICAM.12. In addition to the updates in Table 6, an artificial elevation of the ocean surface albedo for the direct visible wavelength by a factor of 1.35 for the radiation scheme in NICAM.12 was discarded in
NICAM16-S.

The sensitivity experiments with and without the albedo update (the REF and NOALB runs, respectively) show that the use of the new surface albedo values tends to reduce the surface air temperature bias over land ice compared with the previous bias (Figure 11, a, versus Figure 11, c, and green versus red curves in Figure 11, d). Specifically, the cold bias in Greenland, the Himalayas, and the Antarctic is reduced. This is consistent with the reduced surface albedo for the visible wavelengths and the resulting decreased net upward shortwave radiation at the surface (Figure 5, k). Global mean net downward longwave radiation at the surface is increased (Figure 5, j). The increase is attributed primarily to a decrease in upward longwave radiation over the ocean (not shown), consistent with the increased surface albedo for the infrared wavelengths over the ocean. In terms of the TOA radiation budget, OSR worsens by a few watts per square meter (Figure 5, d), which arises from the polar regions (Figure 12, b; green versus red curves).

## 3.7 Treatment of oceans

A mixed-layer slab ocean model similar to that in McFarlane et al. (1992) was implemented in NICAM. The model predicts SST, ICE, snow over sea ice, and snow temperature by solving the heat balance between the ocean, sea ice, snow, and atmosphere. The depth of the slab ocean model is set to 15 m globally, considering the better performance of the simulated precipitation pattern (Kodama et al., 2015) and MJO (Grabowski, 2006). A simple nudging technique is used to force the predicted SST and ICE toward their reference states with a relaxation time of $\tau_{SST}$ and $\tau_{ICE}$, respectively. Specifically, $\tau_{SST} = 7$ days and $\tau_{ICE} = 0$ (i.e., ICE was fixed to the boundary condition) were used in the slab ocean experiments of this study and in the previous climate simulation with NICAM.12 (Kodama et al., 2015). Both $\tau_{SST}$ and $\tau_{ICE}$ were set to zero in the fixed SST/ICE experiments, including the HighResMIP simulations.

In the slab ocean model implemented in NICAM.12 and NICAM16-S, SIC is diagnosed from ICE as

$$SIC = \begin{cases} \sqrt{\dfrac{ICE}{SICCRT}} & \text{for ICE} < \text{SICCRT,} \\ 1 & \text{for ICE} \geq \text{SICCRT} \end{cases} \tag{2}$$

where SICCRT is a parameter in kg m$^{-2}$.

In many cases, including the HighResMIP protocol, only the SST and SIC data are provided to run the model, and, therefore, the ICE data prescribed for the model should be diagnosed from SIC data. In NICAM.12 and NICAM16-S, ICE is diagnosed simply as

$$ICE = SICCRT \times SIC^2. \tag{3}$$

In the previous study using NICAM.12, we often set the value of SICCRT to 300 kg m$^{-2}$, considering Eq. (2). However, this situation leads to an underestimation of ICE over most of the sea ice areas and causes a warm bias over the Arctic (Kodama et al., 2015; Figure 11, b; blue curve in Figure 11, d). Based on an ocean model result (H. Tatebe, personal communication), we performed a series of preliminary annual-scale experiments using NICAM16-7S, with SICCRT values of 1,600 and 3,200, to improve the surface air temperature over the Arctic. As a result of this crude tuning, SICCRT is set to 1,600 kg m$^{-2}$ in NICAM16-S. This led to a significant reduction in the warm bias (Figure 11, b, versus Figure 11, c; blue versus red curves in Figure 11, d) and an excess of OLR at TOA (blue versus red curves in Figure 12, a) over the Arctic.

In the HighResMIP protocol, SST and SIC in the model are fixed to the time-varying boundary conditions. Overall, the global mean impact of the slab ocean model is not very large (circles versus rectangles in Figure 5). Compared with the fixed SST runs, global mean precipitation and OLR showed slight increases in the slab ocean runs, which are associated with a

slightly warmer surface air temperature. In terms of the local climate, however, the fixed SST simulation is known to cause severe bias in the horizontal distribution of clouds and precipitation systems in the tropics (Kodama et al., 2015; Figure 13), and thus the use of the slab ocean model with a 7-day relaxation time is often preferred. The introduction of the slab ocean model considerably affected the horizontal distribution of clouds and precipitation systems (Figure 13). A double ITCZ bias was more prominent in the precipitation, as well as the high cloud fraction and OLR fields (not shown) in the fixed SST runs compared with the slab ocean runs, particularly in the high-resolution run. Our investigation showed that NICAM16-9S with the slab ocean model best simulated the ITCZ peak precipitation and the precipitation pattern. Although the importance of the short-term SST variation driven by the atmosphere for the precipitation pattern is apparent from Figure 13, the introduction of the slab ocean model alone does not resolve its bias. Further analysis is necessary to understand the physical mechanisms of the bias, which may be due to factors such as the convective timescale.

### 3.8 Orographic gravity wave drag

No gravity wave drag scheme is used in NICAM.12. In NICAM16-S, the conventional orographic gravity wave drag scheme (McFarlane, 1987) is used to better simulate the location and strength of the subtropical jet. The wave generation parameter $\alpha$, which is proportional to the product of wave generation efficiency and representative horizontal wavenumber (Eq. 3.1b in McFarlane 1987), was tuned first for NICAM16-9S to improve zonal mean zonal wind and then roughly halved as the horizontal mesh size was doubled. Specifically, $\alpha$ was set to $3.38 \times 10^{-5}$ for NICAM16-7S, $7.12 \times 10^{-5}$ for NICAM16-8S, and $1.46 \times 10^{-4}$ for NICAM16-9S. Figure 14 and Figure 15 show the zonal mean zonal wind in boreal summer and winter, as simulated with and without the gravity wave drag scheme (the REF and NOGWD runs, respectively). The zonal mean zonal wind simulated without the gravity wave drag scheme was biased poleward in both the NICAM16-9S and NICAM16-7S runs. The gravity wave drag scheme decelerated the zonal mean zonal wind at the poleward flank of the subtropical jet, especially during northern hemisphere winter, reducing the locational bias of the jet. The impact of the gravity wave drag scheme is larger in NICAM16-7S than in NICAM16-9S. The pattern of the zonal wind response to the gravity wave drag scheme is similar to that of previous studies (e.g., McFarlane, 1987; Iwasaki et al., 1989).

Although it is believed that even a mesh size of 14 km is insufficient to explicitly simulate the effects of the orographic gravity wave drag on the mean field (Nappo, 2012), introducing such a gravity wave drag scheme will not necessarily lead to an improvement of the simulated climate in the global non-hydrostatic model. The gravity wave drag scheme introduces the parameter $\alpha$, which is uncertain and tuned to best simulate the climatology of the zonal wind for each resolution in general, although we did not tune $\alpha$ for each resolution. There is no solid guideline for determining $\alpha$, including the dependency of the wave generation efficiency and representative horizontal wavenumber on the horizontal and vertical resolutions. Therefore, the use of a gravity wave drag scheme may hinder the pure resolution dependency of the mean field and suppress the advantages of the high-resolution model for simulating large-scale circulation in a seamless manner. Nevertheless, we decided to use the orographic gravity wave drag scheme for HighResMIP to reduce the locational bias of the subtropical jet to improve results for the tropical cyclone track region. It is important to recognize the merits and demerits involved in the use of a gravity wave drag scheme and reconsider its use depending on the main purpose of the simulation.

### 4. Horizontal and temporal resolution dependency

Understanding the dependency of horizontal resolution is a central interest of the HighResMIP. Figure 16 shows the global mean climate in NICAM16-7S (56-km mesh; blue circle), NICAM16-8S (28-km mesh; green circle), and NICAM16-9S (14-

km mesh; red circle), along with its sensitivity to the time step of the models' dynamics (including the gravity wave drag scheme) and radiation scheme. Note that the dependency of the time step of the radiation scheme on the global mean climate is negligible.

Global mean precipitation and TOA OLR decreased as the horizontal resolution increased (Figure 16, b and c), consistent with a previous NICAM study using 3.5- to 14-km meshes (Miyakawa and Miura, 2019). The results did not strongly depend on the time step of the model. As seen in Figure 13, the precipitation pattern in the tropics depends strongly on the resolution: more-dominant double-ITCZ patterns and less-intense local precipitation are simulated as the horizontal resolution is increased. The intense precipitation occurs less frequently in the higher-resolution runs (Figure 17), consistent

with Noda et al. (2012) using older NICAM with 14- and 7-km meshes. The intense precipitation occurs more frequently in the model compared with the Global Precipitation Climatology Project (GPCP) product (Noda et al., 2012; Figure 17), and it is consistent with the results of Maher et al. (2018), who compared precipitation in global climate models without a convection scheme with precipitation in the GPCP product. Na et al. (2020) showed that the frequency of intense precipitation in the GPCP product is lower than that in the TRMM product, and the 14-km mesh NICAM without a

convection scheme could realistically reproduce the intense precipitation observed by the TRMM satellite (Figure 17).

The low cloud amount was substantially underestimated, especially in the NICAM16-7S and NICAM16-8S runs (Figure 16, i), leading to an underestimation of the OSR at the TOA (Figure 16, d). Consistently, the net downward shortwave radiation at the surface decreased (Figure 16, k) and the surface air temperature slightly decreased (Figure 16, a) as the horizontal

resolution increased. This dependency of the low cloud amount and its related variables, in terms of global mean, on horizontal resolution could be reproduced by changing the time step of the dynamics in the model (the DDT2M and DDT1M runs). Specifically, the time loop in the model is based on the dynamics (Section 2.1), and the number of sub-cycles of the microphysics, turbulence, and surface processes were reduced in the DDT2M and DDT1M runs to keep their time steps unchanged (Table 2). This leads to an increase in frequency of the coupling between the dynamics and these physics

schemes. Additional monthly-scale sensitivity experiments suggest that the differences between the REFFIX runs and the DDT2M and DDT1M runs are mostly attributed to the altered frequency of physics-dynamics coupling, not the stability issue of the dynamical core (not shown). The low cloud amount was rather greater in the 56-km mesh run than that in the 14-km mesh run under the fixed time step of the dynamics (red circle in the REFFIX run versus blue circle in the DDT1M run in Figure 16, i). Also, the simulated OSR at the TOA is greater and closer to that of the CERES product in the 56-km mesh

run compared with the 14-km mesh run with the same temporal resolution, though the better performance in the simulated global mean OSR at the TOA in the 56-km mesh run is a result of a strong compensation between a negative bias off the subtropical west coasts of continents and the southern hemisphere storm-track region and a positive bias in the rest of the lower latitudes (not shown). Such a result of dependency of OSR at the TOA on horizontal resolution with a fixed temporal resolution is similar to that found by Goto et al. (2020), who executed 14-km and 56-km mesh online-aerosol NICAM with

the same time steps of 60 s for the dynamics, turbulence, and surface schemes and 10 s for the cloud microphysics scheme. Beside the resolution, HighResMIP requires that the model parameters and configuration be the same between the standard and high-resolution runs. However, adapting the time step to the horizontal resolution, as we have done, is a common approach, mainly for numerical stability reasons. Therefore, we highlighted here the impact of both the horizontal and temporal resolutions on the global mean climate in the NICAM simulations..

## 5. Computational aspects

### 5.1 Simulations

Table 7 describes the computational setting and the simulation year per wall-clock day (SYPD) of the NICAM16-S simulations run on the Earth Simulator 3. The Earth Simulator 3 is an NEC SX-ACE system with 5,120 nodes in total for

computation, and each computation node has 4 cores. We often used 10, 40, and 160 computation nodes to run NICAM16-7S, NICAM16-8S, and NICAM-9S, respectively, to obtain a balance between computational efficiency and wall clock time. An exception was NICAM16-8S for the HighResMIP simulation, for which 160 computation nodes were used to perform a 101-year simulation within a realistic clock time. A file-staging option was used in the NICAM16-8S and NICAM16-9S simulations, whereas the file was directly read and written from a global file system in the NICAM16-7S simulation to

reduce queuing time. The actual SYPD was a few times smaller than that shown in Table 7 for NICAM16-8S and NICAM16-9S.

Figure 18 breaks down the total elapsed time for each component of the model. Unlike a previous evaluation using the K computer (Yashiro et al., 2016), the measurement includes the initial setup and input/output processes. Physics contributes a

major part to the total elapsed time. Among the several physics components, the radiation scheme primarily contributes to the total elapsed time, followed by the cloud microphysics scheme, consistent with Yashiro et al. (2016) on the K computer. As the resolution increased, the percentage of time consumed by dynamics increased and the time consumed by the cloud microphysics and turbulence processes decreased because of their invariant time step among the models with different resolutions. An increase in the percentage of the land surface scheme in the simulations with higher resolution and greater

number of computational nodes seems to be caused by the node imbalance associated with land–ocean distribution. Because the dynamics involves communication at every time step, some parts of the elapsed time counted as the dynamics can actually be a waiting time due to the node imbalance occurring in other processes.

### 5.2 Post-processing

The data are output on the model's native icosahedral grid on the height above sea level (ASL) or on the standard pressure. The vertical interpolation from the terrain-following height to the ASL height or standard pressure is performed online using second-order Lagrange interpolation during the simulation.

Post-processing is performed in the following order: *ico2ll*, *roughen*, and *z2pre*.

(1) *ico2ll*

All the native icosahedral grid data are converted to high-resolution latitude–longitude grid data by area–weight averaging. The interval of latitude and longitude is determined so that the longitudinal interval is close to the average interval of the icosahedral grid (Satoh et al., 2014) on the equator. Specifically, the interval of longitude and latitude is 0.56° for NICAM16-7S, 0.28° for NICAM16-8S, and 0.14° for NICAM16-9S.

(2) *roughen*

All the high-resolution latitude–longitude grid data are coarsened to low-resolution latitude–longitude grid data by area–weight averaging. It is often necessary to reduce the amount of data by such coarsening, although HighResMIP does not ask us to coarsen the data. We prepared 1.0°, 1.25°, and 2.5° data for analysis.

(3) *z2pre*

Several three-dimensional variables are converted from the ASL height to the standard pressure at this point by linear interpolation, if it is necessary.

After post-processing steps (1)–(3), monthly mean data are created.

The pressure velocity is obtained from the vertical velocity and temperature after *ico2ll* under the assumption of hydrostatic balance. The geopotential height is calculated from the linear interpolation of vertical levels using logarithms of pressure, with an assumption of constant gravity acceleration regardless of height, which is a treatment consistent with the model configuration.

## 6. Summary

This paper describes the experimental design, the latest stable version of NICAM prepared for the HighResMIP (NICAM16-S), and impacts of NICAM updates on the simulated climatology using NICAM16-S at different resolutions. The major updates and their impacts on simulation results are summarized as follows:

- update of the cloud microphysics scheme: Snow and cloud ice was increased, leading to decreased high cloud amount and increased OLR at the TOA.
- implementation of the coupling between cloud microphysics and radiation schemes: The negative OLR bias was reduced in association with a larger cloud ice effective radius.
- update of treatment of natural and anthropogenic aerosols: The local surface radiation budget was improved, especially over Africa and South Asia.
- update of land surface model: Overall, the soil moisture over most of the Eurasian and the North American continents was increased.
- update of the surface albedo values: The cold bias in the Greenland, the Himalayas, and the Antarctic was decreased.
- modification of the ICE diagnostics: The warm bias over the Arctic region was decreased.
- introduction of the orographic gravity wave drag scheme: The location and strength of the zonal mean jet were improved.

Comprehensive evaluations and future projections using full HighResMIP data by NICAM16-S will be presented in a forthcoming paper.

## Code and data availability

The exact model source code, input data and scripts to generate them, and scripts for the simulations and post-processing used to produce the results presented in this paper are archived on Zenodo (doi:10.5281/zenodo.3727329). The model source code is shared with the NICAM community and available for those who are interested as long as the user follows the terms and conditions described at http://www.nicam.jp/hiki/?Research+Collaborations. Most of the input data are freely accessible from input4MIPs (https://esgf-node.llnl.gov/projects/input4mips/) for ocean boundary conditions, GHG concentrations, and ozone and solar forcing; from the ECMWF website (https://apps.ecmwf.int/datasets/data/era20c-daily/) for ERA-20C reanalysis; from supplemental materials of MACv2-SP description papers (Fiedler et al., 2019; Stevens et al., 2017) for anthropogenic aerosol data; and from the U.S. Geological Survey website (https://doi.org/10.5066/F7DF6PQS) for GTOPO30 data. The other input data, obtained from ftp://iacftp.ethz.ch/pub_read/luo/CMIP6/MIROC3.2_29/ for volcanic aerosols and from https://lpdaac.usgs.gov/ for the leaf area index, are available on request from the corresponding author. The high-resolution data of the product run requested by HighResMIP are or will be available through the Earth System Grid

Federation (ESGF). All the other product run data such as low resolution, monthly mean, and special variables and sensitivity experiment data are available on request from the corresponding author.

**Author contributions**

CK and ATN managed the overall HighResMIP activity in the NICAM group and prepared the initial and boundary conditions, and MS managed development and scientific activity in the NICAM group. CK added interfaces of the initial and boundary conditions to NICAM. TO added a function to output variables requested by HighResMIP and converted the output data using CMOR3. CK, TO, TS, HY, MN, and YY contributed to the development of NICAM16-S, including debugging, and WS, TN, and DG provided their schemes and/or parameters for the development. CK performed all the
HighResMIP simulations and the sensitivity experiments, transferred the data to ESGF, and wrote a major part of this paper. TS wrote most of Sections 3.1 and 3.2 and TS, ATN, DG, HM, and TN modified the manuscript. All the authors provided advice for the development of NICAM16-S and/or experimental design and reviewed the manuscript.

**Competing interests**

The authors declare that they have no conflict of interest.

**Acknowledgment**

The authors would like to thank Hiroaki Tatebe and Ryosuke Shibuya for discussions on model configurations and Manabu Abe and Takahiro Inoue for technical advice for CMIP6. CK acknowledges Shunsuke Noguchi for discussions on the
vertical resolution of the model. Constructive, and careful comments from two anonymous reviewers and Sophie Valcke, the editor in charge, significantly helped improve the manuscript. CERES, CloudSat, ISCCP, GPCP, and TRMM data were obtained from the National Aeronautics and Space Administration (NASA), JRA-55 data from the JMA, and GridSat data from the National Oceanic and Atmospheric Administration (NOAA). This study was supported by the Environment Research and Technology Development Fund (JPMEERF20172R01) of the Environmental Restoration and Conservation
Agency of Japan (ERCA) and the Integrated Research Program for Advancing Climate Models (TOUGOU) (JPMXD0717935457), the FLAGSHIP2020 project within the priority study4, and Program for Promoting Researches on the Supercomputer Fugaku (Large Ensemble Atmospheric and Environmental Prediction for Disaster Prevention and Mitigation) of the Ministry of Education, Culture, Sports, Science and Technology (MEXT) of Japan and JSPS KAKENHI Grant Number JP20H05728. The HighResMIP simulations and sensitivity experiments were performed on the Earth Simulator at
the Japan Agency for Marine-Earth Science and Technology (JAMSTEC), and some preliminary experiments were performed on the K computer (proposal numbers hp150287, hp160230, hp170234, hp180182, and hp190152).

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

**Tables**

**Table 1: List of HighResMIP simulations.**

| Source ID | HighResMIP Tier | Integration period | Initial atmospheric condition | Initial land condition |
|-----------|-----------------|--------------------|-------------------------------|------------------------|
| NICAM16-7S | 1 & 3 | 1950–2050 (101-yr) | ERA-20C (Poli et al., 2016) | NICAM climatology |
| NICAM16-8S | 1 & 3 | 1950–2050 (101-yr) | ERA-20C | NICAM climatology |
| NICAM16-9S | 1 | 1950–1960 (11-yr) | ERA-20C | NICAM climatology |
| NICAM16-9S | 1 | 2000–2010 (11-yr) | ERA-20C | NICAM climatology |
| NICAM16-9S | 3 | 2040–2050 (11-yr) | The NICAM16-8S Tier 3 run | The NICAM16-8S Tier 3 run |

**Table 2: List of sensitivity experiments.**

| Run name | Descriptions |
|---|---|
| REFFIX | Same as NICAM16-S (with the fixed SST condition; Section 3.7). |
| REFSLB | Same as NICAM16-S but with the slab ocean model and nudging technique (Section 3.7). |
| REF | Alias name of the REFFIX run for 56 km mesh and the REFSLB run for 14 km mesh. |
| NOCLD | Same as the REF run but for using the previous cloud microphysics scheme used in NICAM.12 (Table 5; Section 3.2). |
| NONS | Same as the REF run but for assuming the spherical particle in the radiation table (Section 3.3). |
| NONSI | Same as the NONS run but for using the fixed effective radii of 8 μm for liquid hydrometeors and 40 μm for ice hydrometeors (Section 3.3). |
| NOAER | Same as the REF run but for prescribing zero natural and anthropogenic aerosol mass concentration for the radiation scheme and constant CCN of 50 cm$^{-3}$ for the cloud microphysics scheme (Section 3.4). |
| NOANTAER | Same as the REF run but for prescribing zero anthropogenic aerosol mass concentrations for the radiation scheme (Section 2.3). |
| NOLND | Same as the REF run but for omitting the effects of wetland and water accumulation on land ice (Section 3.5). |
| NOALB | Same as the REF run but for using the previous surface albedo values (Table 6; Section 3.6). |
| NOSIC | Same as the REF run but for using the previous SICCRT value of 300 kg m$^{-2}$ (Section 3.7). |
| NOGWD | Same as the REF run but for switching off the subgrid-scale orographic gravity wave drag scheme (Section 3.8). |
| DDT2M | Same as the REF run but for setting the time step of the dynamics and gravity wave drag scheme to 2 min. Also, the numbers of sub-cycles for cloud microphysics, turbulence and surface schemes were halved to keep their time steps unchanged. |
| DDT1M | Same as the REF run but for setting the time step of the dynamics and gravity wave drag scheme to 1 min. Also, the numbers of sub-cycles for cloud microphysics, turbulence and surface schemes were quartered to keep their time steps unchanged. |
| RDT20M | Same as the REF run but for setting the time step of the radiation scheme to 20 min. |
| RDT10M | Same as the REF run but for setting the time step of the radiation scheme to 10 min. |

**Table 3: List of observational datasetss**

| Short name | Full name | Resolution | Reference |
|---|---|---|---|
| CERES | Clouds and Earth's Radiant Energy System (CERES) Energy Balanced and Filled (EBAF) TOA/SFC Edition 4.0 (Ed4.0) | 1.0°×1.0°, monthly-mean | Kato et al. (2018), Loeb et al. (2018) |
| CloudSat | CloudSat level 2B radar-only cloud water content (2B-CWC-RO) | 0.25°×0.25° | Austin et al. (2009), Austin and Stephens (2001) |
| GPCP | Global Precipitation Climatology Project (version 2.2) | 2.5°×2.5°, monthly-mean | Adler et al. (2003) |
| | (version 1.2) | 1.0°×1.0°, daily-mean | Huffman et al. (2001) |
| GridSat | Gridded Satellite Data – B1 | 0.07°×0.07°, three-hourly | Knapp et al. (2011) |
| ISCCP | International Satellite Cloud Climatology Project | 2.5° × 2.5°, monthly-mean | Rossow and Schiffer (1999) |
| JRA-55 | Japanese 55-year reanalysis | 1.25° × 1.25°, monthly-mean | Kobayashi et al. (2015) |
| TRMM | Tropical Rainfall Measuring Mission (3B42) | 0.25°×0.25°, daily-mean | Huffman et al. (2016) |

**Table 4: Physics schemes in NICAM16-S and NICAM.12.**

| Model | NICAM16-S (NICAM.16 for HighResMIP) | NICAM.12 |
|---|---|---|
| Cloud microphysics | NICAM Single-moment Water 6 (NSW6) (Tomita, 2008; Roh and Satoh, 2014; Roh et al., 2017) | NSW6 (Tomita, 2008; Kodama et al., 2012) |
| Cumulus convection and large-scale condensation | Not used | Not used |
| Radiation | mstrnX (Sekiguchi and Nakajima, 2008), updated radiation table (Seiki et al., 2014), and coupling with cloud microphysics | mstrnX (Sekiguchi and Nakajima, 2008) |
| Turbulence | Mellor-Yamada Nakanishi-Niino level 2 (Nakanishi and Niino, 2006; Noda et al., 2010) | Same |
| Gravity wave | Orographic gravity wave drag (McFarlane, 1987) | Not used |
| Land surface | Minimal Advanced Treatments of Surface Interaction and RunOff (MATSIRO) (Takata et al., 2003) with wetland scheme (Nitta et al., 2017) | MATSIRO (Takata et al., 2003) |
| Ocean surface flux | Bulk surface scheme (Louis, 1979); surface roughness is evaluated following Fairall et al. (2003) and Moon et al. (2007) | Same |
| Ocean treatment | Fixed to observations (or single layer slab ocean with a nudging toward observations) | Single layer slab ocean with a nudging toward observations |

**Table 5: Summary of the key changes in the NSW6 scheme by Roh and Satoh (2014) and Roh et al. (2017)[#].**

| | | NSW6 in NICAM16-S (Roh and Satoh, 2014; Roh et al., 2017) | NSW6 in NICAM.12 (Tomita, 2008; Kodama et al., 2012) |
|---|---|---|---|
| a. | Production of cloud ice | Ice nucleation and vapor deposition are calculated explicitly following Hong et al. (2004). | Cloud water and cloud ice are produced or reduced by saturation adjustment (Tomita, 2008). |
| b. | Terminal velocity of cloud ice | 0 | 0 (same) |
| c. | Size distribution of snow | A bi-modal shape of the rescaled particle size distribution of snow is assumed following Thompson et al., (2008), who used aircraft observations by Field et al. (2005). | Marshall Palmer distributions are assumed for rain, snow, and graupel with global constants of $N_0$ following Lin et al. (1983) and Rutledge and Hobbs (1984), as follow: $N_j(D_j) = N_{0,j}\exp(-\lambda_j D_j), (j = r,s,g)$. |
| d. | Mass and diameter (M-D) relationship of snow | The mass ($m$) and maximum dimension ($D$) relationship of snow assumes two-dimensional fractal shapes ( $m_s = 0.069\,D_s^2$ ) with variable snow density following Thompson et al., (2008). | Ice hydrometeors are assumed as the spherical shape with fixed bulk densities following Rutledge and Hobbs (1984). |
| e. | Intercept parameter in the M-D relationship of graupel | The intercept parameter of graupel $N_{0g} = 4 \times 10^8$ [m$^{-4}$] is used (Gilmore et al., 2004; Knight et al., 1982). | The intercept parameter of graupel $N_{0g} = 4 \times 10^6$ [m$^{-4}$] is used following Rutledge and Hobbs (1984), assuming midlatitude cyclones. |
| f. | Accretion of snow and cloud ice by graupel | Accretion of snow and cloud ice by graupel is ignored following Lang et al. (2007). | Accretion of snow and cloud ice by graupel occurs. |
| g. | Efficiency of accretion of cloud ice by snow | 0.25. | 1.0. |

# The particle size distribution for rain was also revised in the original paper, but the revision is not used in the latest version because it had a small impact and reduced the computational efficiency. In addition, the assumption that cloud ice does not precipitate is inconsistent with some of the other ice cloud microphysics assumptions, but it is used to tune the model to the observed high cloud signals over the tropics.

**Table 6: Surface albedo in NICAM16-S and NICAM.12.**

| | NICAM16-S | NICAM.12 |
|---|---|---|
| Sea ice, VIS and NIR | 0.5 and 0.5 (Hashino et al., 2016) | 0.8 and 0.6 |
| Snow over sea ice, IR | 0.02 (Armstrong and and Brun, 2008; Niwano et al., 2014) | 0 |
| Fresh snow over land, VIS | 0.90 (e.g. Aoki et al. 2011; Yamazaki et al. 1994) | 0.98 |
| Open ocean, IR | 0.05 | 0.005 |

VIS, NIR, and IR stand for visible, near-infrared, and infrared bands, respectively.

**Table 7: Computational aspects of the simulations on the Earth Simulator 3. They are sampled from 6-month simulations (1 July 2004–31 December 2004).**

| Source ID | NICAM16-7S | NICAM16-8S | NICAM16-9S |
|---|---|---|---|
| Number of nodes | 10 | 40 / 160 | 160 |
| Number of MPI processes | 40 | 160 / 640 | 640 |
| File staging | No | Yes / Yes | Yes |
| Simulation year per wall-clock day (SYPD) | 0.42 | 0.37 / 0.63 | 0.22 |

| Source ID | NICAM16-7S | NICAM16-8S | NICAM16-9S |
|---|---|---|---|
| Number of nodes | | | |

**Figures**

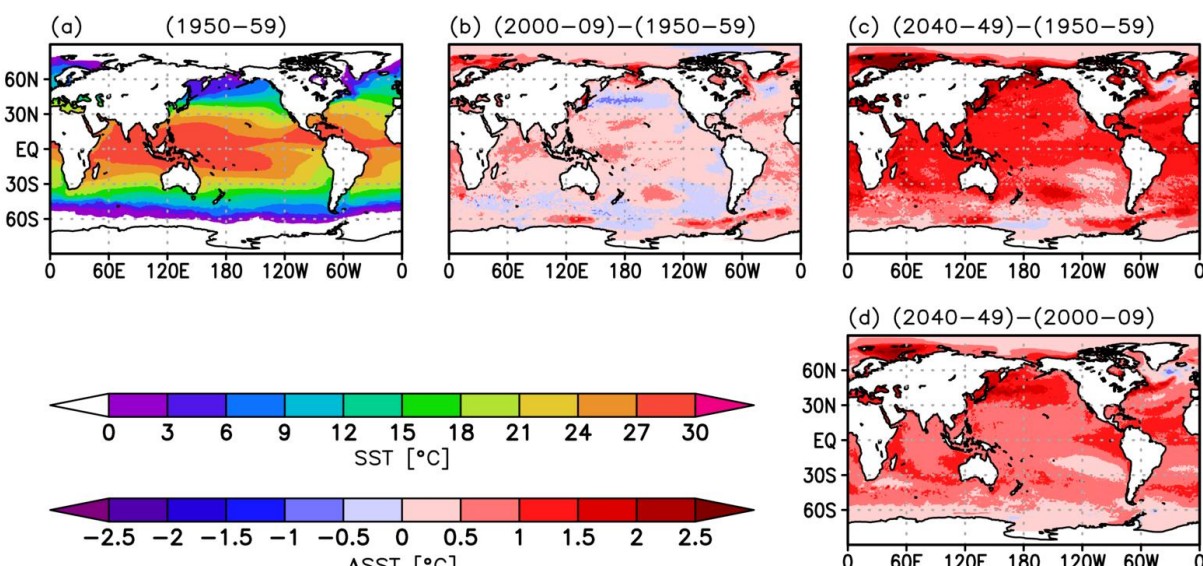

Figure 1: Decadal mean horizontal distribution of sea surface temperature (SST) prescribed in the model averaged for the 1950s (a). Differences between the 2000s and the 1950s (b), the 2040s and the 1950s (c), and the 2040s and the 2000s (d) are also shown. Units are °C.

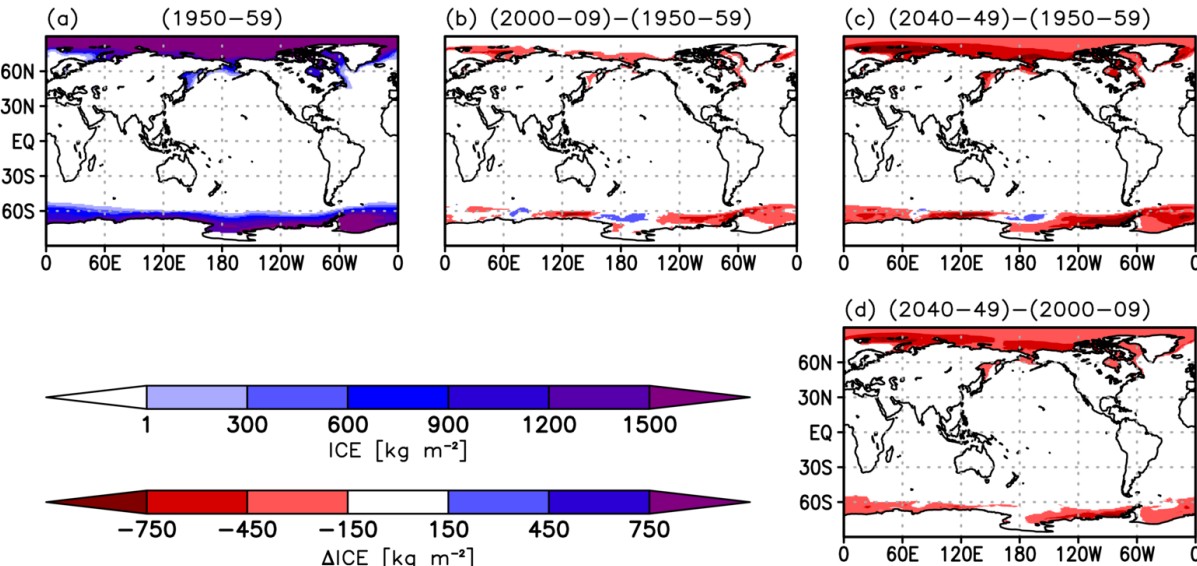

**Figure 2: Same as Figure 1 but for sea ice mass (ICE) in kg m⁻².**

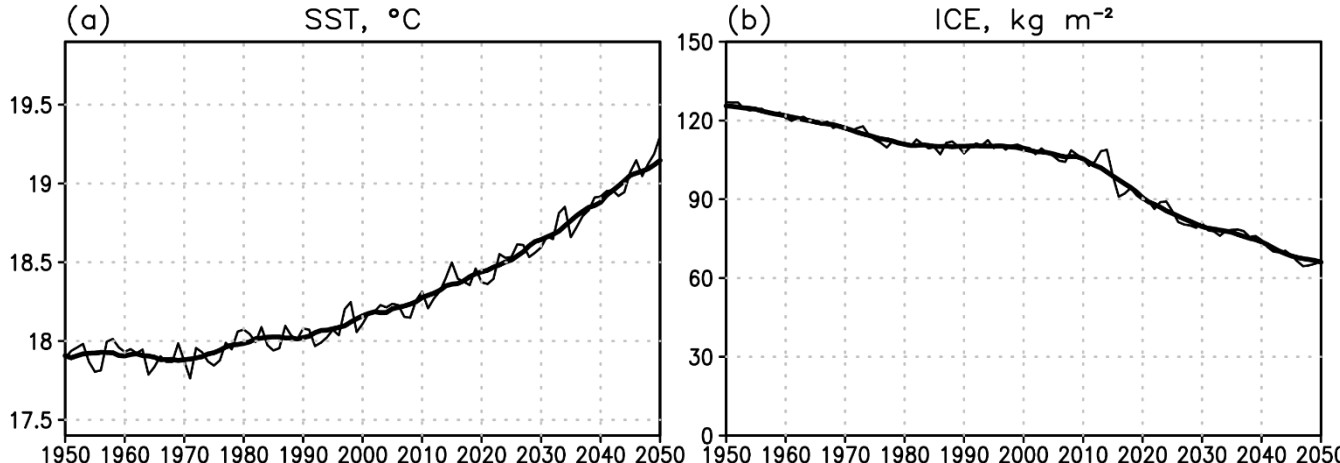

**Figure 3: Global mean SST (a; in °C) and ICE (b; in kg m$^{-2}$) prescribed in the model. Annual mean (thin curve) and decadal running mean (thick curve) are shown.**

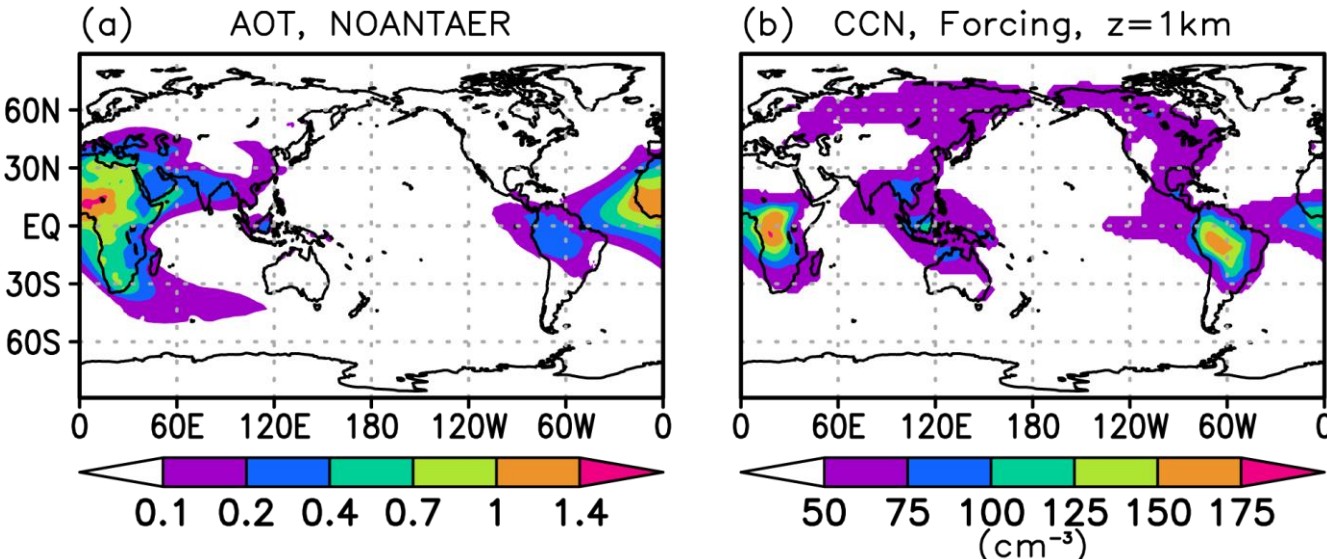

**Figure 4: (a) Annual mean natural aerosol optical thickness averaged for June 2004–May 2005 as simulated by NICAM16-7S (NOANTAER run in Table 2). The data are gridded at 0.56° in longitude and latitude. (b) The annual mean number concentration of cloud condensation nuclei (CCN) from the natural origin at 1 km above sea level prescribed for the model. The data are gridded at 2° in longitude and latitude and the unit is $cm^{-3}$. The lower bound of CCN, 50 $cm^{-3}$ (Section 2.3), is shown in white (no shading).**

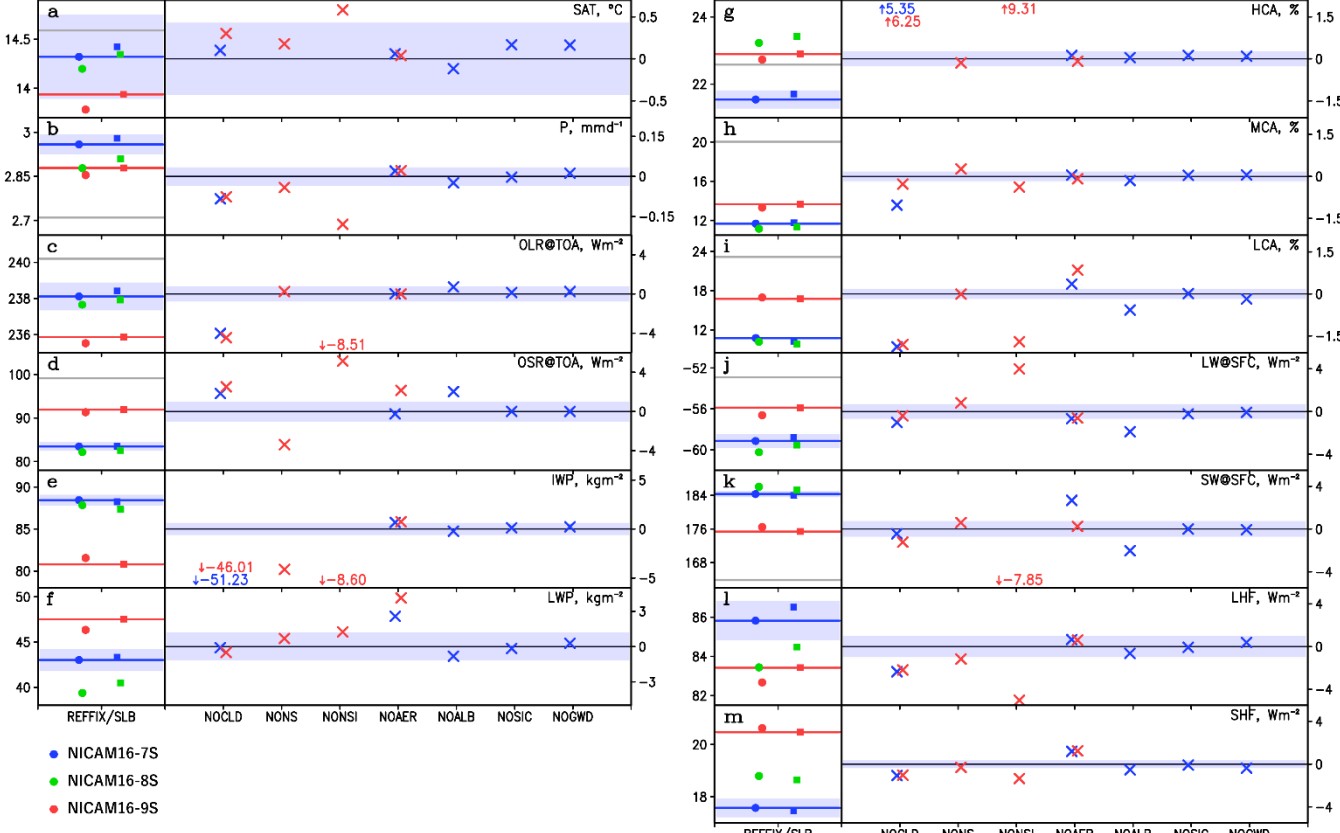

**Figure 5: Global annual means of surface air temperature (a), precipitation (b), top-of-atmosphere (TOA) outgoing longwave radiation (OLR) (c), TOA outgoing shortwave radiation (OSR) (d), ice water path (e), liquid water path (f), high cloud amount (g), middle cloud amount (h), low cloud amount (i), surface net downward longwave radiation (j), surface net downward shortwave radiation (k), surface latent heat flux (l), and surface sensible heat flux (m). They are averaged over June 2004–May 2005, and their units are shown at the upper-right corner of each panel. Blue shading shows the interannual variability (2σ, detrended) estimated from the HighResMIP NICAM16-7S run over 1950–2050 (Table 1). In the left part of each panel contains plots of global annual means simulated by NICAM16-7S (56-km mesh; blue), NICAM16-8S (28-km mesh; green), and NICAM16-9S (14-km mesh; red), which were performed under the fixed SST condition (filled circles; REFFIX run in Table 2) and with the slab ocean condition (filled squares; REFSLB run in Table 2). Blue and red lines are the reference (REF) runs with 56-km mesh and 14-km mesh, respectively. Observational values taken from the JRA-55 reanalysis (surface air temperature), GPCP (precipitation), CERES (radiation), and ISCCP (cloud amount) are shown as gray lines. The right part of each panel shows differences between the REF run and each sensitivity run (NOCLD, NONS, NONSI NOAER, NOALB, NOSIC, and NOGWD runs in Table 2). simulated by NICAM16-7S (56-km mesh; blue multiplication sign), and NICAM16-9S (14-km mesh; red multiplication sign). Results outside the scale range are shown with digits.**

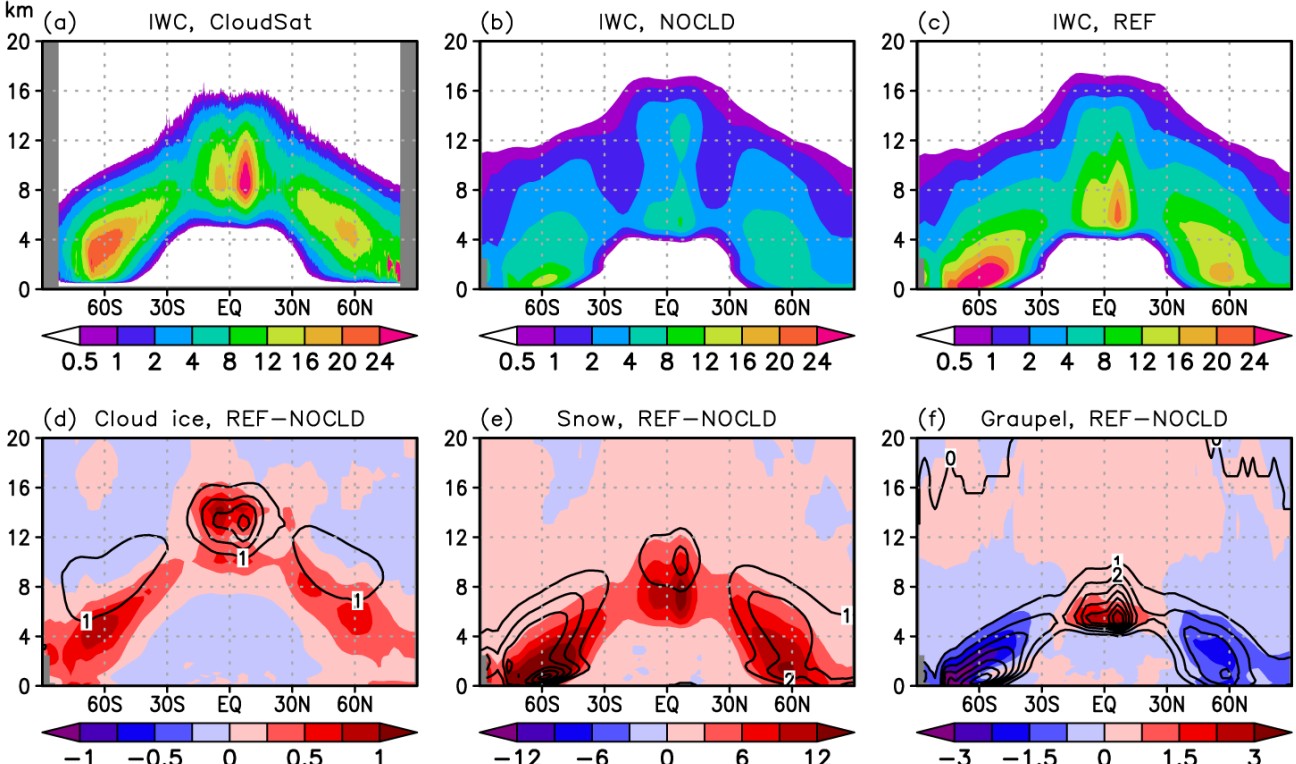

**Figure 6: Annual mean of the zonal mean ice water content ($10^{-6}$ kg m$^{-3}$) in CloudSat observations (a) and the NOCLD (b) and REF (c) runs by NICAM16-9S. Breakdown of the simulated ice water content (IWC) into cloud ice (d), snow (e), and graupel (f) is shown in the bottom panels. The contours show the NOCLD run, and shading shows the difference between the REF and NOCLD runs. The analysis data are 0.25° (a) and 2.5° (b–f) gridded data, and the vertical axis is the altitude in km above sea level.**

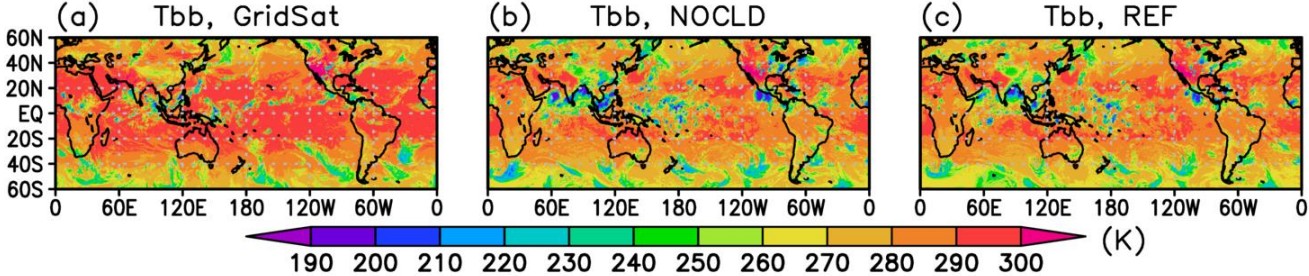

**Figure 7: TOA brightness temperature near 11 μm at 00:00 UTC June 6, 2004, from the GridSat product (a) and the NOCLD (b) and REF (c) runs by NICAM16-9S. The display style follows Figure 1 of Roh et al. (2017). Grid interval in (b) and (c) is 0.14°.**

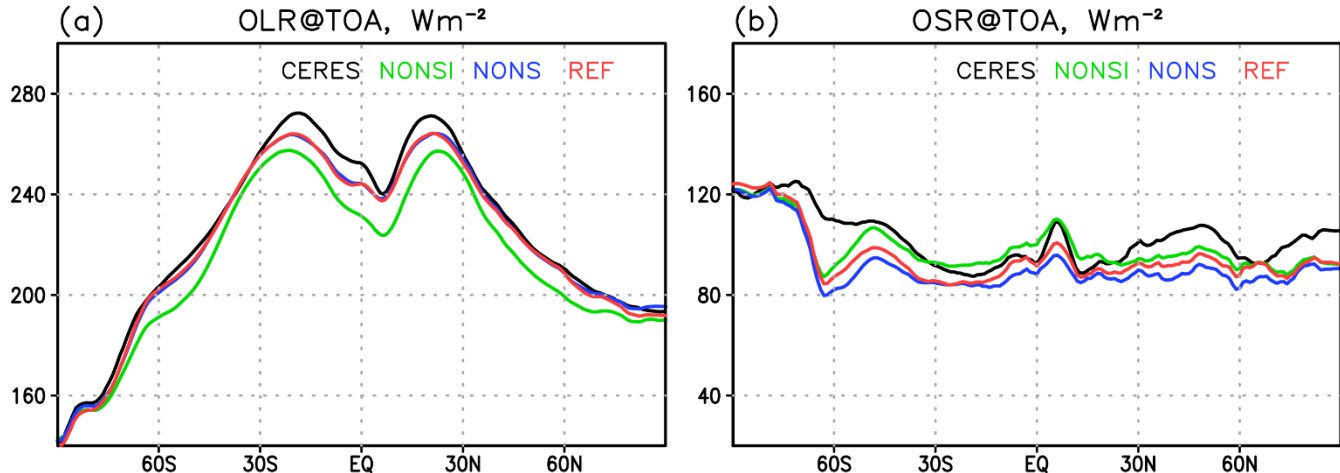

**Figure 8: Annual mean of OLR (a) and OSR (b) at the TOA (W m⁻²), for the CERES product (black) and NICAM16-9S. Green, blue, and red curves show the NONSI, NONS, and REF runs, respectively.**

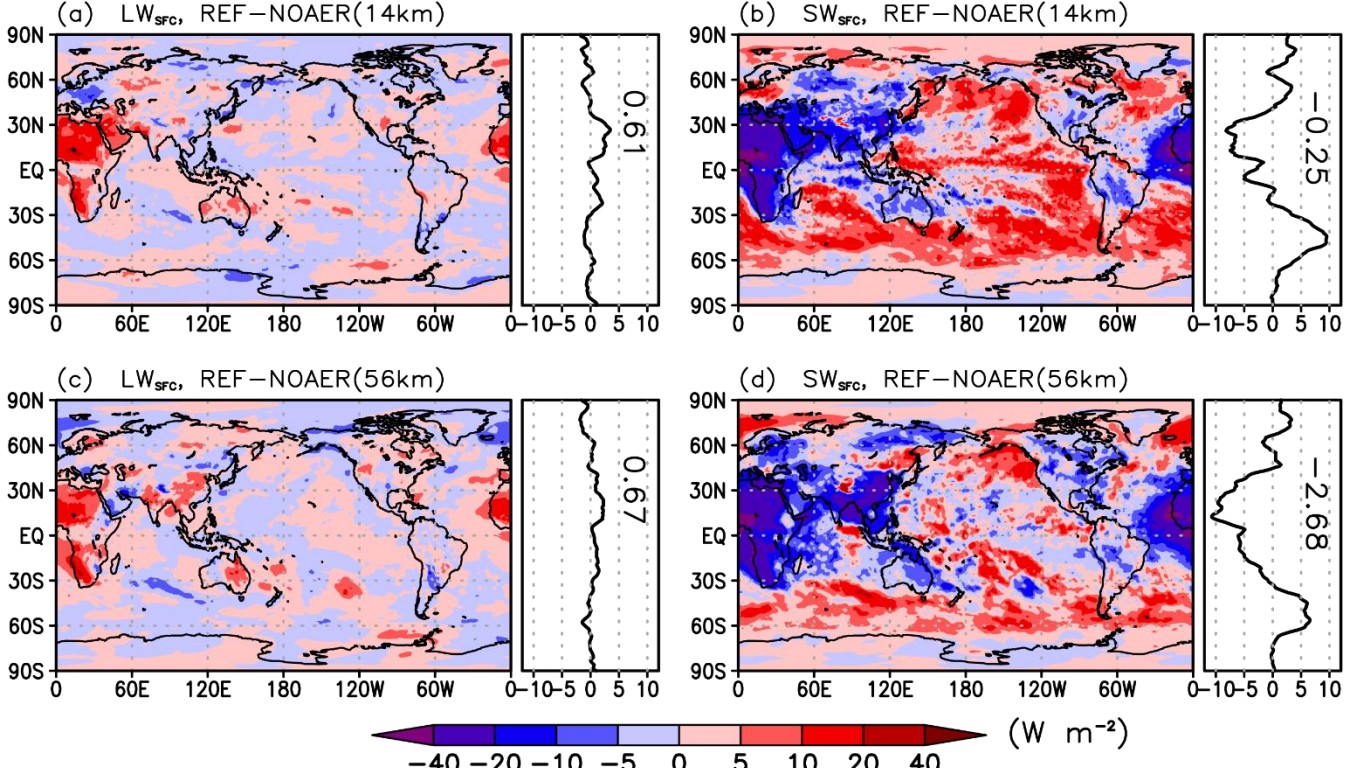

**Figure 9: Impact of the update of the aerosol treatment (the REF run minus the NOAER run). 2-D distribution, zonal, and global means of the net longwave (a, c) and shortwave (b, d) radiation at the surface are shown for the NICAM16-9S (a, b) and NICAM16-7S (c, d) runs. The data are gridded at 1° in longitude and latitude, and the sign of the radiation is downward positive.**

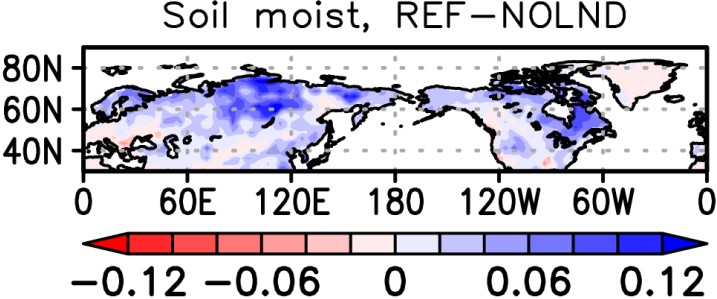

**Figure 10: Impact of the update of the land surface model (the REF run minus the NOLND run) on the simulated soil moisture at the uppermost land surface model level. Simulations using NICAM16-7S were performed for 4 years, and June-July-August data for the last 3 years are averaged. The data are gridded at 2.5° in longitude and latitude.**

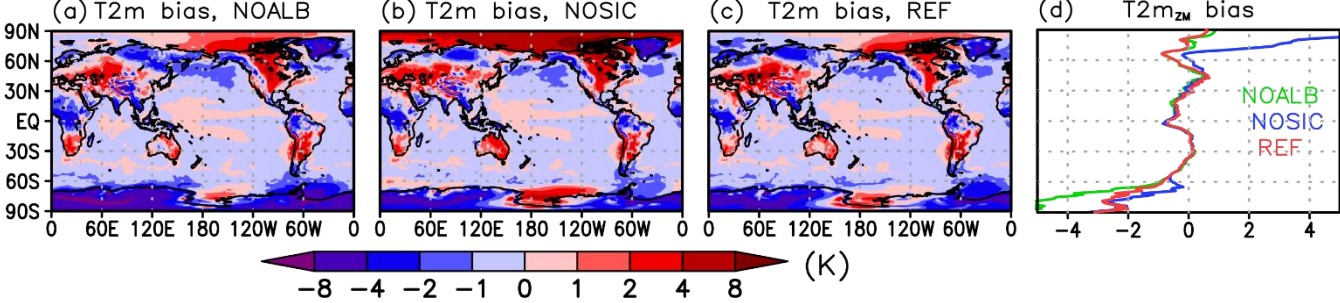

**Figure 11: Bias in the simulated surface air temperature (K) by NICAM16-7S against JRA-55 reanalysis averaged for June 2004–May 2005. The NOALB (a), NOSIC (b), and REF (c) runs are shown. The zonal mean biases of the surface air temperature for the NOALB (green), NOSIC (blue), and REF (red) runs are shown in (d).**

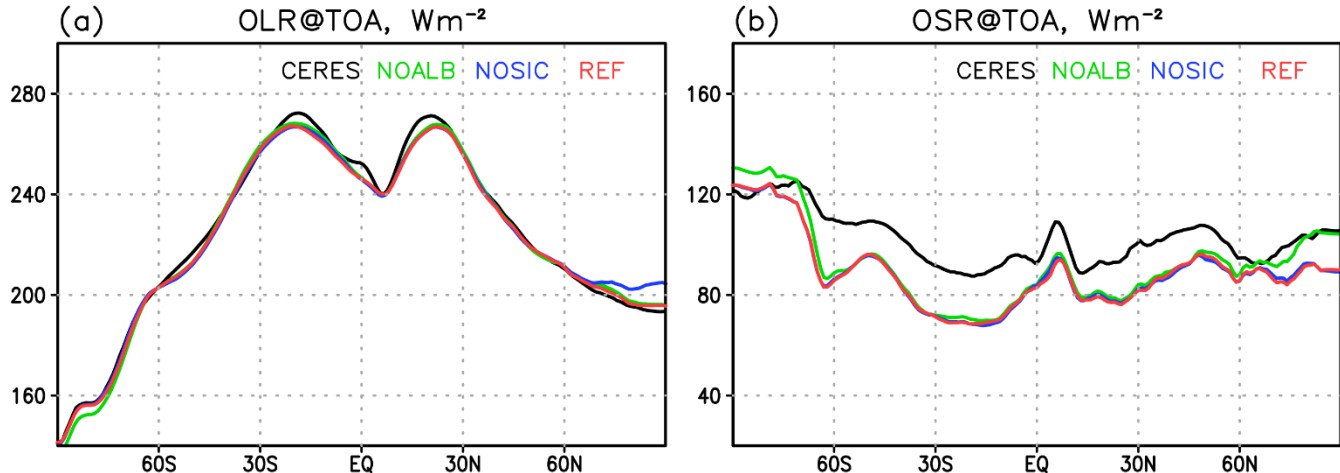

**Figure 12: Same as Figure 8 but for the NOALB (green), NOSIC (blue), and REF (red) runs by NICAM16-7S.**

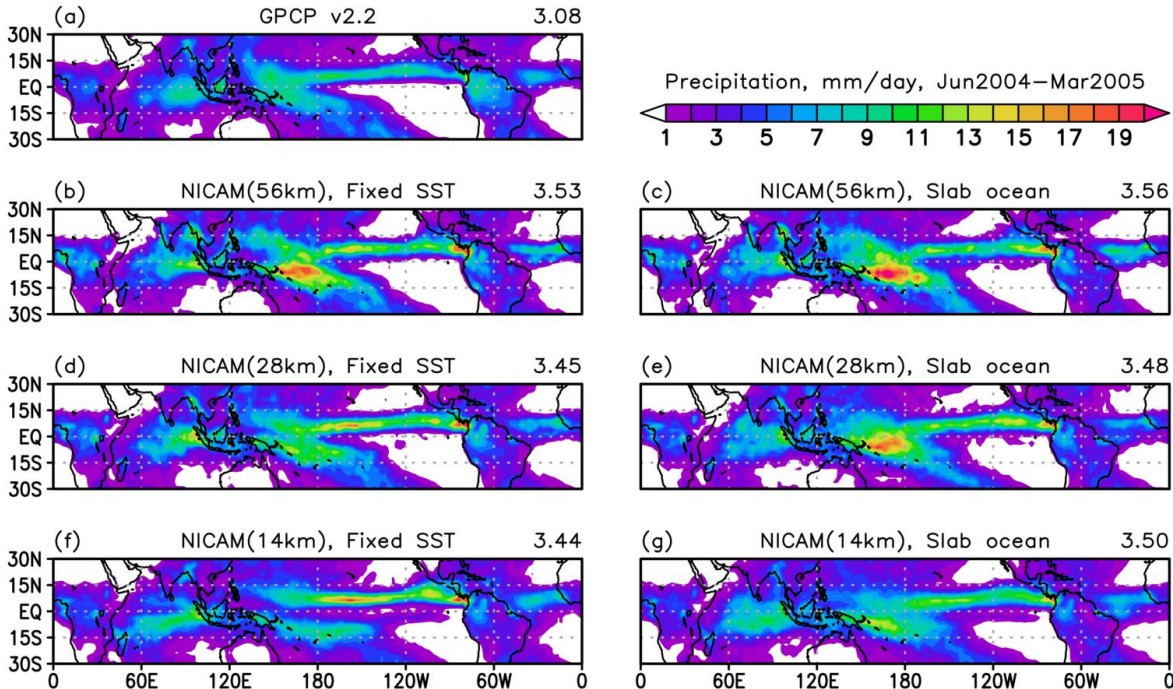

**Figure 13: Annual mean precipitation during June 2004–May 2005 for the GPCP product (a), the REFFIX runs (b, d, f), and the REFSLB runs (c, e, g). Results from NICAM16-7S (b, c), NICAM16-8S (d, e), and NICAM16-9S (f, g) are shown, and tropical mean precipitation is noted at the top-right corner of each panel.**

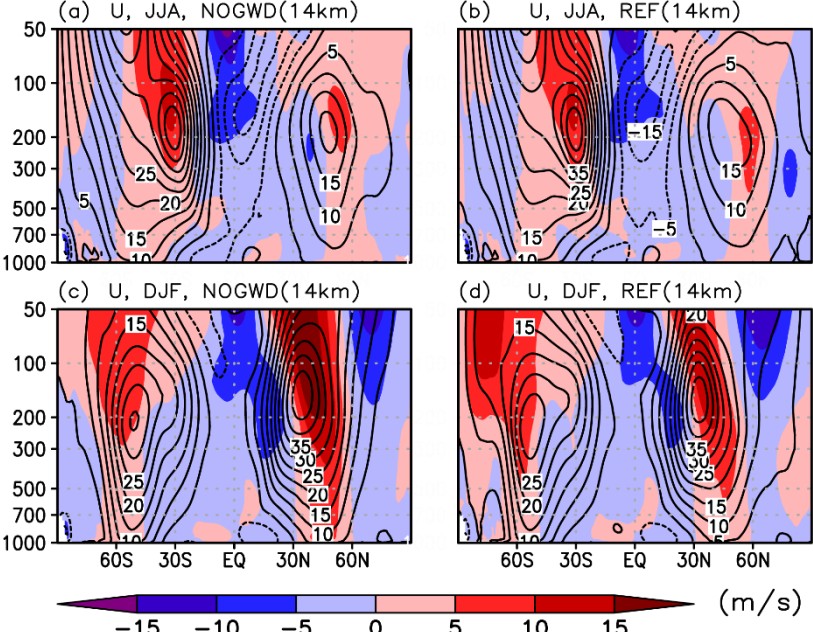

**Figure 14: Zonal mean zonal wind (contours) and its bias from JRA-55 reanalysis (shadings) for June–August 2004 (a, b) and December 2004–February 2005 as simulated by NICAM16-9S without (a, c; NOGWD run) and with (b, d; the REF run) the gravity wave drag scheme.**

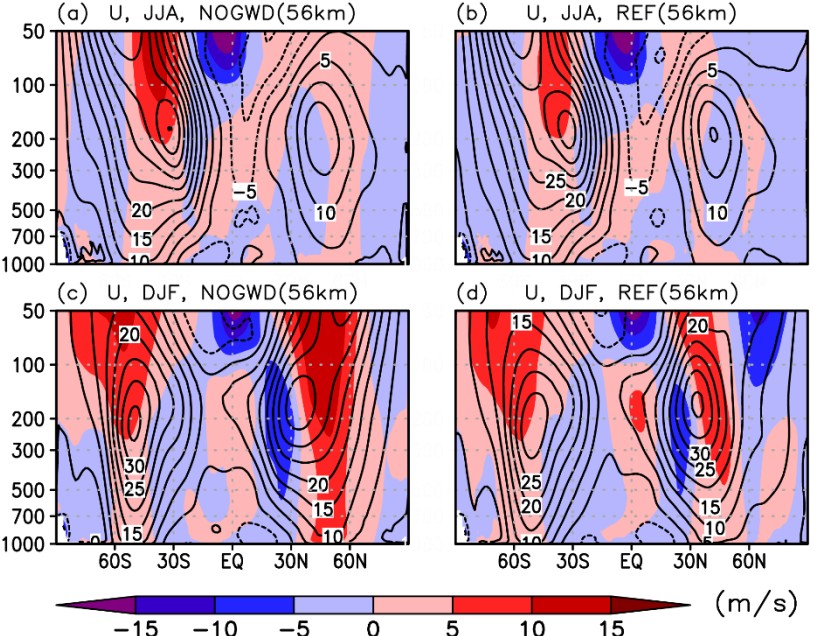

**Figure 15: Same as Figure 14 but for the simulation by NICAM16-7S.**

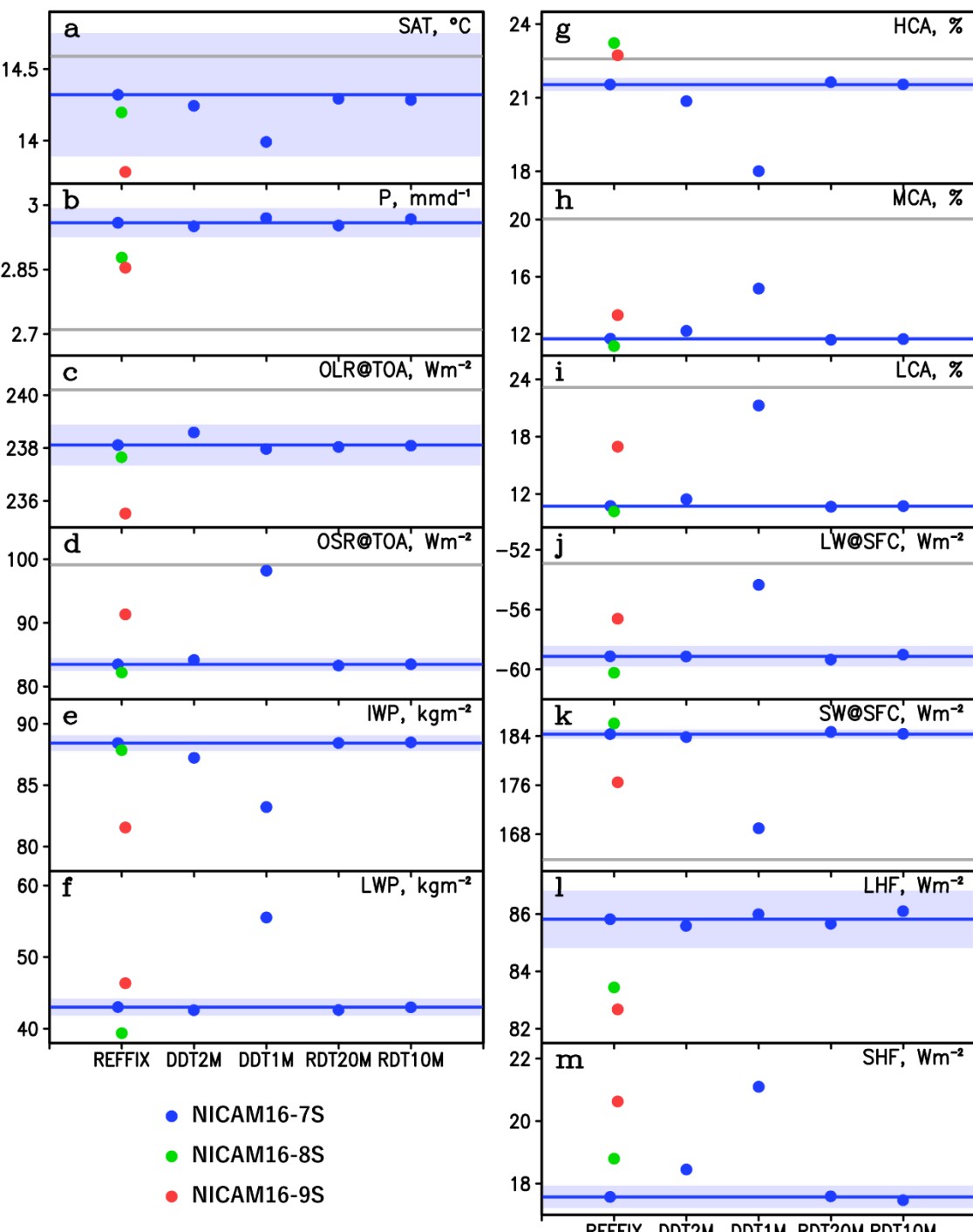

**Figure 16: Same as the left part of Figure 5 but for the REFFIX, DDT2M, DDT1M, RDT20M, and RDT10M runs.**

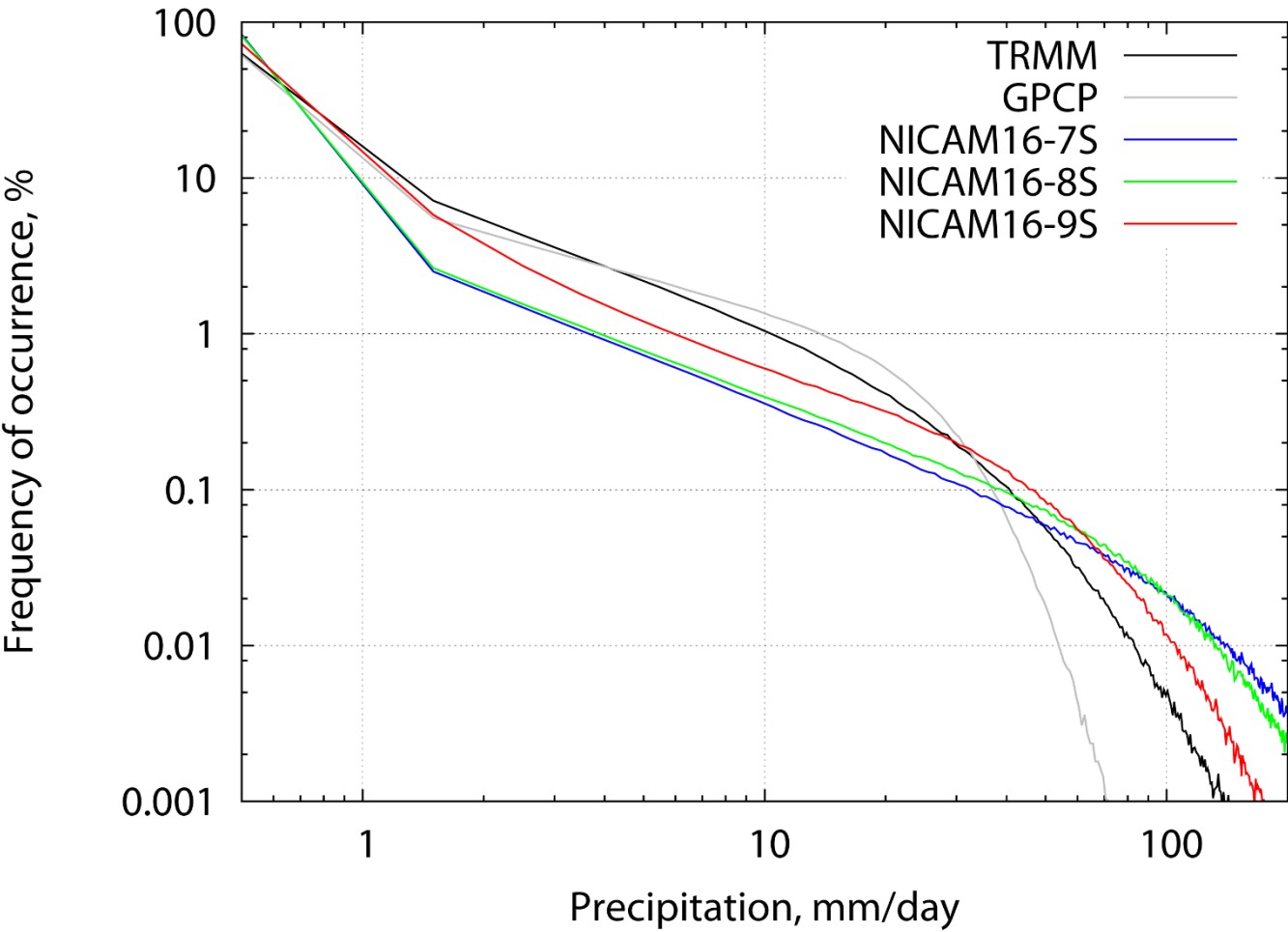

**Figure 17: Frequency of occurrence (%) of daily mean precipitation binned with an interval of 1 mm day$^{-1}$ during 01 June 2004– 31 May 2005 as averaged over 15°S–15°N. The GPCP and TRMM products are shown with black and gray curves, respectively. The REFFIX runs with NICAM16-7S, NICAM16-8S, and NICAM16-9S are shown with blue, green, and red curves, respectively. The data were re-gridded at 1° in longitude and latitude before the sampling.**

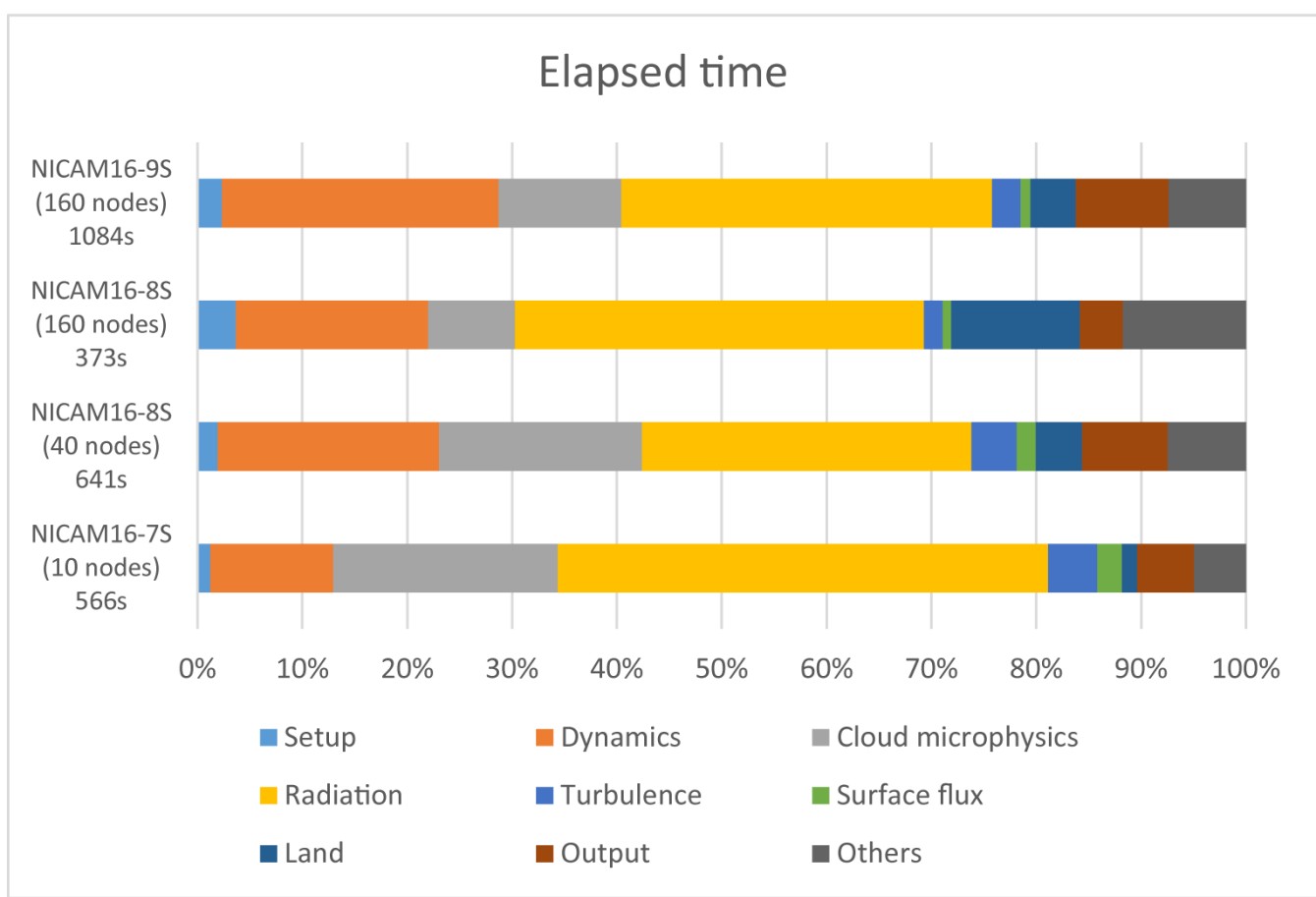

**Figure 18: Percentage of elapsed time for each component of NICAM16-S on the Earth Simulator 3. Times were sampled from 6-month simulations (1 July 2004–31 December 2004). The computational time per 1-day integration is shown on the left.**

