# Peer review of "The non-hydrostatic global atmospheric model for CMIP6 HighResMIP simulations (NICAM16-S): experimental design, model description, and impacts of model updates"

_Geoscientific Model Development, 2019_

## Referee Comment (RC1) · Anonymous Referee #1 · 21 May 2020

The paper describes the new version of the global non-hydrostatic could-system re-solving model, NICAM. The goal of the paper is twofold (1) evaluate several recent updates which brought the model from version NICAM.12 to the newest NICAM.16; (2) describe additional developments that were made necessary to adapt the model to the HighResMIP (CMIP6 endorsed MIP) protocol and that were introduced in the specific configuration NICAM16-S. The description and evaluation of those changes is valu-able, and the paper will undoubtedly serve as a reference for NICAM16-S in studies analysing HighResMIP models. However, the paper does not investigate the impact of

model resolution on the model climatology or only marginally. The reader understands only at the end of the paper that it is a deliberate decision and that the impact of resolution will be presented in another paper in preparation. This is a surprising choice, as most people would expect a reference paper of a new model configuration participating in HighResMIP to have horizontal resolution as its main focus. This makes me wonder if the present paper should not be limited to the description of the new model NICAM.16, leaving the developments for CMIP6 to another HighResMIP paper in which the impact of resolution would be investigated in more detail? The quality of the writing in unequal, some sections (e.g, abstract, introduction of section 3 and section 3.1) fall short of meeting the standards of a journal such as GMD, while other sections (e.g. 3.2, 3.3) are written in a very good english. I would recommend a collective effort to improve and homogenise the quality of the text throughout the manuscript. I believe the paper requires major revisions before being published and I would like the authors to answer more specifically the following comments :

Main comments :

1) Could you please clarify both in the abstract and in the introduction to which MIP of CMIP6, NICAM will participate? Am I right to understand that they will only participate to HighResMIP and they will not submit simulations to the DECK? The author is left long to speculate about that. The confusion also arises from the fact that CMIP6 and HighResMIP are sometimes used interchangeably (e.g. abstract line 16 vs line 19). I would recommend to use HighResMIP as often as possible, as it is more specific.

2) My main issue is that the paper describes a new set-up for HighResMIP but there is absolutely no description of the impact of a change in resolution on the simulated climate. Even section 4, whose title announces an investigation of the dependency to horizontal resolution has only three lines about resolution (l. 7 to 10). Is there a convergence of those statistics when horizontal resolution is increased? I believe you need to inform the reader in the abstract that you do not discuss the impact of resolution.

3) The HighResMIP protocol stipulates : "For a clean evaluation of the impact of horizontal resolution, additional tuning of the high-resolution version of the model should be avoided. The experimental set-up and de- sign of the standard resolution experiments will be exactly the same as for the high-resolution runs" (Haarsma et al., 2016). Have you performed specific retuning at each resolution? Please mention explicitly which resolution has been tuned first and what was the procedure and the parameters which were adjusted. In addition, mention any additional tuning specific of each resolution.

4) You do not comment on the effect of changing the time step of the radiation scheme in NICAM16-7S to 9S (from 40 to 10min) and changing the time step of the dynamics from (240 to 60s) will have on the climatology (precipitation for instance in Table 6 and figure 11). In addition, and related to the previous question, how will changes in the time step be distinguished from the direct impact of increasing horizontal resolution in HighResMIP? Have you done additional sensitivity experiments? I believe the paper should address this issue.

5) There are four levels of labelling in the paper which makes it sometimes difficult to follow : (1) the different versions of NICAM.12 and 16, (2) the configuration for High-ResMIP NICAM16-S, (3) the various resolutions 7S, 8S, 9S, (4) and the sensitivity experiments (g, f, . . .). Labels are sometimes redundant NICAM16-S and g for instance refer to the same simulations. I believe you might need to keep both (to remain consistent with the naming already communicated to CMIP6) but you need to refer consistently to those different labels throughout the paper and I feel it is not always the case. -> NICAM.16-S rather than NICAM16-S is used in many places. -> Most sensitivity experiments are listed in Table 2 but not all. Could you give an experimental id to simulations described in section 3.5 and column Table 2e? -> The experimental id are not mentioned in the text after section 3.4 (only in tables and captions) whereas they are used in the text before section 3.3. Please can you at least recall once what they are (maybe when you list the sensitivity experiments at the beginning of the section).

6) You make the choice not to use a convection scheme. This has been tested

in several models at resolutions where convective processes are not yet resolved : please cite references which have tested a similar approach (see for instance Hohenegger et al 2020, https://www.jstage.jst.go.jp/article/jmsj/98/1/98_2020-005/_html/-char/en and references therein). You explain that not having a convective parameterisation results in more patchy precipitation (page 6, line 27) and it would be interesting to illustrate that (see for instance Figure 2 in Maher et al, 2018, https://agupubs.onlinelibrary.wiley.com/doi/full/10.1002/2017GL076826)

7) The beginning of the introduction is a bit confused, both climate sensitivity and climate impacts are mentioned and it is unclear why. What not saying from the beginning that the accurate treatment of cloud requires high-resolution cloud resolving models. You could also cite the review paper by Bony et al. 2015, https://www.nature.com/articles/ngeo2398) in this paragraph.

8) Section 2.2: Please be more specific on the initial land conditions in NICAM16-7S and NICAM16-8S. Are they derived in a similar way as NICAM16-9S? this is not clear.

9) The change of SICCRT between the AMIP and slab experiments is very large (a factor 5!). Could you please explain if there is any resulting inconsistency between the AMIP and slab experiments for SIC, which I believe is a standard diagnostic of CMIP6?

10) Page 15 : you indicate that you will share regridded data with CMIP6 at resolution of 1degree or coarser. Will you be able to provide higher-resolution fields on demand? HighResMIP has a special focus at fine scale features, such as tropical cyclones, extreme precipitation, for which high-resolution data might be needed.

Specific comments : page 1, line 2 : Experimental -> experimental page 1, line 6 : the Coupled Model Intercomparison Project Phase 6 page 1, line 17 : the land surface model (and everywhere thereafter) page 1, line 18 : an improvement of the coupling page 1, line 19 : and the radiation schemes; ... to follow the protocol of the CMIP6 High... page 1, line 21 : the impacts of the various model updates page 1, line 23 : over Africa and South Asia page 1, line 29 : redistributes mass page 2, line 22 : nonsphericity of ice particles page 2, line 26 : "That is, the interfaces" => unclear, please rephrase page 3, line 9 : NICAM17-nS -> NICAM16-nS. page 3, line 15: including tropical cyclones page 3, line 31: using -> with page 4, line 4: The initial land conditions . . . were page 4, line 6: under present-day conditions . . . the last 5 years of data page 4, line 14 : are derived from "g" . . . 1 une 2004 to ensure consistency with previous NICAM studies page 4, line 19 : Is SST an external forcing? I don't think it is what people mean by external forcing. page 4, line 32 : fixed SST conditions ere used in the 56km meh run page 4, line 33 : was used in the 14 km mesh runs page 5, line 4 : Future change in SST is somewhat similar to the El Nino pattern => personally I don't think so, there is a warming everywhere! You could mention with a larger warming in the equatorial Pacific. page 5, line 9 : ICE is nearly page 5, line 11 : prescribed in the model page 6, line 8 : a smoother is applied -> a spatial filter is applied to smooth page 6, line 18 : could you give a reference for the various schemes? page 6, line 21 : sentence is too long. page 6, line 32 : global means -> global mean climate page 7, line 10 : We used . . . -> the sentence is unclear. Do you use it to show improvements or because it shows improvements? page 7, line 13 : was originated -> originated page 7, line 21 : is the key -> represent a significant change page 7, line 26 : comparing -> compared page 7, line 32 : accounts for the snow category [this is a typical example where a simple grammar mistake can loose the reader. I thought a new special category was created.] page 8, line 5 : midlatitude storm-track page 8, line 7 : 399%, respectively page 8, line 10 : the accretion page 8, line 15 : In addition, the decrease page 8, line 21 : The low cloud amount is increased as a result of a compensation . . . in medium and thick clouds and a decrease in thin clouds. page 8, line 28 : check punctuation page 8, line 31 : consistent assumptions of coupling . . ... can reduce the model biases page 10, line 8-9 : reference needed. page 11, line 5 : capital letters for the model name. page 11, line 7 and line 10 : the model name is NICAM16-S not NICAM.16-S (the same mistake occurs in other places) Figure 9 : is it possible to have an idea of the fractional reduction of the bias? page 12, line 5 : The depth of the slab End of section 3.6 : please mention when a result is not shown. page 13, line 2,3 : the precipitation

-> precipitation; line 5 : remove coma page 13, line 8 : is tested -> is used / or / introduced in the model (and please mention that no gravity wave drag was used in NICAM.12) page 13, line 21 : it may not be a wise choice -> introducing such a gravity wave drag scheme will not necessarily lead to an improvement of the simulated climate. page 14, line 3-7 : this is a repetition of things which have already been introduced in previous sections. page 14, line 9 : NICAM-7S -> NICAM16-7S page 14, line 19 : greater than SYPD ??? How many? page 14, line 20 : please mention the number of cores per node (I guess 4 from the table 9?) page 15, line 2 : in an icosahedral grid -> on the model's native icosahedral grid ? page 15 : Summary section : no capital after " : " in this section page 15, line 28 : describe ... model description => this is a bit redundant page 29, line 4 : hygroscopity -> hygroscopicity page 30 : ocean model -> ocean / or / ocean treatment (this is because you mostly use SST) page 32 : global mean impacts -> difference of global mean variables between control "g" and sensitivity experiments page 32 : it would be nice if you could highlight in bold where the difference is statistically significant? page 33 : NICAM.16-S -> NICAM16-S and phrase the legend similarly to Tab 5. page 34 : NICAM.16-S -> NICAM16-S page 34 : line 5, rephrase the end of the sentence. What does ad hoc mean here? page 36 : what does Output size in latitude-longitude grid per year TB means here? page 38 : The same as -> Same as page 39 : prescribed in the model ; decadal running mean page 40: NICAM16-7 -> NICAM16-7S. Could you add a reference in this figure? page 41 : line 3 : "g3 and g, respectively". page 41 : units of the vertical axis?

---

## Referee Comment (RC2) · Anonymous Referee #2 · 23 May 2020

In this manuscript, the authors detail the particular configuration of NICAM used for the High Resolution Model Intercomparison Project (HighResMIP). This is using NICAM16 instead of NICAM12, a previous version used for CMIP-class experiments. Updates are described in components such as microphysics and the land surface. The mean climatology at three different resolutions (56, 28, and 14km) and a few basic sensitivity experiments are discussed. The authors finish by discussing computational performance and post-processing needs.

I assume the primary purpose of this manuscript is to detail the particular configura-

tion of NICAM that is used in HighResMIP so it can serve as a reference for scientific papers using such datasets. As such, the paper really doesn't describe any new science; rather, just discusses particular aspects of a specific model configuration. This seems acceptable for a journal such as GMD, even if the results are overly novel from a scientific perspective.

I find it to feel somewhat hastily thrown together. Some details regarding NICAM16 are discussed in detail, others are left to the reader to try and track down. Data isn't always presented in the cleanest manner, making jumping from figure to table a bit difficult. Some figures need work, including axis labels and resizing. In some ways, the manuscript feels approximately 75% finished, thrown together a bit quickly with some holes that need to be filled and smoothed over before publication. There also is a bit of a mix of 'model description' and then 'high-resolution evaluation,' although the authors then note that more formal climate evaluation is left for future work. I would perhaps focus most of the time in this manuscript on explicitly defining the precise design choices for the contributed runs.

I recommend major revisions to clean many pieces of this up and make it more useful as a basic reference for future users of HighResMIP data who wish to learn more about how NICAM operates.

The manuscript reads somewhat disjointed, as if multiple authors were e-mailed and asked to 'provide a paragraph or two' and it was eventually stitched together. Some passages are riddled with grammatical errors, while others are much more cleanly written. Although it didn't rise to the level of making the manuscript illegible, I recommend a thorough read-through by one or two proficient English speakers before submission to clean as many of these up before proofreading as possible. Even small corrections to tense and terminology would make for a much more pleasant read.

Major comments

- Tables 5 and 6 need to be better presented. I am not sure why Table 5 only shows differences between model simulations and Table 6 shows a mean climatology for the three different resolutions. Without the mean values, the numbers in Table 5 are relatively meaningless, as it is tough to gauge how large the changes are relative to the base (reference) state and whether these changes are moving values towards or away from observations at the global level. The easiest thing to do here would be to effectively combine Tables 5 and 6, with mean climatology presented in additional columns, so it is trivial for the reader to mentally process what the difference in the sensitivity experiments (e.g., g-g3, etc.) actually mean.

- Page 4, Lines 16-17. Is one year enough to get usable climate signals here? I have generally understood the rule of thumb to be at least a few years, if not a decade to ensure differences are driven by design choices and not internal variability. How are the authors confident they are not confounding these?

- Page 9, Line 28. I cannot find the g9 simulations in the tables, is there a reason they are not included like the other sensitivity experiments?

- The naming convention is fairly confusing and there are times when names are redundant and refer to the same simulation (i.e., the 'g' simulation refers to a control run, which is occasionally referred to as NICAM16 or NICAM16-S). Is there a particular reason why these naming conventions are used. Are there ways to simplify this so that they are more clear 'in-text.'

Minor comments

- Page 2, Lines 6-7. I am not sure exactly what is meant by 'cloud-*system* resolving climate simulations.' I'd argue cloud-resolving simulations really need to be

**[GMDD](/)**

Interactive
comment

O(1km). A cloud 'system' may be a larger feature, but I can't recall seeing this as common parlance.

- Page 2, Lines 26-30. Does this mean that NICAM16 is the first NICAM version to allow for transient CMIP forcing, or does this mean special code was added for only HighResMIP/CMIP6?

- Page 3, Line 14. 38 vertical levels seems low, particularly for a 14km experiment. Assuming the levels are not evenly spaced, this implies a $dz$ of greater than 1km toward model top, which is really pushing the common notion that $dx >> dz$. The authors later discuss higher vertical resolution, more information should be added about the potential impact of this in HighResMIP, especially if prior work can be cited.

- Page 3, Line 25. More information is needed about timestep of the gravity wave drag, boundary layer parameterization, etc. Are these called at the same timestep of the dynamics? Is the dynamics subcycled?

- Page 4, Line 10. How quickly does the land spin up from this state? Within days, weeks, months? This may be important given the some of the short runs.

- Page 4, Line 24. Is SST 'standardized' in HighResMIP (i.e., do all models use the same file?) or was this specific to NICAM16? I would also quibble that this is more of a boundary condition than an 'external forcing.'

- Page 6, Lines 18-22. Regarding the dynamical core, diffusion, boundary layer parameterization, etc. it is critical that they at least cite previous work when discussing these aspects where interested parties can get model details. Preferably, they would use 1-2 sentences to explain such components and then refer readers to more detailed publications for further information.

- Page 6, Lines 22-25. 'Although most climate models... in the future.' I am not sure I philosophically agree with the notion of removing convective parameterization even at 56km (this would imply extremely large grid point updrafts in my experience). That said, this sentence is long and preferably requires further justification. Has anyone from the NICAM team published a paper regarding their philosophy around the lack of convective parameterizations, even coarser than 20km?

- Page 7, Lines 32-33. I am not sure this is 'more than twice,' but this is where the aforementioned reformulation of Tables 5 and 6 would be quite helpful.

- Page 8, Line 2. '... graupel in the simulation.' Which one, the reference?

- Page 8, Line 19. 'was replaced with zero ... whereas it was zero and unchanged in this study.' I'm a bit confused – the sentence makes it seem like the study applied something different than Roh et al. but it seems like it was zero in both cases?

- Page 11, Line 2. This is quite a large resolution sensitivity (the aerosol forcing completely changes sign going from 56km to 14km if I interpret this correctly).

- Fig. 9. This needs to be bigger. Perhaps stack the three panels vertically?

Typographical errors and grammar

- Page 3, Lines 17-18. Awkward grammar and typos.

- Page 11, Line 5. The first letters used in the acronym should be capitalized.

- Table 3. 'Laege' should be 'large.'

- Fig. 4, The color bar should read 50 and not 50.01.

- Fig. 5, Label the order of differencing for the lower three panels (e.g., g-g3).

- Fig. 5., are the units on the vertical axis 'km?'

[Figure]

---

## Author Comment (AC1) · 4 Aug 2020

Please see the supplement pdf for the response.

Please also note the supplement to this comment:
https://gmd.copernicus.org/preprints/gmd-2019-369/gmd-2019-369-AC1-supplement.pdf

2020.

---

## Author Response (AR1)

Reply to the referee comments (RC1 and RC2) on "The non-hydrostatic global atmospheric model for CMIP6 HighResMIP simulations (NICAM16-S): Experimental design, model description, and sensitivity experiments" [gmd-2019-369], by C. Kodama et al.

Thank you for kindly reviewing the manuscript. We are pleased to have constructive and favorable comments from the two anonymous referees, that significantly improved the manuscript. Below we answer all the comments from the referees and show how the manuscript was revised following the referee comments.

In the followings, we show the revision in two steps: (1) changes suggested point-by-point by the referees (and some minor changes by us), as posted on 4 Aug 2020 as the AC1 (https://gmd.copernicus.org/preprints/gmd-2019-369/#discussion) and (2) changes to homogenize the manuscript and to improve English presentation, which are suggested by both referees. The former changes, explained in Page 2-43 in this response file, are the main part of the revision. Please see the above URL for the modified manuscript with the change history in (1). The latter changes, which do not affect the meaning of the manuscript, are summarized in Page 44 in this response file. The final revised manuscript with the change history from the original manuscript is attached in the end of this response file.

**Step 1. Changes suggested point-by-point by the referees (and some minor changes by us)**

In the following, the referee comments (RC1 and RC2) are shown in *maroon italic text* and the original and the revised texts are shown in *quoted purple italic text*. Note that some of the revised texts were overwritten in a minor way in the Step 2. All the section, page, line, figure and table numbers are based on the original manuscript except for authors' specific changes. References used in the responses are listed after the responses to RC1 and RC2.

In addition to the revisions suggested by the referees, we modified the manuscript to enhance its readability and clarity. Specifically, we

- added *"A double-moment cloud microphysics scheme is also available in NICAM.16 (Seiki and Nakajima, 2014; Seiki et al., 2014, 2015b; Satoh et al., 2018). However, the double-moment scheme was not used for the HighResMIP simulations and hence is not described in this paper."* in page 2, line 28 to contrast NICAM16-S with NICAM.16,

- replaceed title of the Section 2.2, *"Initial conditions"*, with *"HighResMIP simulations and sensitivity experiments"* to describe each experiment together in one place for readability,

- moveed *"As noted ... (see Section 3.6)"* in page 7, lines 2-5 to the second paragraph of Section 2.2 and slightly modify it as *"As noted in Section 3.7, we often prefer to use a slab ocean model with nudging toward the boundary SST rather than the fixed SST condition requested by the HighResMIP protocol because of better performance in the simulated precipitation pattern (Kodama et al., 2015), particularly with a horizontal mesh size of 14 km (Section 3.7). Therefore, both the fixed SST and slab ocean configurations (REFFIX and REFSLB runs, respectively) were tested in the sensitivity experiments."* (see **Response1-5** for the modified run name),

- deleted *"In the short-term sensitivity experiments ... unless explicitly specified."* in page 4, lines 30-33 to avoid duplication with the above sentences in Section 2.2,

- deleted *"Above the tropopause ... external conditions (Section 2.3)"* in page 10, lines 24-26 to avoid duplication with a sentence in Section 2.2,

- merged *"However, fixed SST simulation ... is often preferred."* in page 12, lines 21-22 and *"The introduction of the slab ocean model ... the cloud and precipitation system"* in page 12, lines 26-27, as *"However, the fixed SST simulation is known to cause severe bias in the horizontal distribution of clouds and precipitation system in the tropics (Kodama et al., 2015; Figure 13), and thus the use of the slab ocean model with a 7 day relaxation time is often preferred."* in page 12, line 26,

- replaced *"As the resolution increases, ... with different resolutions."* in lines 27-29, page 14 with

*"As the resolution is increased, the percentage of the dynamics is increased and that of the cloud microphysics and turbulence processes is decreased because of their invariant time step interval among the models with different resolutions. An increase in the percentage of the land surface scheme in the simulations with higher resolution and with greater number of computational nodes seems to be caused by the node imbalance associated with land-ocean distribution."* to add sufficient explanation on Figure 14 (Figure 18 in the revised manuscript),

- changed label of Figure 14 (e.g. *"gl8-p640"* -> *"NICAM16-8S (640 nodes)"*) for clarity, and
- removed the run name of the HighResMIP simulations (which is not used in the manuscript) and simplify Table 1, as shown below.

**Table 1: List of HighResMIP simulations.**

| Source ID | HighResMIP Tier | Integration period | Initial atmospheric condition | Initial land condition |
|---|---|---|---|---|
| **NICAM16-7S** | 1 & 3 | 1950–2050 (101-yr) | ERA-20C (Poli et al., 2016) | NICAM climatology |
| **NICAM16-8S** | 1 & 3 | 1950–2050 (101-yr) | ERA-20C | NICAM climatology |
| **NICAM16-9S** | 1 | 1950–1960 (11-yr) | ERA-20C | NICAM climatology |
| **NICAM16-9S** | 1 | 2000–2010 (11-yr) | ERA-20C | NICAM climatology |
| **NICAM16-9S** | 3 | 2040–2050 (11-yr) | The NICAM16-8S Tier 3 run | The NICAM16-8S Tier 3 run |

In addition, we also did some minor modifications and updated the references.

**Response to RC1**

**RC1-0)** *The paper describes the new version of the global non-hydrostatic could-system resolving model, NICAM. The goal of the paper is twofold (1) evaluate several recent updates which brought the model from version NICAM.12 to the newest NICAM.16; (2) describe additional developments that were made necessary to adapt the model to the HighResMIP (CMIP6 endorsed MIP) protocol and that were introduced in the specific configuration NICAM16-S. The description and evaluation of those changes is valuable, and the paper will undoubtedly serve as a reference for NICAM16-S in studies analysing HighResMIP models. However, the paper does not investigate the impact of model resolution on the model climatology or only marginally. The reader understands only at the end of the paper that it is a deliberate decision and that the impact of resolution will be presented in another paper in preparation. This is a surprising choice, as most people would expect a reference paper of a new model configuration participating in HighResMIP to have horizontal resolution as its main focus. This makes me wonder if the present paper should not be limited to the description of the new model NICAM.16, leaving the developments for CMIP6 to another HighResMIP paper in which the impact of resolution would be investigated in more detail? The quality of the writing in unequal, some sections (e.g, abstract, introduction of section 3 and section 3.1) fall short of meeting the standards of a journal such as GMD, while other sections (e.g. 3.2, 3.3) are written in a very good english. I would recommend a collective effort to improve and homogenise the quality of the text throughout the manuscript. I believe the paper requires major revisions before being published and I would like the authors to answer more specifically the following comments :*

**Response1-0)** Thank you very much for your warm and constructive comments. We agree that horizontal resolution dependency is very interesting, and indeed the HighResMIP mainly focuses on it. However, we believe this paper should mainly focus on the description of the model updates and their impact on the simulated climatology (as suggested by RC2) for rapid publication as a reference of the model, considering many HighResMIP analysis papers will soon need such reference. So, we changed the title to show the focus more clearly (also see **Response1-2**). Also, we homogenized the whole manuscript and further use English proofreading service by native speakers (see the Step 2) after making the following modifications.

**RC1-1)** *Main comments :*

*1) Could you please clarify both in the abstract and in the introduction to which MIP of CMIP6, NICAM will participate? Am I right to understand that they will only participate to HighResMIP and*

*they will not submit simulations to the DECK? The author is left long to speculate about that. The confusion also arises from the fact that CMIP6 and HighResMIP are sometimes used interchangeably (e.g. abstract line 16 vs line 19). I would recommend to use HighResMIP as often as possible, as it is more specific.*

**Response1-1)** Thank you for your comment that will clarify the position of NICAM in CMIP6. Your understanding is correct. NICAM group participates in HighResMIP but will not perform the DECK simulations in CMIP6. Though CMIP6 formally positions NICAM as a submodel of MIROC6 (Tatebe et al. 2019), two models are different in most ways and we will not introduce such formality to avoid further confusion. As suggested by the referee, we

- replaced most of the term *"CMIP6"* with *"HighResMIP"* (line 19 in page 1, lines 26 and 27 in page 2, lines 19 and 29 in page 5, line 14 in page 6, line 28 in page 13, line 15 in page 15, lines 2 and 4 in page 17, item name in page 30),

- deleted *"from CMIP6"* in page 12, line 11,

- added *"High Resolution Model Intercomparison Project (HighResMIP)"* after *"Coupled Model Intercomparison Project Phase 6 (CMIP6)"* in page 1, line 16 (instead of page 1, lines 19-20),

- inserted *"The DECK and CMIP historical simulations* (Eyring et al. 2016) *will not be performed at this time because NICAM is an atmosphere-only model while a coupled ocean-atmosphere model NICAM-COCO* (Miyakawa et al. 2017) *is being developed."* in page 2, line 28, and

- moved *"These model updates…as reported later"* in page 2, lines 28-29 to line 24 and replace *"often (but not always)"* with *"generally"* for logicality.

**RC1-2)** *2) My main issue is that the paper describes a new set-up for HighResMIP but there is absolutely no description of the impact of a change in resolution on the simulated climate. Even section 4, whose title announces an investigation of the dependency to horizontal resolution has only three lines about resolution (l. 7 to 10). Is there a convergence of those statistics when horizontal resolution is increased? I believe you need to inform the reader in the abstract that you do not discuss the impact of resolution.*

**Response1-2)** Thank you for your comments. As we have replied in **Response1-0**, this paper mainly aims to describe the model updates and their impacts on the simulated field. So, instead of rephrasing the abstract, we replaced *"sensitivity experiments"* with *"impacts of model updates"* in the title to clarify the main focus.

Even the resolution dependency is not a main focus of this paper, we admit the original description of

the resolution dependency was not satisfactory for many readers and needs improvement. First, we replaced Tables 5 and 6 with a new figure, Figure R1 below, in response to RC2 (see **Response2-1**). See Table 2 in **Response1-5** for revised run name shown in Figure R1. In the left part of Figure R1, the resolution dependency is graphically shown, although it is difficult to see convergence of the statistics in this narrow resolution range.

Following the referee comment and based on Figure R1 and Figure 11 (Figure 13 in the revised manuscript), we

- added more descriptions of the resolution dependency in Section 4, as *"As we have seen in Figure 13, precipitation pattern in the tropics is strongly resolution-dependent: more dominant double-ITCZ pattern and less intense local precipitation are simulated as the horizontal resolution is increased."* in page 14, line 9,

- further replaced *"Surface air temperature…in total cloud fraction"* in page 14, line 10-11 with *"Surface air temperature is slightly decreased as the resolution is increased (Figure 5a) in association with an increase in total cloud fraction (Figure 5g–i) and a decrease in net downward shortwave radiation at the surface (Figure 5k). Consistent with the precipitation, surface latent heat flux is decreased as the resolution is increased (Figure 5l). Dependency of surface sensible heat flux (Figure 5m) mostly cancels that of the surface latent heat flux."*, and

- added a brief description of precipitation intensity in Section 4, as shown in **Response1-6**.

[Figure]

**Figure R1 (Figure 5 in the revised manuscript): Global annual means of surface air temperature (a), precipitation (b), top-of-atmosphere (TOA) outgoing longwave radiation (OLR) (c), TOA outgoing shortwave radiation (OSR) (d), ice water path (e), liquid water path (f), high cloud amount (g), middle cloud amount (h), low cloud amount (i), surface net downward longwave radiation (j), surface net downward shortwave radiation (k), surface latent heat flux (l), and surface sensible heat flux (m). They are averaged over June 2004 – May 2005. Blue shading shows interannual variability (2σ, detrended) estimated from the HighResMIP NICAM16-7S run over 1950 – 2050 (Table 1). In the left part of each panel, global annual means simulated by NICAM16-7S (56 km mesh; blue), NICAM16-8S (28 km mesh; green), and NICAM16-9S (14 km mesh; red), which were performed under the fixed SST condition (filled circle; the REFFIX run in Table 2) and with the slab ocean condition (filled rectangle; the REFSLB run in Table 2), are plotted. Blue and red lines are the reference (REF) runs with 56 km mesh and 14 km mesh, respectively. Observational values taken from JRA-55 reanalysis (surface air temperature), GPCP (precipitation), CERES (radiation) and ISCCP (cloud amount) are shown as gray lines. In the right part of each panel, Differences between the REF run and each sensitivity run (the NOCLD, NONS, NONSI NOAER, NOALB, NOSIC, and NOGWD runs in Table 2) are shown. Those outside the value range are shown in digit.**

**RC1-3)** *3) The HighResMIP protocol stipulates : "For a clean evaluation of the impact of horizontal resolution, additional tuning of the high-resolution version of the model should be avoided. The experimental set-up and de- sign of the standard resolution experiments will be exactly the same as*

*for the high-resolution runs" (Haarsma et al., 2016). Have you performed specific retuning at each resolution? Please mention explicitly which resolution has been tuned first and what was the procedure and the parameters which were adjusted. In addition, mention any additional tuning specific of each resolution.*

**Response1-3)** Thank you for pointing out the important aspect of the simulations. Of course, we agree to add these descriptions. Though we did not fine-tune the model due to heavy computational cost, we have tuned, albeit in a crude manner, parameters of sea ice thickness with 56 km mesh run and orographic gravity wave drag scheme with 14 km mesh run. All the other parameters in the physics schemes are the same as those described in their original scheme description papers. We did not retune the model at each resolution. Following the suggestions, we

- added *"Though we did not fine-tune the model due to heavy computational cost, we turned, albeit in a crude manner, parameters of sea ice thickness with NICAM16-7S (Section 3.7) and gravity wave drag scheme with NICAM16-9S (Section 3.8). We did not return the model at each resolution under the principles of the HighResMIP."* in page 6, line 29,

- replaced *"Based on an ocean model ... to diagnose ICE from SIC"* in page 12, lines 16-17 with *"Based on an ocean model result (H. Tatebe, personal communication), we performed a series of preliminary annual-scale experiments using NICAM16-7S, with SICCRT values of 1,600 and 3,200, respectively, to improve the surface air temperature over the Arctic. As a result of this crude tuning, SICCRT is set to 1,600 kg $m^{-2}$ in NICAM16-S. This leads to a significant reduction in the warm bias (Figure 11b vs. Figure 11c; blue vs. red lines in Figure 11d) and excess of TOA OLR (blue vs. red lines in Figure 12a) over the Arctic."*, and

- replaced *"it is roughly doubled as the horizontal mesh size is halved"* in page 13, lines 10-11 with *"was tuned first for NICAM16-9S to improve zonal mean zonal wind and then roughly halved as the horizontal mesh size is doubled."*

**RC1-4)** *4) You do not comment on the effect of changing the time step of the radiation scheme in NICAM16-7S to 9S (from 40 to 10min) and changing the time step of the dynamics from (240 to 60s) will have on the climatology (precipitation for instance in Table 6 and figure 11). In addition, and related to the previous question, how will changes in the time step be distinguished from the direct impact of increasing horizontal resolution in HighResMIP? Have you done additional sensitivity experiments? I believe the paper should address this issue.*

**Response1-4)** Thank you for your constructive comments. That's very interesting point. First, we modified the descriptions of the time integration for clarity following RC2 (**Response2-8**), as follows:

*"The time step interval of the dynamics (Δt in Satoh et al. 2008) is set to 4, 2 and 1 min in NICAM16-7S, NICAM16-8S, and NICAM16-9S, respectively. The time loop in the model is based on the dynamics, and physics schemes with a time interval smaller or greater than that of the dynamics are subcycled or skipped, appropriately. Specifically, the time step interval of 30 s is used in the cloud microphysics scheme in NICAM16-7S, NICAM16-8S, and NICAM16-9S. The time interval of 1 min is used in the turbulence (mainly for planetary boundary layer) and land and ocean surface schemes in NICAM16-7S, NICAM16-8S, and NICAM16-9S. The radiation scheme, which requires considerable computational time, is executed every 40, 20, and 10 min in NICAM16-7S, NICAM16-8S, and NICAM16-9S, respectively. Gravity wave drag scheme is called at the same time step of the dynamics."*

We additionally performed 56 km mesh sensitivity experiments, in which the time step interval of the dynamics (including gravity wave drag scheme) was set to 2 min (the DDT2M run) or 1 min (the DDT1M run) and that of radiation to 20 min (the RDT20M run) and 10 min (the RDT10M run) and analyzed their impacts, as Figure R2. We found the impact of the changes in the radiation time interval was negligible from the REFFIX, RDT20M and RDT10M runs. The impact of the time step interval of dynamics, as seen in Figure R2, was large in terms of the low cloud amount and shortwave radiation.

[Figure]

**Figure R2 (Figure 16 in the revised manuscript): Same as the left part of Figure R1 but for the REFFIX, DDT2M, DDT1M, RDT20M, and RDT10M runs, respectively.**

Based on these results, we inserted Figure R2 in the manuscript, added *"The DDT2M, DDT1M, RDT20M, and RDT10M runs were performed to check sensitivity of the simulated climate to the time interval of the model."* in the second paragraph of Section 2.2, and rephrased the overall Section 4, as followings:

*"Understanding the dependency of horizontal resolution is a central interest of the HighResMIP. Figure 16 shows the global mean climate in NICAM16-7S (56 km mesh; blue circle), NICAM16-8S (28 km mesh; green circle), and NICAM16-9S (14 km mesh; red circle), along with a sensitivity of the time step interval of the dynamics (including gravity wave drag scheme) and the radiation scheme in the model. Note that dependency of the time step interval of the radiation scheme is negligible in terms of the global mean climate.*

*Global mean precipitation and TOA OLR are decreased as the horizontal resolution is increased (Figure 16b and c), consistent with a previous study using 3.5–14 km mesh NICAM (Miyakawa and Miura, 2019). The results do not strongly depend on the temporal resolution. As we have seen in Figure 13, precipitation pattern in the tropics is strongly resolution-dependent: more dominant double-ITCZ pattern and less intense local precipitation are simulated as the horizontal resolution is increased. The intense precipitation occurs less frequently in the higher-resolution runs (Figure 17), consistent with Noda et al. (2012) using older NICAM with 14–7 km mesh. The intense precipitation occurs more frequently in the model compared with the GPCP product (Noda et al., 2012), and it is consistent with Maher et al, (2018), who compared precipitation in GCMs without convection scheme with that in the GPCP product. Na et al. (2020) showed that the frequency of intense precipitation in the GPCP product is lower than that in the TRMM product, and 14-km mesh NICAM without convection scheme could realistically reproduce the intense precipitation observed by TRMM.*

*Low cloud amount is substantially underestimated, especially in the NICAM16-7S and NICAM16-8S runs (Figure 16i), leading to the underestimation of TOA OSR (Figure 16d). Consistently, the net downward shortwave radiation at the surface is decreased (Figure 16k) and the surface air temperature is slightly decreased (Figure 16a) as the horizontal resolution is increased. This horizontal resolution dependency of the low cloud amount and its related variables in terms of global mean could be reproduced, albeit overly, by changing the time step interval of the dynamics in the model. The low cloud amount is rather greater in the 56 km mesh run than that in the 14 km mesh run under the fixed time step interval of the dynamics (red circle in the REFFIX tun vs. blue circle in the DDT1M run in Figure 16). Also, the simulated TOA OSR is greater and closer to the CERES product in the 56 km mesh run compared with the 14 km mesh run with the same temporal resolution, though better performance in the simulated global mean TOA OSR in the 56 km mesh run is a result of a strong compensation between a negative bias off the subtropical west coasts of continents and the SH storm-track region and a positive bias in the rest of the lower latitudes (not shown). Such a result of horizontal resolution dependency under the fixed temporal resolution in TOA OSR is similar to Goto et al. (2020), who performed 14 km and 56 km mesh online-aerosol NICAM with the same time step interval of 60 s for the dynamics, turbulence, and surface schemes and 10 s for the cloud microphysics scheme. In HighResMIP, there is no protocol on the temporal resolution of the model, and the horizontal resolution dependency may include the effect of temporal resolution change in the HighResMIP models."*

**RC1-5)** *5) There are four levels of labelling in the paper which makes it sometimes difficult to follow : (1) the different versions of NICAM.12 and 16, (2) the configuration for High-ResMIP NICAM16-S,*

*(3) the various resolutions 7S, 8S, 9S, (4) and the sensitivity experiments (g, f, : : :). Labels are sometimes redundant NICAM16-S and g for instance refer to the same simulations. I believe you might need to keep both (to remain consistent with the naming already communicated to CMIP6) but you need to refer consistently to those different labels throughout the paper and I feel it is not always the case. -> NICAM.16-S rather than NICAM16-S is used in many places. -> Most sensitivity experiments are listed in Table 2 but not all. Could you give an experimental id to simulations described in section 3.5 and column Table 2e? -> The experimental id are not mentioned in the text after section 3.4 (only in tables and captions) whereas they are used in the text before section 3.3. Please can you at least recall once what they are (maybe when you list the sensitivity experiments at the beginning of the section).*

**Response1-5)** Thank you for your constructive comment. In particular, the naming convention of *"g"*, *"f"*, which is internally used for computation and friendly only for us, was very confusing for the readers. So, we

- covered all the sensitivity experiments and rename the run names in more straightforward way such as *"REF"*, *"NOCLD"*, and *"NOAER"* runs, as shown in revised Table 2 below and
- useed these run names in Section 3 and 4, and Figure labels and captions.

*"NICAM16-S"* is the formal name for CMIP6 and all the *"NICAM.16-S"* are replaced with *"NICAM16-S"*.

**Table 2: List of sensitivity experiments.**

| Run name | Descriptions |
| --- | --- |
| REFFIX | Same as NICAM16-S (with the fixed SST condition; Section 3.7). |
| REFSLB | Same as NICAM16-S but with the slab ocean model and nudging (Section 3.7). |
| REF | Alias name of the REFFIX run for 56 km mesh and the REFSLB run for 14 km mesh. |
| NOCLD | Same as the REF run but for using the previous cloud microphysics scheme used in NICAM.12 (Table 5; Section 3.2). |
| NONS | Same as the REF run but for considering only the spherical particle in the radiation table (Section 3.3). |
| NONSI | Same as the NONS run but for removing the interaction between radiation and cloud microphysics (Section 3.3). |
| NOAER | Same as the REF run but for prescribing zero natural and anthropogenic aerosol mass concentration for the radiation scheme and constant CCN of 50 cm$^{-3}$ for the cloud microphysics scheme (Section 3.4). |
| NOANTAER | Same as the REF run but for prescribing zero anthropogenic aerosol mass concentration for the radiation scheme (Section 2.3). |
| NOLND | Same as the REF run but for omitting the effects of wetland and water accumulation on land ice (Section 3.5). |
| NOALB | Same as the REF run but for using the previous surface albedo values (Table 6; Section 3.6). |
| NOSIC | Same as the REF run but for using the previous SICCRT value of 300 kg m$^{-2}$ (Section 3.7). |
| NOGWD | Same as the REF run but for switching off the subgrid-scale orographic gravity wave drag scheme (Section 3.8). |
| DDT2M | Same as the REF run but for setting the time step interval of the dynamics and gravity wave drag scheme to 2 min. |
| DDT1M | Same as the REF run but for setting the time step interval of the dynamics and including gravity wave drag scheme to 1 min. |
| RDT20M | Same as the REF run but for setting the time step interval of the radiation scheme to 20 min. |
| RDT10M | Same as the REF run but for setting the time step interval of the radiation scheme to 10 min. |

*RC1-6) 6) You make the choice not to use a convection scheme. This has been tested in several models at resolutions where convective processes are not yet resolved : please cite references which have tested a similar approach (see for instance Hohenegger et al 2020, https://www.jstage.jst.go.jp/article/jmsj/98/1/98_2020-005/_html/-char/en and references therein). You explain that not having a convective parameterization results in more patchy precipitation (page 6, line 27) and it would be interesting to illustrate that (see for instance Figure 2 in Maher et al, 2018, https://agupubs.onlinelibrary.wiley.com/doi/full/10.1002/2017GL076826)*

**Response1-6)** Thank you for your constructive comment. Agree to modify as suggested. We also made a plot of tropical precipitation intensity, Figure R3, and found more frequent intense precipitation in a coarser resolution model as expected from Noda et al. (2012) using 7-14km NICAM. Also, we confirmed that the frequency of occurrence of precipitation in NICAM is comparable with ConvOff in Figure 2 of Maher et al, (2018). Based on these results, we

- added *"Such approach has also been tested in other researchers besides the NICAM users (Maher et al., 2018; Hohenegger et al., 2020)."* in page 6, line 25,
- inserted Figure R3 after Figure 13, and
- added explanation of Figure R3 as *"The intense precipitation occurs less frequently in the higher-resolution runs (Figure 17), consistent with Noda et al. (2012) using older NICAM with 14–7 km mesh. The intense precipitation occurs more frequently in the model compared with the GPCP product (Noda et al., 2012), and it is consistent with Maher et al, (2018), who compared precipitation in GCMs without convection scheme with that in the GPCP product. Na et al. (2020) showed that the frequency of intense precipitation in the GPCP product is lower than that in the TRMM product, and 14-km mesh NICAM without convection scheme could realistically reproduce the intense precipitation observed by TRMM."* in Section 4.

[Figure]

**Figure R3 (Figure 17 in the revised manuscript): Frequency of occurrence (%) of daily mean precipitation binned with an interval of 1 mm day$^{-1}$ during 01 June 2004 – 31 May 2005 averaged over 15°S–15°N. The REFFIX runs with NICAM16-7S, NICAM16-8S, and NICAM16-9S are shown in black, green, and red lines. The data are re-gridded to 1 degree in longitude and latitude before sampling.**

**RC1-7)** *7) The beginning of the introduction is a bit confused, both climate sensitivity and climate*

*impacts are mentioned and it is unclear why. What not saying from the beginning that the accurate treatment of cloud requires high-resolution cloud resolving models. You could also cite the review paper by Bony et al. 2015, https://www.nature.com/articles/ngeo2398) in this paragraph.*

**Response1-7)** Agree. We simplified the beginning of the introduction by replacing *"Natural disasters ... Shukla et al. 2009)"* in page 2, lines 1-6 with *"The accurate treatment of such interaction between clouds and circulation requires high-resolution global cloud resolving models (Bony et al., 2015; Satoh et al. 2019). <PARAGRAPH GAP> This as well as an increasing demand from society to project tropical cyclones and extremes motivated us to perform..."*

**RC1-8)** *8) Section 2.2: Please be more specific on the initial land conditions in NICAM16-7S and NICAM16-8S. Are they derived in a similar way as NICAM16-9S? this is not clear.*

**Response1-8)** Agree. We replaced *"The initial land condition in the past and present-day simulations was taken from ..."* in page 4, lines 4-5 with *"For the simulations starting from 1 January 1950 or 1 January 2000, the initial land condition prescribed for NICAM16-7S, NICAM16-8S, and NICAM16-9S was taken from ..."*.

**RC1-9)** *9) The change of SICCRT between the AMIP and slab experiments is very large (a factor 5!). Could you please explain if there is any resulting inconsistency between the AMIP and slab experiments for SIC, which I believe is a standard diagnostic of CMIP6?*

**Response1-9)** Sorry for the confusion here. Our original description was not exact and we made clear description of the ocean treatment. The sea ice was fixed to the boundary condition not only in the AMIP experiment but also in the slab experiments. In this study, SICCRT value of 1,600 was used for both Eq. (2) and Eq. (3) in both the AMIP and the slab experiments. So, the SIC is equivalent between the two types of experiments by definition. Only a difference between the AMIP and the slab experiments in this study is the nudging relaxation time of SST (0 for AMIP and 7 days for the slab experiments). Based on these, we

- replaced *"A simple nudging technique ... 7 days"* in page 12, lines 6-7 with *"A simple nudging technique is used to force the predicted SST and ICE toward their reference states with a relaxation time of $\tau_{SST}$ and $\tau_{ICE}$, respectively. Specifically, $\tau_{SST}$=7 days and $\tau_{ICE}$=0 (i.e., ICE was fixed to the boundary condition) were used in the slab ocean experiments of this study and in the previous climate simulation with NICAM.12 (Kodama et al., 2015). Both $\tau_{SST}$ and $\tau_{ICE}$ were set to*

*zero in the fixed SST/ICE experiments including the HighResMIP simulations. <PARAGRAPH GAP> In the slab ocean model implemented ...*",

- deleted *"This fixed SST/SIC ... in the slab ocean model."* in page 12, lines 19-21,
- replaced *"where SICCRT is set to 300 kg m$^{-2}$."* in page 12, line 9 with *"where SICCRT is a parameter in kg m$^{-2}$.",*
- replaced "for Eq. (3), the same as that used in Eq. (2)." in page 12, lines 14-15 with *", considering Eq. (2).",* and
- replaced *"for Eq. (3) in NICAM16-S to diagnose ICE from SIC."* in page 12, line 7 with *"in NICAM16-S".*

**RC1-10)** *10) Page 15 : you indicate that you will share regridded data with CMIP6 at resolution of 1degree or coarser. Will you be able to provide higher-resolution fields on demand? HighResMIP has a special focus at fine scale features, such as tropical cyclones, extreme precipitation, for which high-resolution data might be needed.*

**Response1-10)** Sorry also for the confusion here. The high-resolution data requested by HighResMIP are available through ESGF, and low-resolution data are provided on demand. So, we added *"Note that the high-resolution data requested by HighResMIP are or will be available through the Earth System Grid Federation (ESGF). All the other data (low-resolution, monthly-mean, special variables and so on) are or will be available on request from the corresponding author."* after page 15, line 25. We also found a small error in the code and data availability section and replaced *"HighResMIP Tier 1 (3) simulation data are (and will be)"* in page 16, lines 25-26 with *"HighResMIP product run data are or will be".*

**RC1-11)** *Specific comments :*
*page 1, line 2 : Experimental -> experimental*
*page 1, line 6 : the Coupled Model Intercomparison Project Phase 6*

**Response1-11)** Agree. We modified them as suggested.

**RC1-12)** *page 1, line 17 : the land surface model (and everywhere thereafter)*

**Response1-12)** Agree. We inserted *"surface"* as suggested, specifically, at line 17 in page 1, line 21

in page 2, lines 4, 5, and 12 in page 11, line 7 in page 16, and line 2 in page 45.

**RC1-13)** *page 1, line 18 : an improvement of the coupling*
*page 1, line 19 : and the radiation schemes; ... to follow the protocol of the CMIP6 High ...*
*page 1, line 21 : the impacts of the various model updates*
*page 1, line 23 : over Africa and South Asia*
*page 1, line 29 : redistributes mass*
*page 2, line 22 : non-sphericity of ice particles*

**Response1-13)** Agree. We changed them as suggested.

**RC1-14)** *page 2, line 26 : "That is, the interfaces" => unclear, please rephrase*

**Response1-14)** Agree. We replaced *"NICAM.16 has been further modified ... have been implemented"* in page 2, lines 24-27 with *"NICAM.16 has been further modified to support the external forcings of natural and anthropogenic aerosols and the solar cycle defined in the Coupled Model Intercomparison Project Phase 6 (CMIP6) High Resolution Model Intercomparison Project (HighResMIP) protocol (Haarsma et al., 2016)."* for clarity and simplicity.

**RC1-15)** *page 3, line 9 : NICAM17-nS -> NICAM16-nS.*
*page 3, line 15: including tropical cyclones*
*page 3, line 31: using -> with*
*page 4, line 4: The initial land conditions ... were*
*page 4, line 6: under present-day conditions ... the last 5 years of data*
*page 4, line 14 : are derived from "g" ... 1 une 2004 to ensure consistency with previous NICAM studies*

**Response1-15)** Thank you! Agree. We changed them as suggested. For the last suggestion, we rephrased *"These experiments ... in the previous NICAM studies"* in page 4, lines 14-16 with *"These sensitivity experiments were started from 1 June 2004 and integrated for 1 year. An exception was the NOLND and the REF runs, which were performed for 4 years to make land surface state settle down. The initial date was chosen to ensure consistency with previous NICAM studies"* in association with RC2 (**Response2-9**).

**RC1-16)** *page 4, line 19 : Is SST an external forcing? I don't think it is what people mean by external forcing.*

**Response1-16)** Agree. We replaced *"External forcings"* with *"External forcings and boundary conditions"* in page 4, lines 19 and 20. Also, we replaced *"external conditions"* with *"boundary conditions"* in page 12, line 19.

**RC1-17)** *page 4, line 32 : fixed SST conditions ere used in the 56km meh run*
*page 4, line 33 : was used in the 14 km mesh runs*

**Response1-17)** Agree, but we deleted this part to avoid duplication (see the additional modification in page 3 of this response).

**RC1-18)** *page 5, line 4 : Future change in SST is somewhat similar to the El Nino pattern => personally I don't think so, there is a warming everywhere! You could mention with a larger warming in the equatorial Pacific.*

**Response1-18)** Thank you for your comments. Our description here was crude and necessary to be improved. We replaced *"Future change in SST ... the polar regions."* in page 5, lines 4-6 with *"The SST in the 2040s has larger values almost everywhere compared with that in the 2000s, especially in the midlatitudes, the equatorial eastern Pacific, the tropical Atlantic Ocean, and the edge of the Arctic regions."*

**RC1-19)** *page 5, line 9 : ICE is nearly*
*page 5, line 11 : prescribed in the model*
*page 6, line 8 : a smoother is applied -> a spatial filter is applied to smooth*

**Response1-19)** Agree. We changed them as suggested.

**RC1-20)** *page 6, line 18: could you give a reference for the various schemes?*

**Response1-20)** Agree. We added not only references but also brief descriptions of each scheme following RC2. Please see **Response2-11** for details of the changes.

**RC1-21)** *page 6, line 21 : sentence is too long.*

**Response1-21)** Agree. We simplified page 6, lines 21-25 as *"While most climate models use convection and large-scale condensation schemes even for a mesh size around 14 km, we use the cloud microphysics scheme to represent interactions between clouds and circulation in an explicit way. This not only lowers the cost of development, but also reduces the uncertainty of the results arising from highly arbitrary tuning."*

**RC1-22)** *page 6, line 32 : global means -> global mean climate*

**Response1-22)** Agree. We changed it as suggested.

**RC1-23)** *page 7, line 10 : We used ... -> the sentence is unclear. Do you use it to show improvements or because it shows improvements?*

**Response1-23)** Thank you for your comment. Agree. We replaced *"to show the improvements in the climatology of the NICAM simulation"* in page 7, lines 10-11 with *"and found improvements in the climatology of the NICAM simulation, as will be shown later."*

**RC1-24)** *page 7, line 13 : was originated -> originated*
*page 7, line 21 : is the key -> represent a significant change*
*page 7, line 26 : comparing -> compared*
*page 7, line 32 : accounts for the snow category [this is a typical example where a simple grammar mistake can loose the reader. I thought a new special category was created.]*
*page 8, line 5 : midlatitude storm-track*
*page 8, line 7 : 399¥%, respectively*
*page 8, line 10 : the accretion*
*page 8, line 15 : In addition, the decrease*

*page 8, line 21 : The low cloud amount is increased as a result of a compensation ... in medium and thick clouds and a decrease in thin clouds.*

*page 8, line 28 : check punctuation*

*page 8, line 31 : consistent assumptions of coupling ..... can reduce the model biases*

**Response1-24)** Agree. We changed them as suggested.

**RC1-25)** *page 10, line 8-9 : reference needed.*

**Response1-25)** Agree. For the sake of accuracy and completeness of description, we

- replaced *"Finally, ... high pressure belt"* in page 10, lines 8-9 with *"Finally, NICAM16-S still shows strong negative biases in TOA OLR over the tropical to subtropical regions and TOA OSR over the subtropical high-pressure belt compared with the CERES product (Figure 8, black vs. red lines), and these biases are qualitatively similar to those simulated by NICAM.12 (Kodama et al. 2012).",*

- added *"Unlike NICAM.12, a strong negative bias of TOA OSR is also prominent over the Arctic region, and this seems to relate to an update (reduction) of the surface albedo introduced in Section 3.6."* in page 10, line 14,

- inserted a new figure, Figure R4, after Figure 10 (as Figure 12 in the revised manuscript) and insert *"In terms of the TOA radiation budget, OSR is worsen by a few watt per square meter (Figure 5d), that arises from the polar regions (Figure 12b; green vs. red lines)."* after page 11, line 32, and

- fixed typo by deleting *"become stronger"* in page 10, line 12 and by replacing *"the increased net upward shortwave"* in page 11, line 26 with *"the resulting decreased net upward shortwave"*.

[Figure]

**Figure R4 (Figure 12 in the revised manuscript): Same as Figure 8 (Figure 7 in the original manuscript) but for the NOALB run (green), the NOSIC run (blue), and the REF run (red), respectively, by NICAM16-7S.**

**RC1-26)** *page 11, line 5 : capital letters for the model name.*

**Response1-26)** Thank you. In association with **Response2-11** and **Response1-12**, we replaced *"A land model named as minimal advanced treatments of surface interaction and runoff (MATSIRO; Takata et al. 2003), ..."* with *"The land surface land model, MATSIRO (Takata et al. 2003), ..."*.

**RC1-27)** *page 11, line 7 and line 10 : the model name is NICAM16-S not NICAM.16-S (the same mistake occurs in other places)*

**Response1-27)** Thank you for pointing out the typo. We changed them as suggested.

**RC1-28)** *Figure 9 : is it possible to have an idea of the fractional reduction of the bias?*

**Response1-28)** At this stage, it is difficult to say reduction of the bias only by our results here because of the short integration period, as we had written in page 11, line 16. The only clear result in Figure 9 was an increase in soil moisture, which is consistent with Nitta et al. (2017). The impacts of precipitation and temperature were not so clear. Based on these, we
- deleted Figures 9b and c,
- rephrased the second paragraph of Section 3.4 as *"Figure 10 shows an impact of the land surface model update on soil moisture in boreal summer. The soil moisture is increased over most of the*

*Eurasian and the North American continents as expected from Nitta et al. (2017), particularly in the Siberia and around the Great Lakes. Though it is expected from Nitta et al. (2017) that the increased soil moisture leads to an increase in precipitation and a decrease in surface air temperature, the resulting impacts on the precipitation and surface air temperature are still unclear (not shown). It is difficult to show robust reduction of the biases at this stage, and longer integration is needed to assess these impacts appropriately."*, and

- replaced *"Overall, ... over the continent"* page 16, line 7 with *"Overall, the soil moisture increases over most of the Eurasian and the North American continents."*

**RC1-29)** *page 12, line 5 : The depth of the slab*

**Response1-29)** Could you please comment it again? Seemingly the comment was broken due to a technical reason.

**RC1-30)** *End of section 3.6 : please mention when a result is not shown.*

**Response1-30)** Also, could you please comment it again?

**RC1-31)** *page 13, line 2,3 : the precipitation -> precipitation; line 5 : remove coma*
*page 13, line 8 : is tested -> is used / or / introduced in the model (and please mention that no gravity wave drag was used in NICAM.12)*
*page 13, line 21 : it may not be a wise choice -> introducing such a gravity wave drag scheme will not necessarily lead to an improvement of the simulated climate.*

**Response1-31)** Agree. We changed them as suggested. Also, we added *"No gravity wave drag scheme is used in NICAM.12."* in page 13, line 8.

**RC1-32)** *page 14, line 3-7 : this is a repetition of things which have already been introduced in previous sections.*

**Response1-32)** Agree. We removed this and also removed *"The most noticeable change from the previous simulation in terms of the global mean is the IWP. As described in Section 3.1, IWP is*

*drastically increased by the update of cloud microphysics scheme."* in page 14, line 11-12 to avoid repetition. We changed name of Section 4 from *"Preliminary evaluations with observations including dependency of horizontal resolution"* to *"Horizontal and temporal resolution dependency"* to reflect the above change as well as changes in **Response1-6** and **Response1-4**.

**RC1-33)** *page 14, line 9 : NICAM-7S -> NICAM16-7S*

**Response1-33)** Agree. We changed it as suggested.

**RC1-34)** *page 14, line 19 : greater than SYPD ??? How many?*

**Response1-34)** We replaced *"The actual wall clock time … the Earth Simulator 3"* in page 14, lines 18-20 with *"In NICAM16-8S and NICAM16-9S, the actual SYPD was a few times smaller than the SYPD shown in Table 7 for NICAM16-8S and NICAM16-9S."* Also see **Response1-35**.

**RC1-35)** *page 14, line 20 : please mention the number of cores per node (I guess 4 from the table 9?)*

**Response1-35)** Yes and agree. Following the suggestion and considering readability, we rephrased the first paragraph of Section 5.1 as *"Table 7 shows computational setting and the simulation year per wall-clock day (SYPD) of the simulations by NICAM16-S on the Earth Simulator 3 (NEC SX-ACE). The Earth simulator 3 has 5,120 nodes in total for computation and each computation node has 4 cores. We often use 10, 40, and 160 computation nodes to run NICAM16-7S, NICAM16-8S, and NICAM-9S, respectively, considering a balance between computational efficiency and wall clock time. An exception was NICAM16-8S for the HighResMIP simulation, in which 160 computation nodes were used to finish the 101-year product run within a realistic time. A file staging option …"*.

**RC1-36)** *page 15, line 2 : in an icosahedral grid -> on the model's native icosahedral grid ?*

**Response1-36)** Yes. We changed it as suggested.

**RC1-37)** *page 15 : Summary section : no capital after " : " in this section*

*page 15, line 28 : describe ... model description => this is a bit redundant*

*page 29, line 4 : hygroscopity -> hygroscopicity*

*page 30 : ocean model -> ocean / or / ocean treatment (this is because you mostly use SST)*

**Response1-37)** Agree. We changed them as suggested and removed *"description"* in page 15, line 28 to follow the suggestion. For page 29, line 4, the overall footnote in Table 2 was deleted, as replied in **Response1-5** and **Response2-1**.

**RC1-38)** *page 32 : global mean impacts -> difference of global mean variables between control "g" and sensitivity experiments*

**Response1-38)** Thank you for your comment. Table 5 in page 32 is deleted, as replied in **Response2-1** and **Response1-2**.

**RC1-39)** *page 32 : it would be nice if you could highlight in bold where the difference is statistically significant?*

**Response1-39)** Thank you for your constructive comment. Instead of performing a formal statistical test, we calculated an interannual variability using 101-year NICAM16-7S HighResMIP run (Table 1) and added them as shadings in Figure R1 (Figure 5 in the revised manuscript; see **Response1-2**).

**RC1-40)** page 33 : NICAM.16-S -> NICAM16-S and phrase the legend similarly to Tab 5.

page 34 : NICAM.16-S -> NICAM16-S

**Response1-40)** Thank you. We changed them as suggested.

**RC1-41)** *page 34 : line 5, rephrase the end of the sentence. What does ad hoc mean here?*

**Response1-41)** Thank you. Agree. We replaced *"to reproduce ... ad hoc"* in page 34, lines 5-6 with *"to tune the model to the observed high cloud signals over the tropics."* for clarity.

**RC1-42)** *page 36 : what does Output size in latitude-longitude grid per year TB means here?*

**Response1-42)** Thank you. We deleted it from the table, since they were not used in the body text.

**RC1-43)** page 38 : The same as -> Same as

page 39 : prescribed in the model ; decadal running mean}

**Response1-43)** Agree. We changed them as suggested.

**RC1-44)** *page 40: NICAM16-7 -> NICAM16-7S. Could you add a reference in this figure?*

**Response1-44)** Agree. We
- added *"NOANTAER"* and *"Forcing"* to the title of Figure 4a and b, respectively,
- added a description of NOANTAER run in the revised Table 2 (see **Response1-2**) as *"Same as the REF run but for prescribing zero anthropogenic aerosol mass concentration for the radiation scheme (Section 2.3)."*, and
- modified the caption of Figure 4 as *"Annual mean natural aerosol optical thickness averaged for June 2004 – May 2005 simulated by NICAM16-7S (NOANTAER run in Table 2). ..."*.

**RC1-45)** *page 41 : line 3 : "g3 and g, respectively".*

**Response1-45)** Agree. We changed them as suggested.

**RC1-46)** *page 41 : units of the vertical axis?*

**Response1-46)** Thank you. The unit is kilometer. We added *"km"* in Figure 5 and added *"in km"* between *"altitude"* and *"above sea level"* at the end of the figure caption.

**Response to RC2**

**RC2-0)** *In this manuscript, the authors detail the particular configuration of NICAM used for the High Resolution Model Intercomparison Project (HighResMIP). This is using NICAM16 instead of NICAM12, a previous version used for CMIP-class experiments. Updates are described in components such as microphysics and the land surface. The mean climatology at three different resolutions (56, 28, and 14km) and a few basic sensitivity experiments are discussed. The authors finish by discussing computational performance and post-processing needs.*

*I assume the primary purpose of this manuscript is to detail the particular configuration of NICAM that is used in HighResMIP so it can serve as a reference for scientific papers using such datasets. As such, the paper really doesn't describe any new science; rather, just discusses particular aspects of a specific model configuration. This seems acceptable for a journal such as GMD, even if the results are overly novel from a scientific perspective.*

*I find it to feel somewhat hastily thrown together. Some details regarding NICAM16 are discussed in detail, others are left to the reader to try and track down. Data isn't always presented in the cleanest manner, making jumping from figure to table a bit difficult. Some figures need work, including axis labels and resizing. In some ways, the manuscript feels approximately 75¥% finished, thrown together a bit quickly with some holes that need to be filled and smoothed over before publication. There also is a bit of a mix of 'model description' and then 'high-resolution evaluation,' although the authors then note that more formal climate evaluation is left for future work. I would perhaps focus most of the time in this manuscript on explicitly defining the precise design choices for the contributed runs.*

*I recommend major revisions to clean many pieces of this up and make it more useful as a basic reference for future users of HighResMIP data who wish to learn more about how NICAM operates.*

*The manuscript reads somewhat disjointed, as if multiple authors were e-mailed and asked to 'provide a paragraph or two' and it was eventually stitched together. Some passages are riddled with grammatical errors, while others are much more cleanly written. Although it didn't rise to the level of making the manuscript illegible, I recommend a thorough read-through by one or two proficient English speakers before submission to clean as many of these up before proofreading as possible. Even small corrections to tense and terminology would make for a much more pleasant read.*

**Response2-0)** Thank you very much for your warm and constructive comments. Following the suggestions, we improved the presentation of figures and descriptions of the model as follows. We

tried to homogenize the whole manuscript and further used English proofreading service by native speakers (see the Step 2) after making the following modifications.

**RC2-1)** *Major comments*

*Tables 5 and 6 need to be better presented. I am not sure why Table 5 only shows differences between model simulations and Table 6 shows a mean climatology for the three different resolutions. Without the mean values, the numbers in Table 5 are relatively meaningless, as it is tough to gauge how large the changes are relative to the base (reference) state and whether these changes are moving values towards or away from observations at the global level. The easiest thing to do here would be to effectively combine Tables 5 and 6, with mean climatology presented in additional columns, so it is trivial for the reader to mentally process what the difference in the sensitivity experiments (e.g., g-g3, etc.) actually mean.*

**Response2-1)** Thank you for your constructive comments. We admit Tables 5 and 6 are very confusing and need improvement. One reason for the confusion may arise from the different reference experiments among different sensitivity test. So, we first rerun a few sensitivity experiments (f1d, f1, and f runs in the original manuscript) so that all the impacts could be seen as a deviation from the g runs (hereafter REF runs). Then, naming convention of the sensitivity experiments (e.g. g, g3, g6, …) in Table 2 was changed to more straightforward one such as REF, NOCLD, NOAER, etc., as shown in Table 2 below. We added *"we used the REFFIX run with 56 km mesh and the REFSLB run with 14 km mesh as the reference (REF) runs for the other sensitivity experiments."* and *"Impacts of the model updates described in Section 3 on the simulated climate states were individually tested by switching off each update."* in page 4, around line 13.

**Table 2: List of sensitivity experiments**

| Run name | Descriptions |
|---|---|
| REFFIX | Same as NICAM16-S (with the fixed SST condition; Section 3.7). |
| REFSLB | Same as NICAM16-S but with the slab ocean model and nudging (Section 3.7). |
| REF | Alias name of the REFFIX run for 56 km mesh and the REFSLB run for 14 km mesh. |
| NOCLD | Same as the REF run but for using the previous cloud microphysics scheme used in NICAM.12 (Table 5; Section 3.2). |
| NONS | Same as the REF run but for considering only the spherical particle in the radiation table (Section 3.3). |
| NONSI | Same as the NONS run but for removing the interaction between radiation and cloud microphysics (Section 3.3). |
| NOAER | Same as the REF run but for prescribing zero natural and anthropogenic aerosol mass concentration for the radiation scheme and constant CCN of 50 $cm^{-3}$ for the cloud microphysics scheme (Section 3.4). |
| NOANTAER | Same as the REF run but for prescribing zero anthropogenic aerosol mass concentration for the radiation scheme (Section 2.3). |
| NOLND | Same as the REF run but for omitting the effects of wetland and water accumulation on land ice (Section 3.5). |
| NOALB | Same as the REF run but for using the previous surface albedo values (Table 6; Section 3.6). |
| NOSIC | Same as the REF run but for using the previous SICCRT value of 300 kg $m^{-2}$ (Section 3.7). |
| NOGWD | Same as the REF run but for switching off the subgrid-scale orographic gravity wave drag scheme (Section 3.8). |
| DDT2M | Same as the REF run but for setting the time step interval of the dynamics and gravity wave drag scheme to 2 min. |
| DDT1M | Same as the REF run but for setting the time step interval of the dynamics and including gravity wave drag scheme to 1 min. |
| RDT20M | Same as the REF run but for setting the time step interval of the radiation scheme to 20 min. |
| RDT10M | Same as the REF run but for setting the time step interval of the radiation scheme to 10 min. |

Just merging Tables 5 and 6 leads to a large table and may cause another confusion. So, we instead inserted a new figure, Figure R5, after Figure 4 to graphically summarize impact of the updates in more straightforward way. We added *"Figure 5 (right part of each panel) summarizes impacts of the model changes on the global mean climate. All the significant impacts of the model changes shown here can be qualitatively reproduced even if the analysis period was limited to the last six months (not shown). The REFFIX and REFSLB runs with each horizontal mesh and the observations are shown at the left part of each panel in Figure 5. We will discuss these impacts along with the details of the model updates later in this section."* in page 6, line 31. Also see the related change in **Response2-2**.

[Figure]

**Figure R5 (Figure 5 in the revised manuscript): Global annual means of surface air temperature (a), precipitation (b), top-of-atmosphere (TOA) outgoing longwave radiation (OLR) (c), TOA outgoing shortwave radiation (OSR) (d), ice water path (e), liquid water path (f), high cloud amount (g), middle cloud amount (h), low cloud amount (i), surface net downward longwave radiation (j), surface net downward shortwave radiation (k), surface latent heat flux (l), and surface sensible heat flux (m). They are averaged over June 2004 – May 2005. Blue shading shows interannual variability (2σ, detrended) estimated from the HighResMIP NICAM16-7S run over 1950 – 2050 (Table 1). In the left part of each panel, global annual means simulated by NICAM16-7S (56 km mesh; blue), NICAM16-8S (28 km mesh; green), and NICAM16-9S (14 km mesh; red), which were performed under the fixed SST condition (filled circle; the REFFIX run in Table 2) and with the slab ocean condition (filled rectangle; the REFSLB run in Table 2), are plotted. Blue and red lines are the reference (REF) runs with 56 km mesh and 14 km mesh, respectively. Observational values taken from JRA-55 reanalysis (surface air temperature), GPCP (precipitation), CERES (radiation) and ISCCP (cloud amount) are shown as gray lines. In the right part of each panel, Differences between the REF run and each sensitivity run (the NOCLD, NONS, NONSI NOAER, NOALB, NOSIC, and NOGWD runs in Table 2) are shown. Those outside the value range are shown in digit.**

**RC2-2)** *Page 4, Lines 16-17. Is one year enough to get usable climate signals here? I have generally understood the rule of thumb to be at least a few years, if not a decade to ensure differences are driven*

*by design choices and not internal variability. How are the authors confident they are not confounding these?*

**Response2-2)** Thank you for pointing out an important issue. We added caution in the manuscript. In our experience using NICAM, even a monthly-scale integration is often valuable to see qualitative (and sometimes quantitative) features of the simulated key climatology such as cloud, precipitation, and radiation and their sensitivity to model changes (e.g., Noda et al. 2010; Kodama et al. 2012). Such idea has also been supported by many previous studies (e.g., Phillips et al. 2004; Williams et al. 2013; Hohenegger et al. 2020). Also, we tried to distinguish the impacts of the model changes from internal variability by diagnosing interannual variability simulated by NICAM16-7S. We further confirmed that Figure R5 is not significantly affected by limiting the analysis period to the last six months, as Figure R6 below. Based on these considerations, we

- added *"The integration period of 1 year and even less is sufficient to evaluate basic state of the atmosphere such as cloud, precipitation, radiation, and temperature (e.g., Phillips et al., 2004; Noda et al., 2010; Kodama et al., 2012; Williams et al., 2013; Miyakawa et al. 2018; Miyakawa and Miura, 2019; Stevens et al., 2019; Hohenegger et al., 2020) and tropical variability including diurnal cycle, tropical cyclones, and MJO (e.g., Sato et al., 2009; Kinter et al., 2013; Stevens et al., 2019; Matsugishi et al., 2020). An interannual variability of the NICAM16-7S run (Table 1) over 101 years was diagnosed to distinguish the impacts of the model changes from internal variability in a rough manner."* in page 4, line 16,

- added interannual variability of the 56 km mesh NICAM simulation as shadings in Figure R5 (**Response2-1**), that was inserted in the manuscript as Figure 5 in the revised manuscript and

- added *"We confirmed that all the significant impacts of the model changes shown here can be qualitatively reproduced even if the analysis period was limited to the last six months (not shown)."* in page 7, around line 1.

[Figure]

**Figure R6: Same as Figure R5 but for limiting the analysis period to the last six months (December 2004–May 2005). Note that the shading in Figure R5 is omitted here.**

**RC2-3)** *Page 9, Line 28. I cannot find the g9 simulations in the tables, is there a reason they are not included like the other sensitivity experiments?*

**Response2-3)** Originally, the g9 (and g9a) runs were not included in the Tables 5 and 6 to avoid confusion, because the impact of the model changes was very clear and they had been performed just for three months. Now, the g9 and g9a (the NONS and NONSI runs in the new version; see the revised Table 2 in **Response2-1**) runs were finished for one year, and we added them to Figure R5 and extended Figure 7 to annual mean, as shown below.

[Figure]

**Figure 7 (Figure 8 in the revised manuscript): Annual mean of OLR (a) and OSR (b) at TOA, in W m⁻²,**

**for CERES product (black) and NICAM16-9S runs. Green, blue, and red lines show the NONSI, NONS,**

**RC2-4)** *The naming convention is fairly confusing and there are times when names are redundant and refer to the same simulation (i.e., the 'g' simulation refers to a control run, which is occasionally referred to as NICAM16 or NICAM16-S). Is there a particular reason why these naming conventions are used. Are there ways to simplify this so that they are more clear 'in-text.'*

**Response2-4)** Thank you for your constructive comment. As we have replied in **Response2-1**, we changed the naming convention of the run name. *"NICAM16-S"* is the formal name for CMIP6 and *"NICAM.16-S"* was replaced with *"NICAM16-S"*.

**RC2-5)** *Minor comments*

*Page 2, Lines 6-7. I am not sure exactly what is meant by 'cloud-system resolving climate simulations.' I'd argue cloud-resolving simulations really need to be O(1km). A cloud 'system' may be a larger feature, but I can't recall seeing this as common parlance.*

**Response2-5)** Thank you. The term *"cloud-system resolving"* seems to be ambiguous, and we deleted *"the first cloud-system resolving"* in page 2, line 6.

**RC2-6)** *Page 2, Lines 26-30. Does this mean that NICAM16 is the first NICAM version to allow for transient CMIP forcing, or does this mean special code was added for only HighResMIP/CMIP6?*

**Response2-6)** The latter is correct. NICAM16-S is a special version of NICAM.16 to perform the HighResMIP simulations. We rephrased here for clarity following a comment by RC1 (**Response1-13**).

**RC2-7)** *Page 3, Line 14. 38 vertical levels seems low, particularly for a 14km experiment. Assuming the levels are not evenly spaced, this implies a dz of greater than 1km toward model top, which is really pushing the common notion that dx >> dz. The authors later discuss higher vertical resolution, more information should be added about the potential impact of this in HighResMIP, especially if prior work can be cited.*

**Response2-7)** Agree. We rephrased the 2nd paragraph of Section 2.1 as *"The number of vertical levels is 38, with a model top height of around 40 km, equivalent to the previous climate simulations (Kodama et al., 2015). The interval between each vertical layer increases from 160 m to 2 km as the altitude increases from the ground to 25 km (see K38 setting in Figure 1 of Ohno et al. 2019). Even at such a low vertical resolution, atmospheric phenomena of interests may be practically well simulated including tropical cyclones, MJO, and diurnal precipitation cycle, as we have confirmed in the previous study (Kodama et al., 2015). As a caveat, such coarse vertical resolution in the upper atmosphere leads to an overestimation of the cirrus cloud amount (Seiki et al., 2015b; Ohno et al., 2019) and may cause a different response of high cloud amount to warmer climate* (Ohno et al. 2019). *Also, it has been suggested that the vertical resolution should be increased when the horizontal resolution is increased in terms of atmospheric gravity wave* (Lindzen and Fox-Rabinovitz 1989; Polichtchouk et al. 2019). *Such coarse vertical resolution could overly produce vertical propagation of gravity wave and change zonal wind in the stratosphere* (Watanabe et al. 2015).*"*

**RC2-8)** *Page 3, Line 25. More information is needed about timestep of the gravity wave drag, boundary layer parameterization, etc. Are these called at the same timestep of the dynamics? Is the dynamics subcycled?*

**Response2-8)** Thank you for your comment. The time loop in the model is based on the dynamics, and physics schemes with the time step interval less than that in the dynamics are subcycled. Specifically, the boundary layer parameterization (turbulence) is called four times (NICAM16-7S), twice (NICAM16-8S), and once (NICAM16-9S) after the dynamics is executed. This means the time step interval of turbulence is 60 s for all the horizontal resolution. The gravity wave drag scheme is

called at the same timestep of the dynamics.

We rephrased page 3, lines 23-26 as *"The time step interval of the dynamics ("Δt" in Satoh et al. 2008) is set to 4, 2 and 1 min in NICAM16-7S, NICAM16-8S, and NICAM16-9S, respectively. The time loop in the model is based on the dynamics, and physics schemes with a time interval smaller or greater than that of the dynamics are subcycled or skipped, appropriately. Specifically, the time step interval of 30 s is used in the cloud microphysics scheme in NICAM16-7S, NICAM16-8S, and NICAM16-9S. The time interval of 1 min is used in the turbulence (mainly for planetary boundary layer) and land and ocean surface schemes in NICAM16-7S, NICAM16-8S, and NICAM16-9S. The radiation scheme, which requires considerable computational time, is executed every 40, 20, and 10 min in NICAM16-7S, NICAM16-8S, and NICAM16-9S, respectively. Gravity wave drag scheme is called at the same time step of the dynamics."*

**RC2-9)** *Page 4, Line 10. How quickly does the land spin up from this state? Within days, weeks, months? This may be important given the some of the short runs.*

**Response2-9)** As far as we experienced using legacy NICAM, it takes several years for surface air temperature and soil moisture over the land and OLR to settle down without initialization of land surface model. Even with the initialization adopted here, initial shock, albeit weak, seems to occur in some land variables such as soil moisture. Soil moisture and soil temperature at the uppermost layers of the land surface model in 56 km mesh HighResMIP simulation are shown as Figure R7 below for reference to the referees.

[Figure]

**Figure R7: Soil moisture (left) and soil temperature (right) at the uppermost layers of the land surface model in the HighResMIP NICAM16-7S simulation (Table 1). These are averaged over 20°-120°E and 50°-70°N. Black and green lines show monthly mean and annual running mean, respectively.**

We

- inserted *"This could partly reduce the initial shock of the land surface model, even though it may cost more than several years for some land variables such as soil moisture to fully settle down (not shown)."* in page 4, line 7 and
- added *"the NOLND and the REF runs were performed for 4 years to make land surface state settle down."* as a description of sensitivity experiments in page 4, around line 15.

**RC2-10)** *Page 4, Line 24. Is SST 'standardized' in HighResMIP (i.e., do all models use the same file?) or was this specific to NICAM16? I would also quibble that this is more of a boundary condition than an 'external forcing.'*

**Response2-10)** Yes, all the models including NICAM used the same SST files provided by the HighResMIP, and we deleted "basically" in line 20, page 4. We replaced *"External forcings"* with *"External forcings and boundary conditions"* in lines 19 and 20 in page 4.

**RC2-11)** *Page 6, Lines 18-22. Regarding the dynamical core, diffusion, boundary layer parameterization, etc. it is critical that they at least cite previous work when discussing these aspects where interested parties can get model details. Preferably, they would use 1-2 sentences to explain such components and then refer readers to more detailed publications for further information.*

**Response2-11)** Agree. We rephrased page 6, lines 18-21 as follows:

*"Dynamical core and numerical filters in NICAM16-S are the same as those in NICAM.12. NICAM adopts a fully compressible non-hydrostatic system as governing equations of the dynamics (Tomita and Satoh, 2004; Satoh et al., 2008). The horizontal discretization is icosahedral grid system modified with spring dynamics for homogeneity on the sphere (Tomita et al. 2002). Divergence damping and second order Laplacian horizontal diffusion are used to stabilize the integration (Satoh et al., 2008). Additionally, first order Laplacian horizontal diffusion is applied above 20 km in altitude to avoid spurious wave reflection at the model top.*

*Table 4 shows a summary of the physics schemes used in NICAM16-S and NICAM.12 A single-moment bulk cloud microphysics scheme that solves mass concentrations of water vapor, liquid cloud, ice cloud, rain, snow, and graupel (Tomita, 2008; Roh and Satoh, 2014; Roh et al., 2017) is used instead of a combination of convection and large-scale condensation schemes".*

*Also, we will insert the following lines in page 6, line 28: "A modified version of Mellor-Yamada level 2 scheme (Nakanishi and Niino, 2006; Noda et al., 2010) is used to simulate planetary boundary layer. The radiation scheme, mstrnX (Sekiguchi and Nakajima, 2008), is a broadband model with 29 radiation bands here. The land surface model, Minimal Advanced Treatments of Surface Interaction and RunOff (MATSIRO; Takata et al, 2003) solves land states such as soil temperature, soil moisture, and land surface fluxes. Ocean surface fluxes are calculated following Louis (1979) with a modification of roughness length for strong surface wind conditions (Fairall et al., 2003; Moon et al., 2007). The conventional orographic gravity wave drag scheme (McFarlane, 1987) is used to introduce the effect of vertically-propagating subgrid-scale orographic gravity wave on the momentum tendency of the atmosphere."*

**RC2-12)** *Page 6, Lines 22-25. 'Although most climate models... in the future.' I am not sure I philosophically agree with the notion of removing convective parameterization even at 56km (this would imply extremely large grid point updrafts in my experience). That said, this sentence is long and preferably requires further justification. Has anyone from the NICAM team published a paper regarding their philosophy around the lack of convective parameterizations, even coarser than 20km?*

**Response2-12)** In NICAM team, Seiki et al. (2015) performed 28 and 14 km mesh simulations to study an impact of the vertical resolution on the simulated cirrus cloud. They found a similar vertical resolution dependency between 14 and 28 km mesh simulations. Ohno et al. (2019) also used 28 km mesh model and found a reasonable result on the high cloud response to SST increase. For the 56 km mesh, Takasuka et al. (2018) performed an aqua planet experiment with 56 km mesh NICAM to investigate MJO-like disturbances. As you expected, 56 km simulation produces extremely large grid point updrafts and leads to very intense precipitation as seen in Figure R8 below requested by RC1 (see **Response1-6** for details). Meanwhile, pattern of the time-mean precipitation is well simulated, as shown in Figure 11 in the original manuscript, and such results are also found in Maher et al. (2018) using GCMs with 50 km – a few degrees mesh size. Also see **Response1-21**. Based on these and other previous studies, we replaced *"Although ... (see Section 4)"* in page 6, lines 21-29 with the followings:

*"While most climate models use convection and large-scale condensation schemes even for a mesh size around 14 km, we use the cloud microphysics scheme to represent interactions between clouds and circulation in an explicit way. This not only lowers the cost of development, but also reduces the uncertainty of the results arising from highly arbitrary tuning. Such approach has also been tested in other researchers besides the NICAM group (Maher et al., 2018; Hohenegger et al., 2020). Global*

*mean precipitation is constrained by radiative cooling in large-scale clear-sky regions, which can be captured by the relatively coarse resolution model without the convection and large-scale condensation schemes. The simulated climatology of the precipitation pattern, even with the lowest resolution setting (NICAM16-7S), is comparable with the observation, as shown later, although our choice leads to a patchy behaviour of precipitation and dry/wet bias in the middle/lower troposphere in the simulation (Miyakawa et al., 2018). Similar precipitation behaviour was also reported in a climate model study with a mesh size of around $O(10^2$ km) without convection scheme (Maher et al., 2018). In terms of clouds, Seiki et al. (2015b) performed NICAM simulations with 28 and 14 km mesh to study an impact of the vertical resolution on the simulated cirrus cloud. They found a similar vertical resolution dependency between 14 and 28 km mesh simulations. Ohno et al. (2019) used 28 km mesh NICAM and found a reasonable result on the high cloud response to SST increase compared with a result using 7–14 km mesh NICAM* (Iga et al. 2007). *In terms of MJO, Takasuka et al. (2018) performed an aqua planet experiment with 56 km mesh NICAM to investigate MJO-like disturbances. Yoshizaki et al. (2012) and Takasuka et al. (2015) even performed NICAM with a mesh size larger than 100 km without the convection and large-scale condensation schemes and found MJO-like disturbances in the simulation."*

[Figure]

**Figure R8 (Figure 17 in the revised manuscript): Frequency of occurrence (%) of daily mean precipitation binned with an interval of 1 mm day-1 during 01 June 2004 – 31 May 2005 averaged over 15°S–15°N. The REFFIX runs with NICAM16-7S, NICAM16-8S, and NICAM16-9S are shown in black, green, and red lines. The data are re-gridded to 1 degree in longitude and latitude before sampling.**

**RC2-13)** *Page 7, Lines 32-33. I am not sure this is 'more than twice,' but this is where the aforementioned reformulation of Tables 5 and 6 would be quite helpful.*

**Response2-13)** Thank you. We believe *"more than twice"* becomes more apparent by Figure R5, which was inserted to the manuscript. Please see **Response2-1**.

**RC2-14)** *Page 8, Line 2. '... graupel in the simulation.' Which one, the reference?*

**Response2-14)** We replaced *"simulation"* in page 8, line 2 with *"NOCLD and REF runs"*.

**RC2-15)** *Page 8, Line 19. 'was replaced with zero ... whereas it was zero and unchanged in this study.' I'm a bit confused – the sentence makes it seem like the study applied something different than Roh et al. but it seems like it was zero in both cases?*

**Response2-15)** Thank you. We cleared misunderstanding by replacing *"The cloud ice terminal velocity ... in this study"* in page 8, lines 18-19 with *"Eventually, the cloud ice terminal velocity was set to zero in both Roh et al. (2017) and this study. Unlike this study, Roh et al. (2017) performed their reference run with non-zero cloud ice terminal velocity diagnosed by Heymsfield and Donner (1990), and their comparison before and after the scheme update includes the effect of reduction in cloud ice terminal velocity."*

**RC2-16)** *Page 11, Line 2. This is quite a large resolution sensitivity (the aerosol forcing completely changes sign going from 56km to 14km if I interpret this correctly).*

**Response2-16)** Yes, the sign of the aerosol impact on the net radiation reverses from 56 km to 14 km. This is a result of the compensation between longwave and shortwave components, and sign of the changes of each component seems to be reasonable. We added this explanation in the manuscript. Specifically, we
- added *"This links to a decrease in liquid water path (Figure 5f), which was also found in an online aerosol experiment by NICAM (Sato et al., 2018)."* in page 11, line 1 and
- added *"Such a sign reverse among the resolutions might be related to the resolution dependency of the low and middle cloud amount in the REF run (Figure 5h and i), and a detailed analysis is needed to properly understand the mechanism."* in page 11, line 2.

**RC2-17)** *Fig. 9. This needs to be bigger. Perhaps stack the three panels vertically?*

**Response2-17)** Agree. However, we removed (b) and (c) in response to RC1 (**Response1-28**).

**RC2-18)** *Typographical errors and grammar*

*Page 3, Lines 17-18. Awkward grammar and typos.*

*Page 11, Line 5. The first letters used in the acronym should be capitalized.*

*Table 3. 'Laege' should be 'large.'*

*Fig. 4, The color bar should read 50 and not 50.01.*

*Fig. 5, Label the order of differencing for the lower three panels (e.g., g-g3).*

*Fig. 5., are the units on the vertical axis 'km?'*

**Response2-18)** Agree. We changed them as suggested. For page 3, lines 17-18, we rewrote it as replied in **Response2-7**. For Figure 4, we also modified the caption as *"The lower bound of CCN, 50 cm$^{-3}$ (Section 2.3), is shown in white shading."*. The units of the vertical axis in Figure 5 is km, and we added it to the figure.

**Reference list to RC1 and RC2:**

Eyring, V., S. Bony, G. A. Meehl, C. A. Senior, B. Stevens, R. J. Stouffer, and K. E. Taylor, 2016: Overview of the Coupled Model Intercomparison Project Phase 6 (CMIP6) experimental design and organization. *Geosci. Model Dev.*, **9**, 1937–1958, doi:10.5194/gmd-9-1937-2016. http://www.geosci-model-dev.net/9/1937/2016/.

Goto, D., Y. Sato, H. Yashiro, K. Suzuki, E. Oikawa, R. Kudo, T. M. Nagao, and T. Nakajima, 2020: Global aerosol simulations using NICAM.16 on a 14-km grid spacing for a climate study: Improved and remaining issues relative to a lower-resolution model. *Geosci. Model Dev. Discuss.*, doi:10.5194/gmd-2020-34.

Hohenegger, C., L. Kornblueh, D. Klocke, T. Becker, G. Cioni, J. F. Engels, U. Schulzweida, and B. Stevens, 2020: Climate statistics in global simulations of the atmosphere, from 80 to 2.5 km grid spacing. *J. Meteorol. Soc. Japan.*, **98**, 73–91, doi:10.2151/jmsj.2020-005. https://www.jstage.jst.go.jp/article/jmsj/98/1/98_2020-005/_article.

Iga, S., H. Tomita, Y. Tsushima, and M. Satoh, 2007: Climatology of a nonhydrostatic global model with explicit cloud processes. *Geophys. Res. Lett.*, **34**, L22814, doi:10.1029/2007GL031048.

Kodama, C., A. T. T. Noda, and M. Satoh, 2012: An assessment of the cloud signals simulated by NICAM using ISCCP, CALIPSO, and CloudSat satellite simulators. *J. Geophys. Res. Atmos.*, **117**, doi:10.1029/2011JD017317. http://dx.doi.org/10.1029/2011JD017317.

Lindzen, R. S., and M. Fox-Rabinovitz, 1989: Consistent vertical and horizontal resolution. *Mon. Weather Rev.*, **117**, 2575–2583, doi:10.1175/1520-0493(1989)117<2575:CVAHR>2.0.CO;2. http://journals.ametsoc.org/doi/abs/10.1175/1520-0493%281989%29117%3C2575%3ACVAHR%3E2.0.CO%3B2.

Maher, P., G. K. Vallis, S. C. Sherwood, M. J. Webb, and P. G. Sansom, 2018: The impact of parameterized convection on climatological precipitation in atmospheric global climate models. *Geophys. Res. Lett.*, **45**, 3728–3736, doi:10.1002/2017GL076826. http://doi.wiley.com/10.1002/2017GL076826.

Miyakawa, T., H. Yashiro, T. Suzuki, H. Tatebe, and M. Satoh, 2017: A Madden-Julian Oscillation event remotely accelerates ocean upwelling to abruptly terminate the 1997/1998 super El Niño. *Geophys. Res. Lett.*, **44**, 9489–9495, doi:10.1002/2017GL074683.

Na, Y., Q. Fu, and C. Kodama, 2020: Precipitation probability and its future changes from a global cloud-resolving model and CMIP6 simulations. *J. Geophys. Res. Atmos.*, **125**, doi:10.1029/2019JD031926. https://onlinelibrary.wiley.com/doi/abs/10.1029/2019JD031926.

Nitta, T., K. Yoshimura, and A. Abe-Ouchi, 2017: Impact of Arctic Wetlands on the Climate System: Model Sensitivity Simulations with the MIROC5 AGCM and a Snow-Fed Wetland Scheme. *J. Hydrometeorol.*, **18**, 2923–2936, doi:10.1175/JHM-D-16-0105.1. http://journals.ametsoc.org/doi/10.1175/JHM-D-16-0105.1.

Noda, A. T., K. Oouchi, M. Satoh, H. Tomita, S. Iga, and Y. Tsushima, 2010: Importance of the subgrid-scale turbulent moist process: Cloud distribution in global cloud-resolving simulations. *Atmos. Res.*, **96**, 208–217, doi:10.1016/j.atmosres.2009.05.007. http://linkinghub.elsevier.com/retrieve/pii/S0169809509001550.

——, ——, ——, and ——, 2012: Quantitative assessment of diurnal variation of tropical convection simulated by a global nonhydrostatic model without cumulus parameterization. *J. Clim.*, **25**, 5119–5134, doi:10.1175/JCLI-D-11-00295.1. http://journals.ametsoc.org/doi/abs/10.1175/JCLI-D-11-00295.1.

Ohno, T., M. Satoh, and A. Noda, 2019: Fine vertical resolution radiative-convective equilibrium experiments: roles of turbulent mixing on the high-cloud response to sea surface temperatures. *J. Adv. Model. Earth Syst.*, **11**, 1637–1654, doi:10.1029/2019MS001704. https://onlinelibrary.wiley.com/doi/abs/10.1029/2019MS001704.

Phillips, T. J., and Coauthors, 2004: Evaluating Parameterizations in General Circulation Models: Climate Simulation Meets Weather Prediction. *Bull. Am. Meteorol. Soc.*, **85**, 1903–1916, doi:10.1175/BAMS-85-12-1903. https://journals.ametsoc.org/bams/article/85/12/1903/104943/Evaluating-Parameterizations-in-General.

Polichtchouk, I., T. Stockdale, P. Bechtold, M. Diamantakis, S. Malardel, I. Sandu, F. Vána, and N. Wedi, 2019: Control on stratospheric temperature in IFS: resolution and vertical advection. *ECMWF Tech. Memo.*, **847**, doi:10.21957/cz3t12t7e.

Satoh, M., T. Matsuno, H. Tomita, H. Miura, T. Nasuno, and S. Iga, 2008: Nonhydrostatic icosahedral atmospheric model (NICAM) for global cloud resolving simulations. *J. Comput. Phys.*, **227**, 3486–3514, doi:10.1016/j.jcp.2007.02.006. http://dx.doi.org/10.1016/j.jcp.2007.02.006.

Seiki, T., C. Kodama, M. Satoh, T. Hashino, Y. Hagihara, and H. Okamoto, 2015: Vertical grid spacing necessary for simulating tropical cirrus clouds with a high-resolution atmospheric general circulation model. *Geophys. Res. Lett.*, **42**, 4150–4157, doi:10.1002/2015GL064282. http://doi.wiley.com/10.1002/2015GL064282.

Takasuka, D., T. Miyakawa, M. Satoh, and H. Miura, 2015: Topographical effects on internally produced MJO-like disturbances in an aqua-planet version of NICAM. *SOLA*, **11**, 170–176, doi:10.2151/sola.2015-038. https://www.jstage.jst.go.jp/article/sola/11/0/11_2015-038/_article.

——, M. Satoh, T. Miyakawa, and H. Miura, 2018: Initiation processes of the tropical intraseasonal variability simulated in an aqua-planet experiment: what is the intrinsic mechanism for MJO onset? *J. Adv. Model. Earth Syst.*, **10**, 1047–1073, doi:10.1002/2017MS001243. http://doi.wiley.com/10.1002/2017MS001243.

Tatebe, H., and Coauthors, 2019: Description and basic evaluation of simulated mean state, internal variability, and climate sensitivity in MIROC6. *Geosci. Model Dev.*, **12**, 2727–2765,

doi:10.5194/gmd-12-2727-2019. https://www.geosci-model-dev-discuss.net/gmd-2018-155/.

Watanabe, S., K. Sato, Y. Kawatani, and M. Takahashi, 2015: Vertical resolution dependence of gravity wave momentum flux simulated by an atmospheric general circulation model. *Geosci. Model Dev.*, **8**, 1637–1644, doi:10.5194/gmd-8-1637-2015. http://www.geosci-model-dev.net/8/1637/2015/.

Williams, K. D., and Coauthors, 2013: The Transpose-AMIP II Experiment and Its Application to the Understanding of Southern Ocean Cloud Biases in Climate Models. *J. Clim.*, **26**, 3258–3274, doi:10.1175/JCLI-D-12-00429.1. https://journals.ametsoc.org/jcli/article/26/10/3258/34008/The-TransposeAMIP-II-Experiment-and-Its.

Yoshizaki, M., K. Yasunaga, S. Iga, M. Satoh, T. Nasuno, A. T. Noda, H. Tomita, and M. Fujita, 2012: Why do super clusters and Madden Julian Oscillation coexist over the equatorial region? *SOLA*, **8**, 33–36, doi:10.2151/sola.2012-009.

**Step 2. Changes to homogenize the manuscript and to improve English presentation**

Following the suggestions by the referees, we further revised the manuscript to homogenize it, as summarized below:

- unification of terminology (e.g., "historic" -> "historical", "revised" -> "updated", "TOA OLR" -> "OLR at the TOA" …),
- unification of usage of active and passive sentence (e.g., the first part of "Cloud microphysics" section),
- unification of usage of articles (e.g., "REF run" -> "the REF run"), and
- rearrangement of the abbreviations.

After the above homogenization, we used English text editing service by native English speakers to further improve representation of English. The most of the modifications here were grammatical ones – usage of articles, tenses, passive/active sentences, commas, abbreviations, and so on. Also, some sentences were rephrased to clarify the meanings. These modifications suggested by the text editing service were minor and did not change the meaning of the manuscript. Please confirm the final revised manuscript from the next page for the detailed changes.

**The final revised manuscript**

[revised manuscript text omitted]

---

## Referee Report (RR1)

**Second review of 'The non-hydrostatic global atmospheric model for CMIP6 HighResMIP simulations (NICAM16-S): experimental design, model description, and impacts of model updates'**

I commend the authors for significant improvements to this manuscript relative to the initial submission. In particular, I find the naming convention for the simulations to be vastly improved, which makes it much easier to follow the sensitivity experiments and their impact when updating from NICAM12 to NICAM16. The deeper look into the impact of design choices is welcome. The manuscript also has fewer typographical errors and is a much more pleasant read for an English speaker.

I still have some minor comments that probably should be addressed, but following a response to these few points, I recommend publication in GMD. I anticipate this will be a useful reference for those who apply HighResMIP data to investigate the performance of current-generation Earth system models at high spatial resolution.

**Minor comments**

- Page 3, line 11. Would cite Skamarock et al. [2019].

- Page 4, line 6. Should 'interval' be something like 'time discretization' or 'timestep'?

- Section 3.5. This particular section is a bit weak. I hope a future goal of the team is to explore the response to this more thoroughly as it is a pretty robust change in boreal summer soil moisture from Fig. 10 and likely feeds back into surface enthalpy fluxes, etc.

- Page 11, line 13. Is the 15m a global constant? If so, add 'set to 15m globally.'

- Section 4. Am I interpreting it correctly that there is a large dynamics/GWD timestep sensitivity (DDT2M vs. DDT1M) in the model? If so, this is quite a surprising finding, as generally models are relatively insensitive to dynamical core resolution as long as the model is stable. Does this timestep also impact microphysics, convection, etc.? If so, then that is a more common source of timestep sensitivity. See Gross et al. [2018].

- Fig. 17. The text makes reference to an observational comparison ('The intense precipitation occurs more frequently in the model compared with GPCP.'). It is unclear if the authors meant to add this as a reference to this figure for comparison, but adding either TRMM or GPCP as a fourth line would be a helpful benchmark, especially since the precipitation response runs counter to some other HighResMIP models (where precipitation rate is generally higher as resolution increases, at least until $\Delta x = $O(10km).

- Figure 18. Include the node count for 9s and 7s simulations (looks like perhaps 160 and 10 based on the text?) as is done for the 8s runs.

- Page 15. 'Note that the... author.' This can be moved to the acknowledgements section as it isn't scientifically notable.

- Fig. 16. Add legend showing what the red, blue, and green dots represent on the figure (perhaps in the white space below 16f). While I understand the figure is similar to Fig. 5, it is still useful to include as much description as possible, particularly when the figures do not follow one another.

**Typographical errors and grammar**

This list is not meant to be exhaustive, but rather, a few obvious catches I noted while reading.

- Page 1, line 34. Would probably swap the order and say 'extreme weather, such as tropical cyclones.'

- Page 2, line 2. '... oscillations such as the Madden...'

- Page 2, line 21-23. This passage is a bit awkward. Might just say something like '... no fully coupled atm-ocn models are included in this manuscript as it is under development.' or something similar.

- Page 4, line 20. Formatting of 'SSP5-8.5'

- Page 8, line 3. 'versus synoptic systems'

- Page 12, line 10. '... of the bias, which may be due to factors such as the convective timescale.'

- Would check references that are either 'submitted' or 'accepted' as their treatment as citations in journals varies from publisher to publisher. Not sure what GMD's official stance on this currently is.

**References**

M. Gross, H. Wan, P. J. Rasch, P. M. Caldwell, D. L. Williamson, D. Klocke, C. Jablonowski, D. R. Thatcher, N. Wood, M. Cullen, et al. Physics–dynamics coupling in weather, climate, and earth system models: Challenges and recent progress. *Monthly Weather Review*, 146(11):3505–3544, 2018.

W. C. Skamarock, C. Snyder, J. B. Klemp, and S.-H. Park. Vertical resolution requirements in atmospheric simulation. *Monthly Weather Review*, 147(7):2641–2656, 2019.

---

## Author Response (AR2)

Reply to the 2$^{nd}$ referee comments (RC1 and RC2) on "The non-hydrostatic global atmospheric model for CMIP6 HighResMIP simulations (NICAM16-S): experimental design, model description, and impacts of model updates" [gmd-2019-369], by C. Kodama et al.

Thank you for kindly re-reviewing the manuscript. We modified the manuscript following the suggestions by the two referees, as responded point-by-point below.

We also inserted "simulated by NICAM16-7S (56-km mesh; blue multiplication sign), and NICAM16-9S (14-km mesh; red multiplication sign)" after "The right part of each panel shows differences between the REF run and each sensitivity run (NOCLD, NONS, NONSI NOAER, NOALB, NOSIC, and NOGWD runs in Table 2)" in the caption of Figure 5, following the editor's comment.

**Response to RC1**

**RC1-1)** *page 3, line 6 : I doubt tropical cyclone are simulated accurately at a 40-km resolution. Some particular aspects of TCs could be. Could you be more specific?*

**Response1-1)** This part focuses on the "vertical" resolution issue, and "40-km" is a model top height, not a (horizontal) resolution. We inserted "using a 14-km horizontal mesh" in page 3 line 5 to avoid confusion. We agree that a horizontal mesh size of ~40-km is not enough to simulate various aspects of the tropical cyclones, such as intensity and structure, and that's why we conducted climate simulations with 14-km mesh (Section 1, paragraph 2).

**RC1-2)** *page 13, line 30 : I think the HighResMIP protocol does encourage to keep the same time step and several models which followed the protocol did so.*

**Response1-2)** Yes, the HighResMIP suggested that "The experimental set-up and design of the standard resolution experiments will be exactly the same as for the high-resolution runs.", which is also called "the requirement of no additional tuning" in the HighResMIP protocol paper (Haarsma et al. 2016, GMD). However, it is not so obvious that dt should also be kept constant throughout all the simulations with different dx. We believe some models (like NICAM) may halve dt as dx is halved for computational efficiency, and some may dynamically change dt to further improve the efficiency under

the requirement of the CFL condition. It is impossible to distinguish which model changes / unchanges dt as dx is changed in the HighResMIP dataset.

So, we moderated page 13, lines 30-32 by replacing it with "The HighResMIP requires no additional tuning, that is, the experimental set-up and design should be exactly the same between the standard and high-resolution runs. Meanwhile, changing the time step for different horizontal resolutions is a common approach for modelling centers to reduce the computational resources. Here we highlighted an impact of both the horizontal and temporal resolutions on the global mean climate in the NICAM simulations, suggesting the need to pay more attentions to the configuration of time step in the HighResMIP models."

**Response to RC2**

RC2-0) I commend the authors for significant improvements to this manuscript relative to the initial submission. In particular, I find the naming convention for the simulations to be vastly improved, which makes it much easier to follow the sensitivity experiments and their impact when updating from NICAM12 to NICAM16. The deeper look into the impact of design choices is welcome. The manuscript also has fewer typographical errors and is a much more pleasant read for an English speaker.

I still have some minor comments that probably should be addressed, but following a response to these few points, I recommend publication in GMD. I anticipate this will be a useful reference for those who apply HighResMIP data to investigate the performance of current-generation Earth system models at high spatial resolution.

**Response2-0)** Thank you for your positive comment.

**RC2-1)** *Page 3, line 11. Would cite Skamarock et al. [2019].*

**Response2-1)** Thank you. Added as suggested. We also replaced "over-produce" with "change" in page 3, line 10.

**RC2-2)** *Page 4, line 6. Should 'interval' be something like 'time discretization' or 'timestep'?*

**Response2-2)** Thank you. We replaced "time interval" with "time step" here and also in other parts.

**RC2-3)** *Section 3.5. This particular section is a bit weak. I hope a future goal of the team is to explore the response to this more thoroughly as it is a pretty robust change in boreal summer soil moisture from Fig. 10 and likely feeds back into surface enthalpy fluxes, etc.*

**Response2-3)** Thank you for your suggestion. Though we agree that Section 3.5 is a bit weak, it is still difficult to find robust impact of the surface fluxes from a few-year sensitivity experiments due to insufficient spin-up period for the land model. We will analyze the impact more thoroughly when climate-scale simulation becomes easier in terms of computational resource. Specifically, we will further update the land model (Nitta et al. 2020) in near future and perform in-depth analysis of the simulated result then.

Nitta, T., T. Arakawa, M. Hatono, A. Takeshima, and K. Yoshimura, 2020: Development of Integrated Land Simulator. Progress in Earth and Planetary Science, 7, 68, doi:10.1186/s40645-020-00383-7.

**RC2-4)** *Page 11, line 13. Is the 15m a global constant? If so, add 'set to 15m globally.'*

**Response2-4)** Added as suggested.

**RC2-5)** *Section 4. Am I interpreting it correctly that there is a large dynamics/GWD timestep sensitivity (DDT2M vs. DDT1M) in the model? If so, this is quite a surprising finding, as generally models are relatively insensitive to dynamical core resolution as long as the model is stable. Does this timestep also impact microphysics, convection, etc.? If so, then that is a more common source of timestep sensitivity. See Gross et al. [2018].*

**Response2-5)** Thank you for teaching us very valuable reference. Yes, we also investigated the time step sensitivity issues after the 1st round of the revision, considering in a similar way as Section 2 of Gross et al. (2018).

We found the time step sensitivity seen in the DDT2M and DDT1M runs is related to a type of physics-dynamics coupling issues discussed in Gross et al. (2018) rather than the stability issue of dynamical core alone. In the DDT2M and DDT1M runs, time step of dynamics (and GWD) is halved and

quartered, respectively, compared with the REF run. Because the time loop in the model is based on the dynamics (as described in page 3, line 14), the numbers of sub-cycles of the microphysics, turbulence, and surface processes are also halved and quartered to keep their time step unchanged between the REF run and the DDT2M and DDT1M runs. This means that, during a fixed integration period, the coupling frequency between the dynamics and these physics schemes in the DDT2M and DDT1M runs are twofold and fourfold compared with the REF run. To remove an influence of the increased dynamics-physics coupling frequency, we further performed another short-term sensitivity test (DDT2MS and DDT1MS runs), in which effective time step of dynamics is halved and quartered by introducing sub-cycling in the dynamics routine (without changing nominal time step of dynamics used for model time loop) so that the dynamics-physics coupling frequency is unchanged. We found that the sensitivity to the altered number of sub-cycling of the dynamics seems to be small, as shown in Figure R1.

Based on this fresh tentative result, we inserted "(the DDT2M and DDT1M runs). Specifically, the time loop in the model is based on the dynamics (Section 2.1), and the number of sub-cycles of the microphysics, turbulence, and surface processes were reduced in the DDT2M and DDT1M runs to keep their time steps unchanged (Table 2). This leads to an increase in frequency of the coupling between the dynamics and these physics schemes. Additional monthly-scale sensitivity experiments suggest that the differences between the REFFIX runs and the DDT2M and DDT1M runs are mostly attributed to the altered frequency of physics-dynamics coupling, not the stability issue of the dynamical core (not shown)." in page 13, line 21. We also added more explanation on the DDT2M and DDT1M runs, like "Also, the number of sub-cycles for cloud microphysics, turbulence and surface schemes was halved/quartered to keep their time steps unchanged.", in Table 2.

[Figure]

Figure R1: Same as Figure 16 but for the DDT2MS and DDT1MS runs. Monthly means for June 2004 are plotted.

**RC2-6)** *Fig. 17. The text makes reference to an observational comparison ('The intense precipitation occurs more frequently in the model compared with GPCP.'). It is unclear if the authors meant to add*

*this as a reference to this figure for comparison, but adding either TRMM or GPCP as a fourth line would be a helpful benchmark, especially since the precipitation response runs counter to some other HighResMIP models (where precipitation rate is generally higher as resolution increases, at least until $\Delta x = O(10km)$.*

**Response2-6)** Thank you for your suggestion. We added precipitation intensity from both the TRMM and GPCP products in Figure 17 and appropriately added caption and dataset information.

**RC2-7)** *Figure 18. Include the node count for 9s and 7s simulations (looks like perhaps 160 and 10 based on the text?) as is done for the 8s runs.*

**Response2-7)** Added as suggested. We also fixed a typo on the number of nodes in NICAM16-8S. Thank you!

**RC2-8)** *Page 15. 'Note that the... author.' This can be moved to the acknowledgements section as it isn't scientifically notable.*

**Response2-8)** Thank you. Moved & merged in the acknowledgements section as suggested.

**RC2-9)** *Fig. 16. Add legend showing what the red, blue, and green dots represent on the figure (perhaps in the white space below 16f). While I understand the figure is similar to Fig. 5, it is still useful to include as much description as possible, particularly when the figures do not follow one another.*

**Response2-9)** Thank you. Agreed & modified as suggested. We also added the similar legend in Figure 5.

**RC2-10)** *Typographical errors and grammar*
*This list is not meant to be exhaustive, but rather, a few obvious catches I noted while reading.*
• *Page 1, line 34. Would probably swap the order and say 'extreme weather, such as tropical cyclones.'*
• *Page 2, line 2. '... oscillations such as the Madden...'*

**Response2-10)** Thank you. Modified as suggested.

**RC2-11)** *Page 2, line 21-23. This passage is a bit awkward. Might just say something like '... no fully coupled atm-ocn models are included in this manuscript as it is under development.' or something similar.*

**Response2-11)** This part had been inserted in the 1st round revision in response to the other reviewer's suggestion for clarifying whether the DECK simulation was performed or not. So, we slightly changed the part to introduce your suggestion, like "NICAM.16 is not fully coupled with ocean, and the DECK and CMIP historical simulations (Eyring et al., 2016) were not presented in this study. Note that a coupled ocean-atmosphere model, NICAM-COCO (Miyakawa et al. 2017), is being developed."

**RC2-12)** *Page 4, line 20. Formatting of 'SSP5-8.5'*
• *Page 8, line 3. 'versus synoptic systems'*
• *Page 12, line 10. '... of the bias, which may be due to factors such as the convective timescale.'*

**Response2-12)** Thank you for your careful readings. Modified as suggested.

**RC2-13)** *Would check references that are either 'submitted' or 'accepted' as their treatment as citations in journals varies from publisher to publisher. Not sure what GMD's official stance on this currently is.*

**Response2-13)** Thank you. First, we replaced "Takahashi et al., accepted" with "Takahashi et al. (2020)". We understand your concern on the grey literature and I normally try to avoid "to-be-submitted" reference. However, "Thomason et al. to be submitted" seems to be the only reference to show data source of the CMIP6 stratospheric aerosol. I also checked the journal stance and found "*Works "submitted to", "in preparation", "in review", or only available as preprint should also be included in the reference list*" in the GMD submission guideline.

[revised manuscript text omitted]

---

## Author Response (AR3)

Reply to the editor comments on "The non-hydrostatic global atmospheric model for CMIP6 HighResMIP simulations (NICAM16-S): experimental design, model description, and impacts of model updates" [gmd-2019-369], by C. Kodama et al.

Thank you for your positive evaluation on the manuscript. We agree to modify the manuscript exactly following the suggestions 1) and 2) by the editor, as attached. We also updated the Acknowledgment Section.

[revised manuscript text omitted]

---

## Author Response (AR4)

Reply to the editor comments on "The non-hydrostatic global atmospheric model for CMIP6 HighResMIP simulations (NICAM16-S): experimental design, model description, and impacts of model updates" [gmd-2019-369], by C. Kodama et al.

Thank you for your acceptance decision. We uploaded the files necessary for the production. Note that the Acknowledgment Section was slightly modified, as shown below.

Federation (ESGF). All the other product run data such as low resolution, monthly mean, and special variables and sensitivity experiment data are available on request from the corresponding author.

**Author contributions**

5  CK and ATN managed the overall HighResMIP activity in the NICAM group and prepared the initial and boundary conditions, and MS managed development and scientific activity in the NICAM group. CK added interfaces of the initial and boundary conditions to NICAM. TO added a function to output variables requested by HighResMIP and converted the output data using CMOR3. CK, TO, TS, HY, MN, and YY contributed to the development of NICAM16-S, including debugging, and WS, TN, and DG provided their schemes and/or parameters for the development. CK performed all the

10  HighResMIP simulations and the sensitivity experiments, transferred the data to ESGF, and wrote a major part of this paper. TS wrote most of Sections 3.1 and 3.2 and TS, ATN, DG, HM, and TN modified the manuscript. All the authors provided advice for the development of NICAM16-S and/or experimental design and reviewed the manuscript.

**Competing interests**

15  The authors declare that they have no conflict of interest.

**Acknowledgment**

The authors would like to thank Hiroaki Tatebe and Ryosuke Shibuya for discussions on model configurations and Manabu Abe and Takahiro Inoue for technical advice for CMIP6. CK acknowledges Shunsuke Noguchi for discussions on the

20  vertical resolution of the model. Constructive, and careful comments from two anonymous reviewers and Sophie Valcke, the editor in charge, significantly helped improve the manuscript. CERES, CloudSat, ISCCP, GPCP, and TRMM data were obtained from the National Aeronautics and Space Administration (NASA), JRA-55 data from the JMA, and GridSat data from the National Oceanic and Atmospheric Administration (NOAA). This study was supported by the Environment Research and Technology Development Fund (JPMEERF20172R01) of the Environmental Restoration and Conservation

25  Agency of Japan (ERCA) and the Integrated Research Program for Advancing Climate Models (TOUGOU) (JPMXD0717935457), the FLAGSHIP2020 project within the priority study4, and Program for Promoting Researches on the Supercomputer Fugaku (Large Ensemble Atmospheric and Environmental Prediction for Disaster Prevention and Mitigation) (JPMXP1020351142) of the Ministry of Education, Culture, Sports, Science and Technology (MEXT) of Japan and JSPS KAKENHI Grant Number JP20H05728. The HighResMIP simulations and sensitivity experiments were performed on the

30  Earth Simulator at the Japan Agency for Marine-Earth Science and Technology (JAMSTEC), and some preliminary experiments were performed on the K computer (proposal numbers hp150287, hp160230, hp170234, hp180182, and hp190152).